# Expanding the Chaos: Neural Operator for Stochastic (Partial) Differential Equations

## Abstract

Stochastic differential equations (SDEs) and stochastic partial differential equations (SPDEs) are fundamental tools for modeling stochastic dynamics across the natural sciences and modern machine learning. Developing deep learning models for approximating their solution operators promises not only fast, practical solvers, but may also inspire models that resolve classical learning tasks from a new perspective. In this work, we build on classical Wiener–chaos expansions (WCE) to design neural operator (NO) architectures for SPDEs and SDEs: we project the driving noise paths onto orthonormal Wick–Hermite features and parameterize the resulting deterministic chaos coefficients with neural operators, so that full solution trajectories can be reconstructed from noise in a single forward pass. On the theoretical side, we investigate the classical WCE results for the class of multi-dimensional SDEs and semilinear SPDEs considered here by explicitly writing down the associated coupled ODE/PDE systems for their chaos coefficients, which makes the separation between stochastic forcing and deterministic dynamics fully explicit and directly motivates our model designs. On the empirical side, we validate our models on a diverse suite of problems: classical SPDE benchmarks, diffusion one-step sampling on images, topological interpolation on graphs, financial extrapolation, parameter estimation, and manifold SDEs for flood prediction, demonstrating competitive accuracy and broad applicability. Overall, our results indicate that WCE-based neural operators provide a practical and scalable way to learn SDE/SPDE solution operators across diverse domains.

## 1 Introduction

Stochastic Differential Equations (SDEs) and Stochastic Partial Differential Equations (SPDEs) are fundamental tools for modelling complex dynamical systems across different scientific domains. Examples range from the Ginzburg–Landau equation for superconductivity (Temam, 2012) and the stochastic Navier–Stokes equations for incompressible fluid dynamics in SPDEs (Boyer and Fabrie, 2012) to the Ornstein–Uhlenbeck (OU) process for modelling the velocity of Brownian particles in statistical mechanics (Martin et al., 2021) and the Heston model for financial volatility forecasting (De Spiegeleer et al., 2018).

Recently, these equations have emerged as cornerstones in machine learning. A prominent example is the OU process, whose time-reversal properties now form the theoretical backbone of diffusion generative models, which have achieved remarkable results in image generation (Song et al., 2020), data interpolation (De Bortoli et al., 2021), and time series forecasting (Lin et al., 2024). However, despite their broad applicability, developing deep learning tools for approximating solutions to these equations, or a fundamental understanding of their complex solution structures, remains in its early stages (Neufeld and Schmocker, 2024).

With the recent success of NOs for learning mappings between function spaces (Li et al., 2020; Kovachki et al., 2023), a growing literature has applied NOs to learn the solution operators of SPDEs and SDEs, including Neural SPDE (Salvi et al., 2022), GINO (Li et al., 2023), FNO (Li et al., 2020), and DeepONet (Lu et al., 2019). Their key advantage is one-shot evaluation: solutions at any query time can be obtained from a single forward pass, rather than iterative time stepping. However, incorporating stochasticity remains non-trivial (Salvi et al., 2022). The general strategy is to condition

on the stochastic forcing terms by providing the noise path or its features, thereby reducing the problem to learning a deterministic operator.

Building on this principle, we propose novel NO models for SPDEs and SDEs grounded in the Wiener Chaos Expansion (WCE). By projecting the driving noise onto orthonormal Wick–Hermite features and conditioning on them, our models learn deterministic operators that disentangle stochastic forcing from deterministic dynamics, thereby reconstructing full trajectories. On the theoretical side, we build on classical WCE representations for the class of multi-dimensional SDEs and semilinear SPDEs by explicitly writing down the coupled ODE/PDE systems for their chaos coefficients. In particular, we show how the propagator system takes an explicit ODE form for SDEs driven by multi-dimensional Brownian motion and a corresponding PDE form for SPDEs. We leverage these formulations directly to motivate our NO parameterization. Empirically, we evaluate our models on classical SPDE benchmarks and several SDE-based downstream tasks (diffusion one-step sampling, graph interpolation, financial extrapolation, parameter estimation, and manifold SDEs), achieving competitive accuracy and strong agreement with reference solution operators across all settings.

**Our Contributions**   We develop a NO framework for SPDEs and SDEs that is built on classical Wiener–chaos expansions: solutions are represented via chaos coefficients, and the corresponding deterministic propagator functions are parameterized by neural operators. On the analytical side, we restate and slightly extend known WCE representations by deriving explicit coupled ODE/PDE systems for their Wiener–chaos coefficients. On the empirical side, we apply this framework to classical SPDE benchmarks and many SDE-based downstream tasks, and observe competitive accuracy with one-shot evaluation. To the best of our knowledge, this is the first work to systematically deploy NO models for both SPDE and SDE solution operators across such a diverse and extensive set of experiments.

## 2   MATHEMATICAL PRELIMINARIES

This section introduces the preliminaries for SPDE and SDE, following the standard notation in (Neufeld and Schmocker, 2024; Huschto and Sager, 2014; Eigel and Miranda, 2024).

**SPDE**   Given $T > 0$ and a filtered probability space $(\Omega, \mathcal{F}, \mathbb{F}, \mathbb{P})$, we consider the following semi-linear SPDE[1]:

$$dX_t = (AX_t + F(t, \cdot, X_t))dt + B(t, \cdot, X_t)dW_t, \quad X_0 = \chi_0 \in \mathcal{H}, \qquad \text{(SPDE)}$$

for some linear operator $A : \mathcal{H} \to \mathcal{H}$, operators $F \colon [0, T] \times \Omega \times \mathcal{H} \to \mathcal{H}$, and $B \colon [0, T] \times \Omega \times \mathcal{H} \to L_2(\widetilde{\mathcal{H}}; \mathcal{H})$, where $\mathcal{H}, \widetilde{\mathcal{H}}$ are separable Hilbert Spaces for $X_t$ and $W_t$ respectively and initial value $\chi_0 \in \mathcal{H}$ is deterministic. The process $W \coloneqq (W_t)_{t \in [0, T]} \colon [0, T] \times \Omega \to \widetilde{\mathcal{H}}$ is Q-Brownian that satisfies: (1) $W_0 = 0 \in \widetilde{\mathcal{H}}$; (2) for $t \in [0, T]$, the function $t \to W_t(\omega)$ is continuous for almost all $\omega \in \Omega$; and (3) for any $t_1, t_2 \in [0, T]$ with $t_2 > t_1$, $W_{t_2} - W_{t_1}$ is a centered Gaussian random variable with covariance operator $(t_2 - t_1)Q$ that is self-adjoint, non-negative and has finite trace. Under mild assumptions on Lipschitz & linear-growth (see Appendix B for details), a $\mathbb{F}$-predictable process $X \colon [0, T] \times \Omega \to \mathcal{H}$ is a *mild solution* of (SPDE) if: $\int_0^t \|X_t\|_{\mathcal{H}}^2 < \infty$; and for every $t \in [0, T]$,

$$X_t = S_t \chi_0 + \int_0^t S_{t-s} F(s, \cdot, X_s) ds + \int_0^t S_{t-s} B(s, \cdot, X_s) dW_s, \qquad (1)$$

holds almost surely. Here $S = e^{tA} : \mathcal{H} \to \mathcal{H}$ refers to a $C_0$-semigroup generated by operator $A$. In addition, we will denote a *single realization* of the solution process for a fixed random event $\omega \in \Omega$ by $u(t, x)$, where $x \in \mathcal{D}$ is the spatial variable.

**SDE**   The SDE is a special form of SPDE with $A = 0$ and $\mathcal{H}, \widetilde{\mathcal{H}}$ reduced to finite-dimensional vector spaces. Specifically,

$$dX_t = F(t, X_t)dt + B(t, X_t)dW_t, \quad X_0 = x_0 \in \mathbb{R}^d, \qquad \text{(SDE)}$$

where $W_t$ is an $m$-dimensional Brownian motion with covariance $Q \in \mathbb{R}^{m \times m}$ (i.e., $\mathbb{E}[(W_{t_2} - W_{t_1})(W_{t_2} - W_{t_1})^\top] = (t_2 - t_1)Q$), and $F(t, \cdot) \colon \mathbb{R}^d \to \mathbb{R}^d$, $B(t, \cdot) : \mathbb{R}^d \to \mathbb{R}^{d \times m}$ are known as the

---

[1]An SPDE is linear if both $A, F, B$ are linear, or semi-linear if $A$ is linear but $F, B$ are non linear.

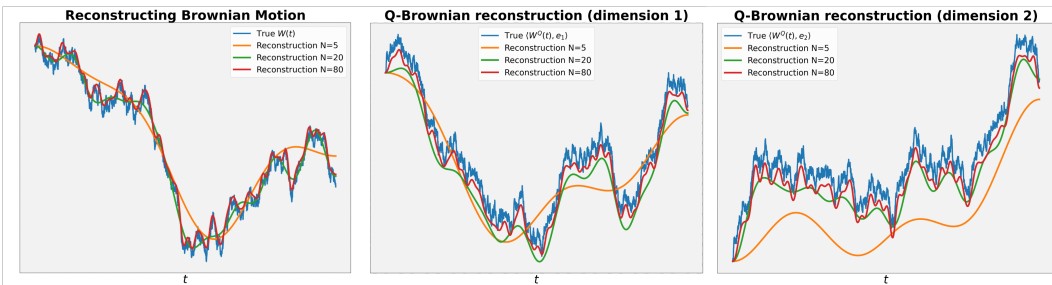

Figure 1: Reconstruction of a one-dimensional Brownian motion and a two-dimensional Q-Brownian motion from truncated chaos expansions with different truncation orders $n$. The blue curve shows the true trajectory, while the coloured curves show reconstructions using $n$ temporal modes. For very small $n$ (e.g., $n = 5$) the approximation is intentionally coarse and appears oversmoothed, especially in the second coordinate of the Q-Brownian motion, but for larger $n$ the reconstructions closely track the true paths, consistent with Lemmas 1 and 2.

drift and diffusion term of the SDE, respectively. Without loss of generality, in this work, we only consider the standard Brownian motion, i.e., $Q = I$, yet our conclusions can be naturally extended to general cases. The strong solution of (SDE) is

$$X_t = x_0 + \int_0^t F(s, X_s)ds + \int_0^t B(s, X_s)dW_s. \tag{2}$$

Analogous to SPDE, we denote a single realization of the solution process $X_t$ by the deterministic vector-valued function $u(t)$.

**Problem setup.** We aim to learn data-driven solution operators associated with (SPDE) and (SDE). Let $X = (X_t)_{t \in [0,T]}$ denote the mild (for (SPDE)) or strong (for (SDE)) solution, and write $u(t, x)$ (or $u(t)$ in the SDE case) for a single realization of $X_t$. The corresponding (sample-wise) solution operator can be written abstractly as $\mathcal{S} : (\chi_0, W) \mapsto u(\cdot)$, where $\chi_0$ is the (generic) initial condition and $W$ is the driving Brownian motion (or $Q$–Brownian motion in the SPDE setting). In practice, we are given a finite training set consisting of $N$ independent realizations of the underlying stochastic system, i.e., $\mathcal{D} = \{(\chi_0^{(i)}, W^{(i)}, u^{(i)})\}_{i=1}^N$, where each $u^{(i)}$ is the numerically simulated solution trajectory on a fixed space–time (or time) grid corresponding to the initial condition $\chi_0^{(i)}$ and the driving noise $W^{(i)}$. Our goal is to learn a parameterized operator, that is $\widehat{\mathcal{S}}_\theta : (\chi_0, \eta) \mapsto \hat{u}(\cdot)$, that approximates $\mathcal{S}$, where $\eta$ denotes a finite–dimensional representation of the driving noise (constructed from $W$ via a Wiener–chaos based projection; see Wick polynomials in Eq. (3)). The model parameters $\theta$ are obtained by minimizing an empirical $L^2$–type discrepancy between $\hat{u}$ and $u$ over $\mathcal{D}$, which in implementation corresponds to a MSE loss over the discrete space–time grid.

## 3 WIENER CHAOS EXPANSION OF SPDEs AND SDEs

In this section, we recall the WCE for SDEs and SPDEs and adapt it to the setting of this paper. We first review how Brownian and Q-Brownian motions can be represented via countable Gaussian coordinates constructed from Brownian increments, and then show how this representation leads to WCEs of SDE/SPDE solutions. Our presentation follows the classical theory (Luo, 2006); we restate and slightly refine the main ingredients needed for our NO construction, in particular by making the associated propagator systems explicit, all proofs are in Appendix C for completeness.

### 3.1 RECONSTRUCTION OF BROWNIAN MOTIONS

We begin by illustrating some standard conclusions on reconstruction of Brownian and Q-Brownian motions through their increments (Da Prato and Zabczyk, 2014).

**Definition 1** (Gaussian Random Variable). *Let $\{e_j\}_{j \in \mathbb{N}} \subset L^2([0,T])$ be a complete orthonormal family that is continuous almost everywhere. Let $W = (W^{(1)}, \ldots, W^{(d)})$ be a $d$-dimensional standard Brownian motion on $(\Omega, \mathcal{F}, \mathbb{F}, \mathbb{P})$. For every $i \in \{1, \ldots, d\}$ and $j \in \mathbb{N}$, define $\xi_{ij} :=$*

$\int_0^T e_j(s)\,dW_s^{(i)}$. *Then the collection* $\{\xi_{ij}\}_{i,j}$ *consists of mutually independent standard Gaussian random variables* (Neufeld and Schmocker, 2024).

**Lemma 1** (Reconstruction of Brownian Motion)**.** *Let the functions* $G_j(t) := \int_0^t e_j(s)\,ds$ *for* $t \in [0,T]$. *For each* $i \in \{1,\dots,d\}$ *and truncation level* $n \in \mathbb{N}$, *define the approximating process* $\widehat{W}_t^{(n,i)} := \sum_{j=1}^n \xi_{ij} G_j(t)$. *Assume* $\sum_{j=1}^\infty \|G_j\|_{C([0,T])}^2 < \infty$, *then the sequence of* $(\widehat{W}^{(n,i)})_{n\in\mathbb{N}}$ *converges uniformly to* $W^{(i)}$ *almost surely.*

We now extend our conclusion to show how the Q-Brownian motion in $\mathcal{H}$ can be reconstructed.

**Lemma 2** (Reconstruction of Q-Brownian Motion)**.** *Let* $\{(\lambda_k, f_k)\}_{k\in\mathbb{N}}$ *be the orthonormal eigensystem of* $Q$. *The Q-Brownian* $W$ *admits the Karhunen-Loève expansion* $W_t = \sum_{k=1}^\infty \sqrt{\lambda_k} \beta_t^k f_k$, $t \in [0,T]$, *where* $\{\beta_t^k\}_{k\in\mathbb{N}}$ *is a sequence of independent, one-dimensional standard Brownian motions. Let* $\{e_j\}_{j\in\mathbb{N}} \subset L^2([0,T])$ *be the basis defined in Definition 1, such that each scalar Brownian motion* $\beta^k$ *can be represented as* $\beta_t^k = \sum_{j=1}^\infty \xi_{kj} G_j(t)$. *By truncating the series for both* $\beta^k$ *and* $W$, *we define the approximation for* $K, n \in \mathbb{N}$ *as* $\widehat{W}_t^{(K,n)} := \sum_{k=1}^K \sqrt{\lambda_k} \left( \sum_{j=1}^n \xi_{kj} G_j(t) \right) f_k$. *Assume* $\sum_{j=1}^\infty \|G_j\|_{C([0,T])}^2 < \infty$, *then the approximation converges to* $W$ *in* $C([0,T]; \mathcal{H})$ *almost surely.*

Figure 1 shows simple examples of the reconstructions of both Brownian and Q-Brownian motions. One can check that as the truncation order increases, the reconstructed paths rapidly approach the true trajectories in all coordinates. Since the SDE/SPDE solutions we consider are square-integrable functionals of $W$, they can subsequently be expanded in the corresponding Wiener–chaos basis, which is the starting point for the WCE-based approximations developed in the next subsection.

## 3.2 Chaos Expansion on SPDEs and SDEs

We now recall how the SPDEs, SDEs solutions can be coupled with a polynomial basis constructed from the Gaussian random variables introduced above. We start by defining the Wick polynomials.

**Definition 2** (Wick Polynomial)**.** *Let* $\Xi = \{\xi_{mj}\}_{m,j\in\mathbb{N}}$ *be the family of Gaussian random variables constructed from Definition 1. The index* $m$ *corresponds to the spatial/component mode, while* $j$ *relates to the temporal basis. Let* $\mathcal{J}$ *be the set of all multi-indices* $\alpha = (\alpha_{mj})_{m,j\in\mathbb{N}}$ *with finite support. For any* $\alpha \in \mathcal{J}$, *the corresponding normalized Wick monomial is*

$$\xi_\alpha(\omega) := \frac{1}{\sqrt{\alpha!}} \prod_{m,j} h_{\alpha_{mj}}(\xi_{mj}(\omega)), \tag{3}$$

*where* $h_k$ *is the* $k$-*th (probabilist's) Hermite polynomial,* $h_k(x) = (-1)^k e^{x^2/2} \frac{d^k}{dx^k} e^{-x^2/2}$, *and* $\alpha! := \prod_{m,j} \alpha_{mj}!$. *A finite linear combination of such monomials is called a Wick polynomial.*

The family $\{\xi_\alpha\}_{\alpha\in\mathcal{J}}$ forms a complete orthonormal basis for the space of square-integrable random variables measurable with respect to $\Xi$, i.e., $\mathbb{E}[\xi_\alpha \xi_\beta] = \delta_{\alpha\beta}$, $\forall \alpha, \beta \in \mathcal{J}$. Furthermore, the linear span of this basis is *dense* in $L^p(\Omega, \mathcal{F}, \mathbb{P})$ for any $p \in [1,\infty)$ (Neufeld and Schmocker, 2024). This completeness is fundamental, as it allows any random variable in this space to be represented as a series expansion in this basis. This representation effectively transforms stochastic equations into an infinite system of deterministic equations for the coefficients of the expansion. We first apply this framework to the case of SDEs before generalizing to the more complex setting of SPDEs.

**Theorem 1** (SDE Propagator System)**.** *Let* $X_t \in \mathbb{R}^d$ *be the strong solution to the* $d$-*dimensional SDE* (SDE) *driven by an* $m$-*dimensional Brownian motion* $W_t$. *The solution's WCE is given by*

$$X_t = \sum_{\alpha\in\mathcal{J}} u_\alpha(t) \xi_\alpha, \tag{4}$$

*where the propagators are* $u_\alpha(t) := \mathbb{E}[X_t \xi_\alpha] \in \mathbb{R}^d$. *The evolution of these propagators is governed by a coupled system of ODEs. For* $j \in \{1,\dots,d\}$, *the ODE for* $u_\alpha^{(j)}(t)$ *is:*

$$\frac{d}{dt} u_\alpha^{(j)}(t) = \mathbb{E}\left[ F^{(j)}(t, X_t) \xi_\alpha \right] + \sum_{k=1}^m \sum_{i=1}^\infty \sqrt{\alpha_{ki}}\, e_i(t)\, \mathbb{E}\left[ B^{(j,k)}(t, X_t) \xi_{\alpha-e_{ki}} \right], \tag{5}$$

*with* $u_\alpha^{(j)}(0) = x_0^{(j)} \delta_{\alpha 0}$, *and* $e_{ki}$ *denotes the multi-index that is 1 at position* $(k,i)$ *and 0 elsewhere.*

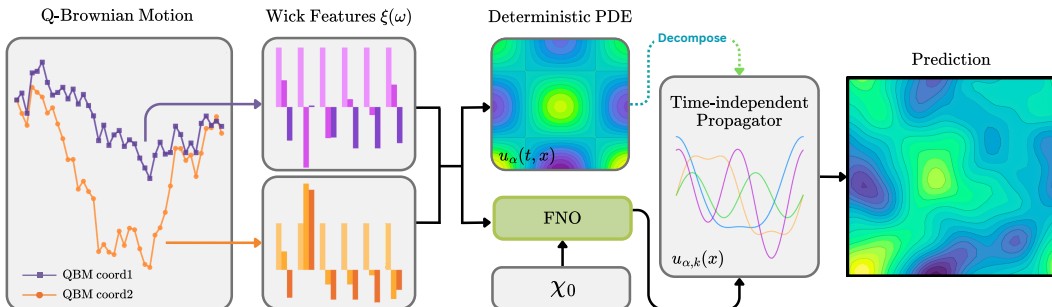

**Figure 2:** General working flow of $\mathcal{F}$-SPDENO: a discrete Q-Brownian path is simulated, Wick features computed from its increments are concatenated with the initial condition $\chi_0$ and passed through an FNO to produce time-independent propagator fields, which are combined with a fixed temporal basis to reconstruct the full trajectory.

The one-dimensional version of this propagator system for (SDE) has been used in the SDEONet framework of (Eigel and Miranda, 2024). Building on classical WCE results, here we write out the multi-dimensional Brownian case explicitly: the chaos indices now carry both a component index $k$ and a temporal basis index $i$, and the right-hand side of Eq. (5) depends only on $t$ (and on fixed model coefficients), not on the randomness $\omega$. Each coefficient $u_\alpha(t)$ therefore solves a deterministic ODE in time, which makes these propagators natural targets for neural networks that take $t$ as input. The same idea extends to SPDEs: Theorem 2 shows that the chaos coefficients become deterministic space–time fields $u_\alpha(t, x)$ satisfying a coupled system of PDEs.

**Theorem 2** (SPDE Propagator System). *The evolution of propagator fields $u_\alpha(t, x)$ for (SPDE) is governed by a coupled system of deterministic PDEs*

$$\frac{\partial}{\partial t} u_\alpha(t, x) = (Au_\alpha)(t, x) + \mathscr{F}_\alpha(\{u_\gamma(t, x)\}_{\gamma \in \mathcal{J}}), \tag{6}$$

*with initial condition $u_\alpha(0, x) = \chi_0(x)\delta_{\alpha 0}$, and the nonlinear term $\mathscr{F}_\alpha$ is an operator uniquely determined by the nonlinearities $F$ and $B$.*

As discussed in the SPDE WCE literature (see, e.g., Remark 2.13 in Neufeld and Schmocker, 2024), and (Lototsky and Rozovskii, 2006a) the chaos coefficients of semilinear SPDEs satisfy a coupled system of deterministic evolution equations. Theorem 2 makes this structure explicit for the class of SPDEs in (SPDE) by writing the propagator system in the form (6); examples and concrete expressions for $\mathscr{F}_\alpha$ are given in Appendix C. While the precise form of $\mathscr{F}_\alpha$ depends on the choice of $F$ and $B$, the key point for our purposes is that each $u_\alpha(t, x)$ solves a deterministic PDE in space–time. This observation enables us to use standard deterministic PDE neural operators such as FNO (Li et al., 2020) and GINO (Li et al., 2023) to approximate the propagator fields, with the stochasticity handled entirely by the Wick features of the noise trajectories.

## 4 MODEL ARCHITECTURE

$\mathcal{F}$-**SPDENO** As discussed above, solutions of (SPDE) admit a Wiener–chaos representation $X_t(x) = \sum_{\alpha \in \mathcal{J}} u_\alpha(t, x) \xi_\alpha$, so that once the Q-Brownian motion $W$ and the initial condition $\chi_0$ are observed, the randomness is entirely carried by the chaos variables $\{\xi_\alpha\}$, while the propagators $u_\alpha(t, x)$ remain deterministic. In practice, we simulate a discrete Q-Brownian path on a time grid, compute its increments, and from them construct the Gaussian variables $\{\xi_{mj}\}$ and Wick monomials $\{\xi_\alpha\}$; a truncated collection of $\{\xi_\alpha(W)\}$ is stacked into the Wick feature vector $\eta(W)$ (Section 2), which serves as the stochastic input to our model. To make the propagators suitable NO targets, we further expand them in a temporal basis $\{\phi_k(t)\}_{k=1}^K$ as $u_\alpha(t, x) \approx \sum_{k=1}^K u_{\alpha,k}(x) \phi_k(t)$, so that we only need to learn the time-independent coefficient fields $\{u_{\alpha,k}(x)\}$. $\mathcal{F}$-SPDENO then uses an FNO with parameters $\theta$ to approximate the deterministic operator $(\chi_0, \eta(W)) \mapsto \{u_{\alpha,k}(x)\}$, and the final prediction is obtained by combining the FNO outputs with the chaos and temporal bases. Figure 2 summarises this workflow: the block labelled "Deterministic PDE" is a conceptual illustration of the propagator system in Theorem 2, but in practice we do not solve this PDE system explicitly and instead train the FNO to approximate the mapping $(\chi_0, \eta(W)) \mapsto \{u_{\alpha,k}(x)\}$ directly.

Table 1: Relative $L^2$ error on the dynamic $\Phi_1^4$, – indicates that the model is not applicable.

| Model | #Parameters | Inference time (s) | $N = 1000$ | | $N = 10000$ | |
|---|---|---|---|---|---|---|
| | | | $W \mapsto X$ | $(X_0, W) \mapsto X$ | $W \mapsto X$ | $(X_0, W) \mapsto X$ |
| NCDE | 272 672 | 0.503 | 0.134 | 0.168 | 0.050 | 0.064 |
| NRDE | 2 164 356 | 0.105 | 0.129 | 0.150 | 0.058 | 0.089 |
| NCDE-FNO | 48 769 | 1.845 | 0.077 | 0.061 | 0.091 | 0.060 |
| FNO | 1 647 105 | 0.165 | 0.029 | – | 0.034 | – |
| NSPDE | 265 089 | 0.248 | 0.021 | 0.020 | 0.013 | 0.021 |
| $\mathcal{F}$-SPDENO (Ours) | 108 629 | 0.128 | **0.012** | **0.015** | **0.011** | **0.017** |

**SDENO** For the SDE case, we follow the same chaos-based representation and treat the propagators as deterministic functions of time that can be approximated by standard neural networks. As illustrated in the previous section, once the Brownian motion has been encoded into Wick features, each chaos coefficient $u_\alpha(t)$ satisfies a deterministic ODE in $t$, so one can use a wide range of architectures (even simple MLPs) to approximate these time-dependent propagators. Concretely, we construct Wick features from the driving noise trajectory $\{W_t\}_{t \in [0,T]}$ and feed the time variable $t$ (or its positional encoding) into a backbone network, whose outputs are then combined with the Wick features via an explicit inner product. This yields a flexible SDENO family with different backbones: for example, we use a UNet for diffusion SDEs on images ($\mathcal{U}$-SDENO) and graph neural networks for diffusion interpolation on graphs ($\mathcal{G}$-SDENO). In $\mathcal{U}$-SDENO, the time dependence is modeled inside the UNet, whereas in $\mathcal{G}$-SDENO it is handled by a separate temporal network. Further architectural details and additional experiments for the SDE settings are provided in the appendix.

## 5 NUMERICAL EXPERIMENTS

In this section, we present a comprehensive empirical evaluation of our neural operator (NO) models across five settings: (i) solving SPDEs (the dynamic $\Phi_1^4$ model and the 2D Navier-Stokes equation); (ii) solving SDEs in the image and graph domains, yielding a diffusion one step sampler and a topological interpolator; (iii) extrapolation on the financial Heston model; (iv) parameter estimation with our generalized *Meta*-SDENO model; (v) learning SDE solution on manifold. All experiments are executed on an NVIDIA® H200 SXM GPU (141 GB HBM3e) within an HPC cluster. Additional experimental details and results, e.g., empirical parameter sensitivity, are provided in Appendix F.

### 5.1 APPROXIMATION OF SPDE SOLUTIONS

**Dynamic $\Phi_1^4$ Model**   We begin to evaluate our $\mathcal{F}$-SPDENO model by the dynamic $\Phi_1^4$ model, which is widely applied in the realm of superconductivity (Temam, 2012). The equation has the form

$$dX_t = \left(\Delta X_t + 3X_t - X_t^3\right) dt + dW_t, \ X(0,x) = \chi_0 \in L^2(\mathbb{T}^1), \quad X_t(0) = X_t(1), \quad (7)$$

where $(t,x) \in [0,T] \times [0,1]$, and $W$ is the Q-Brownian motion. We follow the experimental settings of Salvi et al. (2022); Hu et al. (2022) in generating sample paths, initial conditions, ground truth solutions, and train-test splits. We conduct two tasks with different training observations $N = 1000$ and 10000, respectively, and evaluate our model with two settings, (i) $W \mapsto X$ which assumes $W$ is observed but the initial condition is fixed all the time and (ii) $(X_0, W) \mapsto X$ where $W$ is observed and $X_0$ changed across the samples. We compare $\mathcal{F}$-SPDENO with various baselines such as NCDE, NCDE-FNO, NRDE (Lu et al., 2022), FNO (Li et al., 2020) and NSPDE (Salvi et al., 2022). We note that we did not compare our model to architectures specifically designed for rough or even singular SPDEs, such as (Hu et al., 2022; Gong et al., 2023). In the one-dimensional dynamic $\Phi_1^4$ setting considered here, the solution enjoys positive Hölder regularity (Li et al., 2025), so the regularity-structure machinery that underpins these models is not essential. The results are contained in Table 1. Our model achieves remarkable results relative to all other baselines, with lower $L^2$ errors across both small and large training schemes. Interestingly, increasing $N$ from 1000 to 10000 only leads to modest changes in the relative $L^2$ error for NSPDE and $\mathcal{F}$-SPDENO, especially on the $(X_0, W) \mapsto X$ task. This behaviour is consistent with the fact that both methods already achieve very small errors in the $N = 1000$ regime. In $\Phi_1^4$, $N = 1000$ trajectories already capture most of the variability of the dynamics, so increasing $N$ to 10000 brings only limited additional benefit. This aligns with the recent benchmarking study in SPDE (Li et al., 2025).

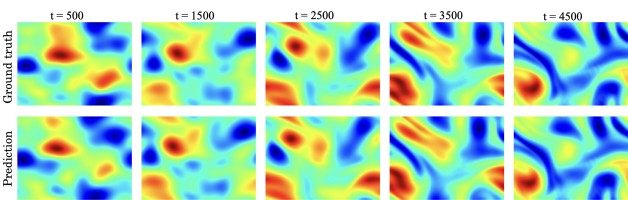

Figure 3: Comparison between ground truth and $\mathcal{F}$-SPDENO solutions of the 2D stochastic Navier–Stokes equation at several time instances.

Table 2: Relative $L^2$ error for the 2D NS equation.

| Model | $W \mapsto X$ | $(X_0, W) \mapsto X$ |
|---|---|---|
| NCDE | 0.406 | 0.893 |
| NCDE-FNO | 0.379 | 0.208 |
| FNO | 0.051 | 0.073 |
| NSPDE | 0.040 | 0.047 |
| $\mathcal{F}$-SPDENO (Ours) | **0.037** | **0.031** |

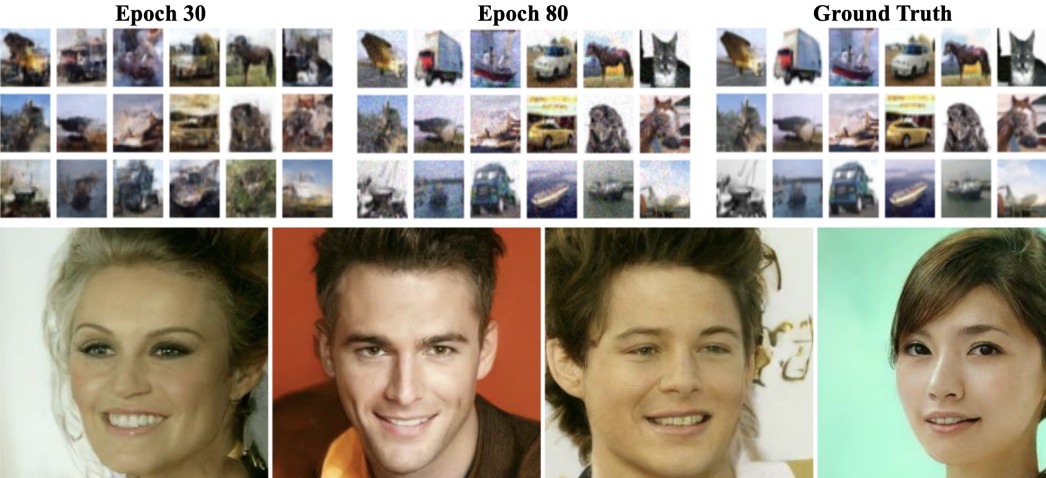

Figure 4: Top panel: Comparisons between $\mathcal{U}$-SDENO results and CIFAR10 ground truth (pretrained model outputs) at epochs 30 and 80. Bottom panel: Unconditional sampling on CelebA-HQ.

**Navier-Stokes Equation**    In the second experiment, we evaluate $\mathcal{F}$-SPDENO on the 2D incompressible Navier-Stokes equations in vorticity form (Li et al., 2020). The dynamic takes the form

$$dX_t = \Big(\nu \, \Delta X_t \; - \; U[X_t]{\cdot}\nabla X_t \; + \; f\Big)\, dt \; + \; \sigma \, dW_t, \;\; (t,x) \in [0,T] \times \mathbb{T}^2, \;\; X_0 = \chi_0 \in L_0^2(\mathbb{T}^2).$$

Here we take $U[X_t] \coloneqq \nabla^\perp(-\Delta)^{-1}X_t$ with $\nabla^\perp \coloneqq (-\partial_{x_2}, \partial_{x_1})$ where $\partial$ denotes the partial derivative of the spatial variable (i.e., $x = (x_1, x_2)$). The parameter $\nu > 0$ is the viscosity; $f$ is a deterministic vorticity forcing. Similarly, we generate the samples and ground truth solutions, followed by the work in (Salvi et al., 2022). We train and evaluate our model on 64×64 grid.

We present learning outcomes in Table 2 and Figure 3, following the qualitative protocol used in NSPDE (Salvi et al., 2022). $\mathcal{F}$-SPDENO attains the lowest error among all competitors in Table 2. Compared to the strongest baseline NSPDE, the error is reduced from 0.040 to 0.037 on $W \mapsto X$ (about 7.5% relative improvement), and more substantially from 0.047 to 0.031 on $(X_0, W) \mapsto X$ (about 34% relative improvement). This indicates that our model better captures the joint effect of the random forcing and varying initial conditions. We also observe that deterministic operator learners such as FNO already perform competitively on $W \mapsto X$, but degrade notably when the initial condition varies (from 0.051 to 0.073). By contrast, $\mathcal{F}$-SPDENO consistently improves over FNO in both settings (from 0.051 to 0.037 and from 0.073 to 0.031), showing that enriching FNO with WCE-based noise features is crucial for handling stochastic SPDEs. In contrast to iterative inference methods like NSPDE, our framework first uses WCE to convert the stochastic forcing into static features. This enables a single FNO to learn the global solution operator in a one-shot forward pass, while still achieving superior accuracy.

### 5.2    APPROXIMATION ON SDE SOLUTIONS

**Diffusion One-step Sampler**    We now turn our attention to applying our model to SDEs. As we have mentioned, one of the key advantages of the NO method is that it enables one-shot evaluation.

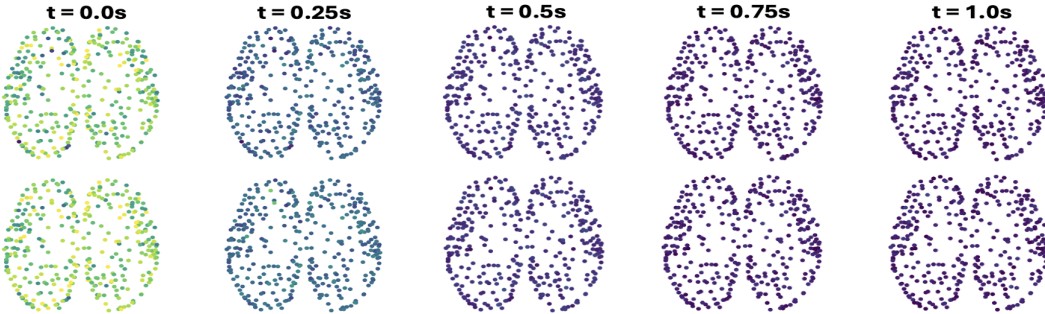

Figure 5: Interpolation from TSBM (Top line, WD=$9.51 \pm 0.12$) and $\mathcal{G}$-SDENO (Bottom line, WD=$9.60 \pm 0.09$) on brain signals, both models are trained in 5 runs (Yang, 2025).

This naturally makes SDENO a one-step sampler for diffusion generative models (Ho et al., 2020; Song et al., 2020). In this experiment, we apply our $\mathcal{U}$-SDENO on pretrained DDPM models on CIFAR10 and CelebA-HQ [2]. Specifically, the canonical DDPM reverse SDE takes the form

$$dX_t = \left[ -\tfrac{1}{2}\,\beta(t)\,X_t \;-\; \beta(t)\,\nabla_x \log p_t(X_t) \right] dt \;+\; \sqrt{\beta(t)}\,d\overline{W}_t. \tag{8}$$

At each reverse step $t$, we sample $\widehat{X}_t$ from pretrained models, and compute the DDPM posterior mean $\mu_t$ and variance $\sigma_t^2$ from the fixed noise schedule $\{\beta_s\}_{s=1}^T$, and then draws $X_{t-1} = \mu_t + \sigma_t z_t$ with $z_t \sim \mathcal{N}(0, I)$ (no noise at $t$=0). Operationally, $z_t$ realizes the reverse-time increment $d\overline{W}_t \approx \sqrt{\Delta t}\, z_t$. We sample $d\overline{W}$ trajectories followed by the previous work in (Zheng et al., 2023). Figure 4 shows the generated images of $\mathcal{U}$-SDENO at the last prediction step compared with the pretrained DDPM outputs on CIFAR-10, and unconditional samples on CelebA-HQ. One can see that $\mathcal{U}$-SDENO reconstructs the images with high fidelity and produces unconditional samples of comparable quality. Unlike standard diffusion models, which learn the score fields and integrate the reverse SDE step by step, SDENO learns the entire solution operator that maps a full noise trajectory $d\overline{W}_{1:T}$ directly to the final sample in a single forward pass. At sampling time, we only need to draw a Gaussian noise trajectory and evaluate SDENO once, without repeatedly invoking the large UNet denoiser. The goal of this experiment is therefore not to compete with specialized one-step diffusion schemes, e.g., consistency models (Song et al., 2023), but to demonstrate that the same WCE-based NO framework can also handle high-dimensional SDEs in the image domain.

**Topological Interpolation** In this experiment, we show SDENO also possesses strong interpolation power (i.e., accuracy over intermediate steps) on the graph domain. To achieve this goal, we pretrain a topological Schrodinger bridge matching (TSBM) model (Yang, 2025) on cortical fMRI data and train SDENO match high-energy brain signals to their aligned low-energy states (Van Essen et al., 2013). The brain graph is constructed, followed by (Glasser et al., 2016). The training of TSBM is conducted by using an alternating scheme and both forward and backward processes governed by the graph stochastic heat diffusion equation $dX_t = -cLX_t dt + g_t dW_t$, where $L$ is the normalized graph Laplacian. Fixing $X_T \sim p_T$ (prior) and $X_0 \sim p_0$ (posterior), TSBM seeks to fit two (graph) neural networks $\phi_\sharp$ and $\phi_\square$ for forward and backward SDEs for brain signals,

$$dX_t = (-cLX_t + g_t^2 \nabla_x \log \phi_\sharp(t)(X_t))dt + g_t dW_t. \tag{9}$$

Here, $\nabla_x$ is the gradient of the feature vectors and the backward SDE can be obtained by replacing $\phi_\sharp$ with $\phi_\square$. TSBM is then evaluated by the 1-Wasserstein distance metric (the lower the better). We only sample the noise of the trained TSBM forward process, and the propagators are computed by inputting the positional encoding of time steps into an MLP. The results are presented in Figure 5, where both TSBM and $\mathcal{G}$-SDENO show similar interpolation performance on all intermediate times. Different from $\mathcal{U}$-SDENO, in this experiment, the propagator is computed outside the GNN model, and the final node features is obtained by the inner product between GNN and MLP outputs.

**Extrapolation** We further explore the model's performance under extrapolation settings. We consider two widely used SDEs, namely the Ornstein-Uhlenbeck (OU) process

$$dX_t = -\theta(X_t - \mu)dt + \sigma dW_t, \tag{10}$$

---

[2]Model id :`google/ddpm-cifar10-32` and `google/ddpm-celebahq-256`.

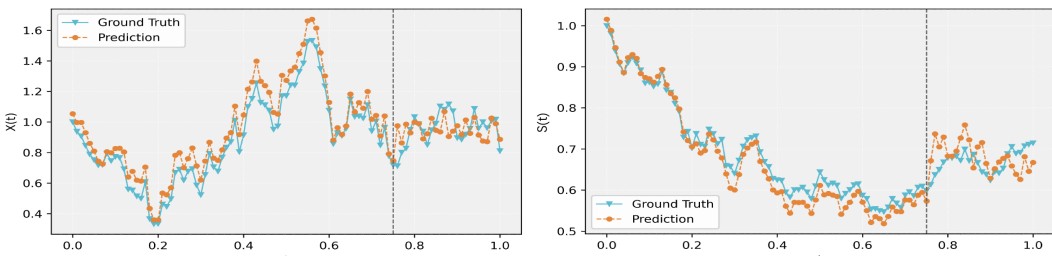

Figure 6: Extrapolation performance on the OU process (left) and Heston model (right).

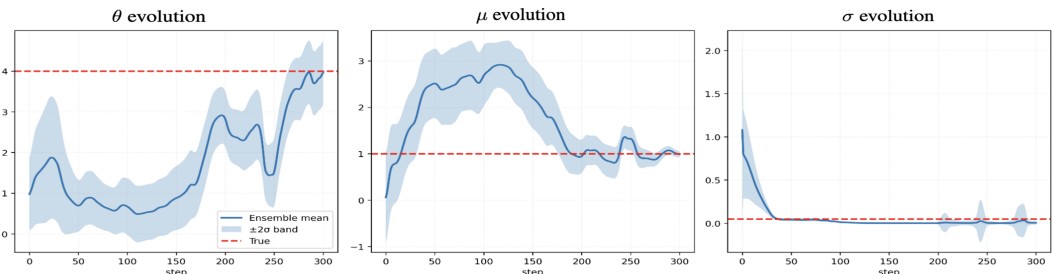

Figure 7: Parameter estimation of the OU process.

and financial Heston model (De Spiegeleer et al., 2018)

$$dS_t = \mu S_t dt + \sqrt{V_t} S_t dW_t^S, \quad dV_t = \kappa(\theta - V_t)dt + \zeta \sqrt{V_t} dW_t^V, \quad d\langle W^S, W^V \rangle_t = \rho dt, \quad (11)$$

where $S_t > 0$ is the asset price and $V_t \geq 0$ is the instantaneous variance with $\sqrt{V_t}$ known as volatility, and $\rho \in [-1, 1]$ represents the correlation between Brownian motions $W^S$ and $W^V$. The propagator for OU process is simply learned by one MLP whereas for Heston, after we simulate $W_t^S$, we obtain $W_t^V$ by letting $W_t^V = \rho dW_t^S + \sqrt{1 - \rho^2} dW_t^2$, where $dW_t^2$ is another Wiener increment that is independent of $dW_t^S$. Then we concatenate their individual ($\xi^S$, $\xi^V$) and crossed Wick features ($\xi^S \xi^V$) and feed it to one additional MLP whose output is merged with estimated propagators by inner product. To achieve the extrapolation setting, we first sample 500 paths for each process with 100 steps with $t \in [0, 1]$, and we feed our model the first 75 steps for training, whereas 100 steps for evaluation. The results are in Figure 6, one can see that our model not only fits the training data well but also demonstrates strong extrapolation power into the unseen future. This suggests a promising potential for our framework in forecasting applications.

### 5.3 *Meta*-PATH, *Meta*-SDENO AND PARAMETER ESTIMATION

In this experiment, we consider a parametric family of SDEs with parameters $\gamma \in \Gamma$. We again consider the OU process in Eq. (10). We define *Meta-paths* as a family of trajectories generated by the OU process with different $\gamma$ and *Meta-SDENO* denotes an operator model trained on these *Meta-paths*. Our hypothesis is if a *Meta-SDENO* shows good fit on *Meta-paths*, then it serves as a faithful forward model for parameter estimation under the ensemble Kalman filter (EnKF).

**Stepwise Supervision** We let $\gamma = [\theta, \mu, \sigma] \in \mathbb{R}^3$, and we set the ground truths to $\theta^\star = 4$, $\mu^\star = 1$ and $\sigma^\star = 0.05$, respectively. With a fixed OU process structure, we initially sample $P = 300$ paths with $T = 300$, $dt = 0.01$, and initial condition $X_0 = 0$. We train a *Meta-SDENO* model for 100 epochs using the same model architecture for the extrapolation experiment. More importantly, in this task, *Meta-SDENO* is trained using a *stepwise* supervision scheme. Specifically, within each epoch, we uniformly sample (then fix) $\ell = 250$ combinations (priors) from predefined distributions, e.g., $\theta \sim U(0, 5)$, $\mu \sim U(-3, 3)$ and $\sigma \sim U(0, 0.5)$. For each triplet, we reuse the same $dW$ channels obtained from the initial simulation process, producing 300 new paths (with shared driver) for this specific triplet. For time $t \in [0, \ldots, T-1]$, we feed the model with the corresponding $dW$ records up to $t$ to let the model predict $\widehat{X}_{t+1} \in \mathbb{R}^P$. Finally, the total MSE loss of the model is $\mathcal{L}_{\text{MSE}} = \frac{1}{\ell \times P \times T} \sum_{l=1}^{\ell} \sum_{p=1}^{P} \sum_{t=1}^{T} \|\widehat{X}_{t+1}(p, l) - X_{t+1}(p, l)\|_2^2$.

**EnKF and Parameter Estimation Process**    After pretraining, we freeze *Meta-SDENO* and treat $\gamma = [\theta, \mu, \sigma] \in \mathbb{R}^3$ as the latent state to be inferred with a standard EnKF (Evensen, 2003; Aanonsen et al., 2009). The EnKF iteratively adjusts the parameter ensemble so that model predictions match the observed *Meta-paths*, using the sample covariance between parameters and predicted observations to form a gain. This procedure reduces ensemble spread while steering the mean toward parameters that better explain the data. Full dynamics for EnKF is provided in Appendix F.6. We report the parameter estimate results as the average over the last five steps. As shown in Figure 7, after 300 updates *Meta-SDENO*, coupled with EnKF, successfully recovers the ground truth parameters, suggesting a harmonic collaboration between *Meta-SDENO* and EnKF. We highlight that these findings could include further mean-field analyses of the EnKF within stochastic domains (Law et al., 2016).

## 5.4 MANIFOLD SDEs

Finally, we test our model on manifold SDEs. We pretrain the Riemannian Score-Based Generative Model (RSGM) (De Bortoli et al., 2022) on the Flood dataset (Brakenridge, 2016). Let $X \in \mathbb{S}^2$, we consider the forward SDE as manifold Langevin dynamics, $dX_t = -\frac{1}{2} \nabla U(X_t)\, dt + dW_t^{\mathcal{M}}$, where $\nabla$ is the Riemannian gradient. Setting $U(x) \equiv 0$ yields intrinsic Brownian motion on the compact manifold (i.e., $dX_t = dW_t^{\mathcal{M}}$), and the time-reversed diffusion becomes $dX_t = \nabla \log p_{T-t}(X_t)\, dt + dW_t^{\mathcal{M}}$. After pretraining, we sample $dW^{\mathcal{M}}$ trajectories on the tangent bundle using the standard geodesic random walk (GRW) sampler (De Bortoli et al., 2022). We then parallel-transport the recorded $dW^{\mathcal{M}}$ trajectory to a fixed reference plane to obtain Wick features. In this experiment, we build SDENO with two separate MLPs: one provides additional encoding on Wick features and the other approximates the ODE solutions. We apply a retraction (via the exponential map) to project both predictions and ground truth back onto $\mathbb{S}^2$. Finally, the model is trained via the MSE loss between trajectories on the tangent space. We note that RSGM is pre-trained with its original likelihood-based (score-matching) objective as in De Bortoli et al. (2022), whereas our manifold SDENO is trained by minimizing an MSE loss on the tangent space. For evaluation, however, we place both models on equal footing by measuring the same NLL between their generated trajectories and the reference flood data, yielding NLL $= 0.49 \pm 0.20$ for RSGM and NLL $= 0.51 \pm 0.12$ for our manifold SDENO. Figure 8 shows a representative sample of SDENO generations on $\mathbb{S}^2$, and Table 3 summarises the NLL comparison with baseline methods. In Appendix F.7 (Figure 15), we visualize the prediction differences between SDENO and RSGM.

Table 3: NLL on the Flood dataset over 5 runs (lower is better).

| Method | NLL |
|---|---|
| Mixture of Kent (Peel et al., 2001) | 0.73±0.07 |
| RCNF (Mathieu and Nickel, 2020) | 1.11±0.19 |
| Moser Flow (Rozen et al., 2021) | 0.57±0.10 |
| RSGM (De Bortoli et al., 2022) | 0.49±0.20 |
| Manifold-SDENO (ours) | 0.51±0.12 |

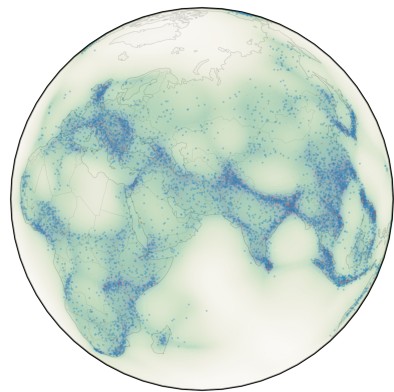

Figure 8: Flood events prediction on $\mathbb{S}^2$.

## 6 CONCLUSION

In this work, we presented a family of NO models for solving SDEs and SPDEs that are built on classical Wiener–chaos expansions of their solutions. Our approach projects the driving noise onto orthonormal Wick–Hermite features and learns deterministic propagator functions parameterised by neural operators, so that full trajectories can be reconstructed from noise in a single forward pass. Building on existing WCE results, we made the associated propagator systems for multi-dimensional SDEs and semilinear SPDEs explicit and used them to motivate our architectures, and we demonstrated competitive accuracy and broad applicability across classical SPDE benchmarks, diffusion one-step sampling, graph interpolation, coupled Heston extrapolation, parameter estimation, and manifold SDEs, while retaining the one-shot evaluation advantage of neural operators.

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

**LLM Acknowledgment** The authors acknowledge the use of LLMs in this working for grammar corrections.

### General Structure of Appendix

The general structure of the appendix can be divided into **Theoretical part** and **Empirical part**.

- **Theoretical part**: From Appendix B to E, we conduct a thorough analysis on both WCE theory and our proposed models.
- **Empirical Part**: In appendix F, we show details of our experiments, including data generation, model architecture, training and evaluation, as well as some additional results and analysis.

## Appendices Contents

**Reproducibility Statement.** We aim to make all results in this paper reproducible. The SPDE/SDE formulations, noise constructions, and Wiener–chaos based models are fully specified in Sections 2–3, and all theoretical assumptions and proofs are provided in Appendices. For each experiment (dynamic $\Phi_1^4$, stochastic Navier–Stokes, diffusion one-step samplers, topological interpolation/extrapolation, and manifold SDEs), we describe the datasets, preprocessing steps, model architectures, training hyperparameters, and evaluation metrics in Appendix. All reported quantitative results are averaged over multiple runs with different random seeds, and we report mean $\pm$ standard deviation where appropriate. The datasets we use are either standard benchmarks or publicly available from the cited works, and we follow their original splits and preprocessing pipelines. Upon acceptance, we will release the full implementation and scripts to reproduce all tables and figures, together with configuration files for each experiment.

## A    RELATED WORKS

**Wiener Chaos Expansion** Wiener chaos expansion (WCE), also known as polynomial chaos expansion, is a classical technique in stochastic analysis that represents random processes via an orthogonal polynomial basis. Originating from Norbert Wiener's homogeneous chaos in 1938 and the Cameron–Martin expansion in 1947 (Wiener, 1938; Cameron and Martin, 1947). In the initial stage, WCE was first applied to the field of stochastic finite element analysis (Ghanem and Spanos, 2003) and extended to the theory of probability distributions, known as Wiener–Askey polynomial chaos (Xiu and Karniadakis, 2002). WCE allows one to expand the solution of SDE or SPDE in terms of polynomial functionals with deterministic coefficients, known as propagators. Spectral Galerkin methods based on WCE have been successfully applied to propagate uncertainty in complex systems, e.g., efficiently solving stochastic Burgers and Navier–Stokes equations driven by random forcing (Hou et al., 2006). To handle strong nonlinearity or long-time integration, advanced variants, such as multi-element polynomial chaos, decompose the random input domain and build local expansions in each element, thereby improving convergence for problems with discontinuities or large perturbations (Wan and Karniadakis, 2006). In recent years, there is a growing interest in combining WCE with machine learning techniques. For instance, neural chaos methods replace the fixed orthogonal polynomials with trainable neural network basis functions to learn an optimal expansion from data (Sharma et al., 2025). Likewise, WCE has been employed as a data-driven operator-learning approach in scientific machine learning, achieving competitive accuracy in learning solution maps of PDEs while providing built-in uncertainty quantification (Sharma et al., 2024). These developments demonstrate how WCE has evolved from a purely theoretical construct into a versatile computational tool for both numerical uncertainty quantification and modern ML-based modeling.

**SPDE and SDE Solvers** Classical approaches for solving SDEs and SPDEs build on established numerical discretization methods such as finite difference schemes, finite element methods, and spectral Galerkin approximations (Platen and Bruti-Liberati, 2010; Orszag, 1971). Monte Carlo simulation is another cornerstone, often employed to handle stochastic inputs or to estimate statistical properties of the solution (Jentzen and Kloeden, 2009). These techniques have been extensively validated on canonical models: for example, finite difference methods reliably capture solutions and sensitivities in financial SDE benchmarks like the Heston stochastic volatility model (Milstein et al., 2005), while spectral Galerkin (e.g. polynomial chaos) methods and fine-grid CFD solvers have been applied to SPDEs such as turbulent Navier–Stokes equations with random forcing (Abraham et al.,

2018). These learning approaches show promise (often alleviating the curse of dimensionality or enabling inverse analyses), but the well-established numerical schemes above remain the standard reference for accuracy and robustness in solving stochastic (partial) differential equations.

**Neural Operators, and NO for SPDE and SDE** generalize neural networks to learn mappings between infinite-dimensional function spaces, enabling discretization-invariant (resolution-generalizable) solution of parametric PDEs (Kovachki et al., 2023). Foundational frameworks like DeepONet and the Fourier Neural Operator (FNO) demonstrated this paradigm by accurately learning nonlinear solution operators for diverse PDE families (Lu et al., 2019; Li et al., 2020). These models learn an entire family of equations rather than a single instance, with FNO achieving zero-shot super-resolution across mesh resolutions. Building on these advances, recent works have extended neural operators to stochastic domains. For SPDEs, specialized architectures incorporate the effects of random forcing into the operator learning process: for example, a Neural SPDE model can parameterize solution operators depending simultaneously on the initial condition and a realization of the noise (Salvi et al., 2022), and the Neural Operator with Regularity Structure (NORS) leverages Hairer's regularity structure theory to handle the low regularity of SPDE solutions (Gong et al., 2023; Hu et al., 2022). In the SDE setting, SDE-ONet augments the DeepONet architecture with polynomial chaos expansions to efficiently learn solution operators for stochastic differential equations (Eigel and Miranda, 2024). These developments illustrate the growing capability of neural operators to serve as efficient surrogates for both deterministic and stochastic systems, while preserving their core advantages of infinite-dimensional approximation and mesh-independent generalization. In recent years, machine learning-based solvers have also emerged – e.g., physics-informed neural networks and GAN-driven models that learn solution mappings – to tackle high-dimensional or data-driven SDE/SPDE problems (Li and Feng, 2022).

## B    FORMULATION DETAILS AND PROOFS FOR SECTION 2

In this section, we provide a detailed formulation of the concepts in Section 2 and some conclusions regarding the uniqueness of their solutions. Several results are standard and included only for completeness; for these, we give brief proof sketches and refer the readers to introductions of SDEs in Øksendal and Øksendal (2003) and SPDEs in Holden et al. (1996); Liu and Röckner (2015).

### B.1    A Thorough Formulation of SPDE and SDE

To support the Appendix, we present full SDE and SPDE formulations below, supplementing the SDE (SDE) and SPDE (SPDE). Similar, to align with previous works (Neufeld and Schmocker, 2024; Huschto and Sager, 2014; Eigel and Miranda, 2024), we adopt the following notations for self-completeness.

Given $T > 0$ and filtered probability space $(\Omega, \mathcal{F}, \mathbb{F}, \mathbb{P})$, we consider the semi-linear SPDE, in the following form:

$$dX_t = (AX_t + F(t, \cdot, X_t))dt + B(t, \cdot, X_t)dW_t, \quad X_0 = \chi_0 \in \mathcal{H}. \tag{12}$$

Throughout this work, the symbol $d$ in stochastic integrals such as $\int_0^t (\cdot)\, dW_s$ denotes the Itô differential, though our definition and conclusions can be smoothly transferred to Stratonovich form. The solution of (SPDE) is known as a stochastic process $X : [0, T] \times \Omega \to \mathcal{H}$ where $\mathcal{H}$ is a separable Hilbert Space $(\mathcal{H}, \langle \cdot, \cdot \rangle_{\mathcal{H}})$ with initial value $\chi_0 \in \mathcal{H}$ usually deterministic. We also denote $Q \in L_1(\widetilde{\mathcal{H}}; \widetilde{\mathcal{H}})$ where $L_1(\widetilde{\mathcal{H}}; \widetilde{\mathcal{H}})$ is a vector subspace of non-negative self-adjoint nuclear operators, i.e., $L_1(\widetilde{\mathcal{H}}; \widetilde{\mathcal{H}}) \subseteq L(\widetilde{\mathcal{H}}; \widetilde{\mathcal{H}})$ the bounded vector space of linear operators in $\widetilde{\mathcal{H}}$(Neufeld and Schmocker, 2024). A process $W \coloneqq (W_t)_{t \in [0,T]} : [0, T] \times \Omega \to \widetilde{\mathcal{H}}$ is called a Q-Brownian if $W_0 = 0 \in \mathcal{H}$ and $W$ is with continuous sample path, i.e., for $t \in [0, T]$, the function $t \to W_t(\omega)$ is continuous for almost all $\omega \in \Omega$ and for any two time points $t_1, t_2 \in [0, T]$ and $t_2 > t_1$, $W_{t_2} - W_{t_1}$ is a centered Gaussian random variable with covariance operator $(t_2 - t_1)Q \in L_1(\widetilde{\mathcal{H}}; \widetilde{\mathcal{H}})$. We further let the linear operator $A : \mathrm{dom}(A) \subseteq \mathcal{H} \to \mathcal{H}$, operator $F : [0, T] \times \Omega \times \mathcal{H} \to \mathcal{H}$, and $B : [0, T] \times \Omega \times \mathcal{H} \to L_2(\widetilde{\mathcal{H}}_0; \mathcal{H})$ where $\widetilde{\mathcal{H}}_0 \coloneqq Q^{1/2}\widetilde{\mathcal{H}} \subset \widetilde{\mathcal{H}}$ is the reproducing kernel Hilbert space associated with Q-Brownian motion $W$, and $L_2(\widetilde{\mathcal{H}}_0; \mathcal{H})$ is the space of Hilbert–Schmidt operators.

We now recall the mild solution of SPDE. One can call an $\mathbb{F}$-predictable (Da Prato and Zabczyk, 2014) process $X : [0, T] \times \Omega \to \mathcal{H}$ a *mild solution* of (SPDE) if $\mathbb{P}[\int_0^T \|X_t\|_{\mathcal{H}}^2 dt < \infty] = 1$ and for every $t \in [0, T]$ it holds that

$$X_t = S_t \chi_0 + \int_0^t S_{t-s} F(s, \cdot, X_s) ds + \int_0^t S_{t-s} B(s, \cdot, X_s) dW_s. \quad \mathbb{P}, a.s. \quad (13)$$

where $S = e^{tA} : \mathcal{H} \to \mathcal{H}$ refers as a $C_0$(strongly continuous)-semigroup generated by the linear operator $A$. In terms of the formulation of SDEs. One can set $A = 0$, and both $\mathcal{H}$ and $\widetilde{\mathcal{H}}$ are finite-dimensional vector space, e.g., $\mathbb{R}^d$ and $\mathbb{R}^m$, respectively, the above equation reduces to

$$dX_t = F(t, X_t)dt + B(t, X_t)dW_t, \quad X_0 = x_0 \in \mathbb{R}^d, \quad (14)$$

which is known as Itô's SDE, with $W_t$ as a $m$-dimensional Brownian motion with covariance $Q \in \mathbb{R}^{m \times m}$, i.e., $\mathbb{E}[(W_{t_2} - W_{t_1})(W_{t_2} - W_{t_1})^\top] = (t_2 - t_1)Q$, and $F(t, \cdot) : \mathbb{R}^d \to \mathbb{R}^d$, $B(t, \cdot) : \mathbb{R}^d \to \mathbb{R}^{d \times m}$ are known as the drift and diffusion term of the SDE, respectively. Without loss of generality, in this work, we only consider the standard Brownian motion, i.e., $Q = I$, and our conclusions can be smoothly extended to the general case. Then the strong solution of (SDE) is

$$X_t = x_0 + \int_0^t F(s, X_s)ds + \int_0^t B(s, X_s)dW_s. \quad (15)$$

## B.2 EXISTENCE AND UNIQUENESS OF SPDE, SDE SOLUTIONS

Ensuring SDEs and SPDEs considered in the body admit unique solutions is necessary to enable NO-approximation of their solutions. We begin with the SPDE case, assuming the following.

**Assumption 1.** *We assume the following conditions hold:*

1. *(Semigroup generation) The operator $A \colon \operatorname{dom}(A) \subseteq \mathcal{H} \to \mathcal{H}$ is the infinitesimal generator of a $C_0$-semigroup $(S_t)_{t \in [0,T]}$ on $\mathcal{H}$.*

2. *(Drift measurability) The map $F \colon [0, T] \times \Omega \times \mathcal{H} \to \mathcal{H}$ is $(\mathcal{P}_T \otimes \mathcal{B}(\mathcal{H}))/\mathcal{B}(\mathcal{H})$-measurable, where $\mathcal{P}_T$ is the predictable $\sigma$-algebra on $[0, T] \times \Omega$.*

3. *(Diffusion measurability) The map $B \colon [0, T] \times \Omega \times \mathcal{H} \to L_2(\widetilde{\mathcal{H}}_0; \mathcal{H})$ is $(\mathcal{P}_T \otimes \mathcal{B}(\mathcal{H}))/\mathcal{B}(L_2(\widetilde{\mathcal{H}}_0; \mathcal{H}))$-measurable.*

4. *(Lipschitz continuity and linear growth) There exists a constant $C_{F,B} > 0$ such that for all $t \in [0, T]$, $\omega \in \Omega$, and $x, y \in \mathcal{H}$, the following inequalities hold:*
   $$\|F(t, \omega, x) - F(t, \omega, y)\|_{\mathcal{H}}^2 + \|B(t, \omega, x) - B(t, \omega, y)\|_{L_2(\widetilde{\mathcal{H}}_0; \mathcal{H})}^2 \le C_{F,B}^2 \|x - y\|_{\mathcal{H}}^2,$$
   $$\|F(t, \omega, x)\|_{\mathcal{H}}^2 + \|B(t, \omega, x)\|_{L_2(\widetilde{\mathcal{H}}_0; \mathcal{H})}^2 \le C_{F,B}^2 (1 + \|x\|_{\mathcal{H}}^2).$$

5. *The initial condition $\chi_0 \in \mathcal{H}$ is deterministic.*

With the assumptions above, we now show the classic results on the existence and uniqueness of the SPDE mild solution.

**Proposition 1** (SPDE Solution Existence and Uniqueness (Neufeld and Schmocker, 2024; Da Prato and Zabczyk, 2014)). *Let $p \in [1, \infty)$ and assume that Assumption 1 holds. Then, the (SPDE) has a unique $\mathbb{F}$-predictable mild solution $X \colon [0, T] \times \Omega \to \mathcal{H}$. Furthermore, there exists a constant $C_{p,T}$, depending only on $p$, $C_{F,B}$, $T$, and $C_S := \sup_{t \in [0,T]} \|S_t\|_{L(\mathcal{H}; \mathcal{H})}$, such that the following moment bound holds:*

$$\mathbb{E}\Big[\sup_{t \in [0,T]} \|X_t\|_{\mathcal{H}}^p\Big] \le C_{p,T}(1 + \|\chi_0\|_{\mathcal{H}}^p) < \infty. \quad (16)$$

*Moreover, the process $X$ admits a continuous modification.*

*Proof.* The proof of this proposition is a cornerstone result in the theory of stochastic evolution equations and can be found in detail in standard literature, for instance, in Da Prato and Zabczyk (2014). The complete argument relies on applying the Banach fixed-point theorem to an appropriate operator on a space of stochastic processes, combined with maximal inequalities for stochastic convolutions. Thus we omit the proof here. $\square$

Having shown the uniqueness guarantee for SPDEs, we now show a similar conclusion for SDEs.

**Assumption 2.** *The drift coefficient $F\colon [0,T] \times \mathbb{R}^d \to \mathbb{R}^d$ and the diffusion coefficient $B\colon [0,T] \times \mathbb{R}^d \to \mathbb{R}^{d \times m}$ are measurable functions for which constants $L > 0$ and $K > 0$ exist, such that the following conditions hold:*

1. *(Global Lipschitz continuity) For all $t \in [0,T]$ and all $x, y \in \mathbb{R}^d$:*
$$\|F(t,x) - F(t,y)\| + \|B(t,x) - B(t,y)\|_{Fro} \le L\|x - y\|.$$

2. *(Linear growth) For all $t \in [0,T]$ and all $x \in \mathbb{R}^d$:*
$$\|F(t,x)\|^2 + \|B(t,x)\|_{Fro}^2 \le K^2(1 + \|x\|^2).$$

*Here, $\|\cdot\|$ denotes the Euclidean norm for vectors and $\|\cdot\|_{Fro}$ denotes the Frobenius norm for matrices. The initial condition $x_0 \in \mathbb{R}^d$ is a deterministic vector.*

**Proposition 2** (SDE Solution Existence and Uniqueness ). *Let Assumption 2 hold. Then, the (SDE) admits a unique, $\mathbb{F}$-adapted, and continuous strong solution $X\colon [0,T] \times \Omega \to \mathbb{R}^d$. Furthermore, for any $p \in [1,\infty)$, there exists a constant $C_{p,T}$, depending only on $p$, $T$, and the constants $L$ and $K$ from Assumption 2, such that the solution's moments are bounded:*

$$\mathbb{E}\Big[\sup_{t \in [0,T]} \|X_t\|^p\Big] \le C_{p,T}\big(1 + \|x_0\|^p\big) < \infty. \tag{17}$$

*Proof.* This is a classical result in the theory of stochastic differential equations, with a detailed proof available Øksendal and Øksendal (2003). The proof for existence and uniqueness is typically established by constructing a sequence of approximations via Picard iteration and showing that this sequence converges to a limit that is the unique solution. This is achieved by proving that the corresponding integral operator is a contraction mapping on a suitable Banach space of stochastic processes. □

## C  PROOFS FOR CONCLUSIONS IN SECTION 3

### C.1  PROOFS OF THE CONCLUSIONS IN SECTION 3.1

**Proof of Lemma 1 (sketch).**  This is a classical result (see, e.g., Da Prato and Zabczyk (2014, Proposition 4.3) or Neufeld and Schmocker (2024, Section 2)); we briefly recall the main idea. For a fixed component $W^{(i)}$, the indicator $\mathbf{1}_{[0,t]}$ admits the $L^2([0,T])$ expansion $\mathbf{1}_{[0,t]}(s) = \sum_{j \ge 1} G_j(t)\, e_j(s)$, which yields $W_t^{(i)} = \int_0^T \mathbf{1}_{[0,t]}(s)\, dW_s^{(i)} = \sum_{j \ge 1} G_j(t)\, \xi_{ij}$ by linearity and the Itô isometry. The truncated processes $\widehat{W}_t^{(n,i)} = \sum_{j=1}^n G_j(t)\, \xi_{ij}$ converge in $L^2(\Omega)$ for each fixed $t$, and applying a maximal inequality (e.g., Doob or BDG) together with the summability assumption $\sum_j \|G_j\|_{C([0,T])}^2 < \infty$ gives convergence in $L^2(\Omega; C([0,T]))$. Almost sure uniform convergence then follows by a standard Borel–Cantelli argument; see the above references for details.

**Proof of Lemma 2 (sketch).**  This is the natural extension of Lemma 1 to a $Q$-Brownian motion on a Hilbert space and follows standard arguments based on the Karhunen–Loève expansion (see, e.g., (Da Prato and Zabczyk, 2014, Section 3.3) and (Lototsky et al., 2017, Section 2)). Writing

$$W_t = \sum_{k \ge 1} \sqrt{\lambda_k}\, \beta_t^k f_k,$$

with eigenpairs $(\lambda_k, f_k)$ of the trace-class covariance operator $Q$ and independent scalar Brownian motions $\beta^k$, we first truncate the spatial expansion and define $W_t^{(K)} = \sum_{k=1}^K \sqrt{\lambda_k}\, \beta_t^k f_k$. Using Doob-type maximal inequalities for martingales and the summability of $(\lambda_k)$, one shows that

$$\mathbb{E}\Big[\sup_{t \in [0,T]} \|W_t - W_t^{(K)}\|_{\mathcal{H}}^2\Big] \longrightarrow 0 \quad \text{as } K \to \infty,$$

so the spatial truncation error vanishes in $L^2(\Omega; C([0,T]; \mathcal{H}))$. For each fixed $k$, Lemma 1 yields a temporal expansion $\beta_t^k = \sum_{j \ge 1} \xi_{kj} G_j(t)$ with truncated approximations $\widehat{\beta}_t^{(n,k)}$ that converge

uniformly in $t$ in $L^2(\Omega)$ as $n \to \infty$. Since only finitely many modes $k \le K$ are retained in $W^{(K)}$, the temporal truncation error

$$W_t^{(K)} - \widehat{W}_t^{(K,n)} = \sum_{k=1}^{K} \sqrt{\lambda_k} \left( \beta_t^k - \widehat{\beta}_t^{(n,k)} \right) f_k$$

can be controlled by a finite sum of the one-dimensional errors and therefore also vanishes in $L^2(\Omega; C([0,T]; \mathcal{H}))$ as $n \to \infty$. Combining the spatial and temporal truncations and using standard arguments (e.g., Borel–Cantelli) yields the claimed almost sure convergence in $C([0,T]; \mathcal{H})$.

### C.2    PROOFS OF THE CONCLUSIONS IN SECTION 3.2

In this section, we briefly recall standard properties of Wick polynomials that are used in Section 3.2; full details can be found in classical references on Wiener–Itô chaos such as (Lototsky et al., 2017; Luo, 2006; Neufeld and Schmocker, 2024).

**Wick polynomials.**    For convenience we restate Definition 2.

**Definition 3** (Wick polynomial; repeat of Definition 2). *Let $\Xi = \{\xi_{mj}\}_{m,j\in\mathbb{N}}$ be the family of Gaussian random variables constructed in Definition 1. The index $m$ corresponds to the spatial/component mode, while $j$ relates to the temporal basis. Let $\mathcal{J}$ be the set of all multi-indices $\alpha = (\alpha_{mj})_{m,j\in\mathbb{N}}$ with finite support. For any $\alpha \in \mathcal{J}$, the corresponding normalised Wick monomial is*

$$\xi_\alpha(\omega) := \frac{1}{\sqrt{\alpha!}} \prod_{m,j} h_{\alpha_{mj}}(\xi_{mj}(\omega)), \tag{18}$$

*where $h_k$ is the $k$-th (probabilist's) Hermite polynomial, $h_k(x) = (-1)^k e^{x^2/2} \frac{d^k}{dx^k} e^{-x^2/2}$, and $\alpha! := \prod_{m,j} \alpha_{mj}!$. A finite linear combination of such monomials is called a Wick polynomial.*

We recall the basic orthogonality of Hermite polynomials with respect to the Gaussian measure.

**Proposition 3** (Hermite polynomials). *For a standard normal variable $Z \sim \mathcal{N}(0,1)$,*

$$\mathbb{E}[h_n(Z) \, h_k(Z)] = n! \, \delta_{nk}. \tag{19}$$

This is a standard identity; see, for example, Lototsky et al. (2017, Chapter 1) or Luo (2006, Section 2).

**Lemma 3** (Orthonormality of Wick polynomials). *The collection of Wick monomials $\{\xi_\alpha\}_{\alpha\in\mathcal{J}}$ forms an orthonormal system in the Hilbert space $L^2(\Omega, \sigma(\Xi), \mathbb{P})$, i.e.,*

$$\mathbb{E}[\xi_\alpha \xi_\beta] = \delta_{\alpha\beta} \quad \text{for all } \alpha, \beta \in \mathcal{J}.$$

*Sketch.* By Definition 2 and the independence of the Gaussian variables $\{\xi_{mj}\}$, we can write

$$\mathbb{E}[\xi_\alpha \xi_\beta] = \frac{1}{\sqrt{\alpha! \, \beta!}} \prod_{m,j} \mathbb{E}\left[ h_{\alpha_{mj}}(\xi_{mj}) \, h_{\beta_{mj}}(\xi_{mj}) \right].$$

Each factor is computed using Proposition 3, yielding $\mathbb{E}[h_{\alpha_{mj}}(\xi_{mj}) \, h_{\beta_{mj}}(\xi_{mj})] = \alpha_{mj}! \, \delta_{\alpha_{mj},\beta_{mj}}$. If $\alpha \ne \beta$, there exists at least one $(m,j)$ with $\alpha_{mj} \ne \beta_{mj}$, so the product vanishes. If $\alpha = \beta$, then all Kronecker deltas are one and we obtain $\mathbb{E}[\xi_\alpha^2] = 1$ by the definition of $\alpha!$. This proves orthonormality; see, e.g., Lototsky et al. (2017, Section 2) for a textbook derivation. $\square$

For completeness, we also recall that the Wick monomials form a complete basis in the corresponding $L^2$-space.

**Theorem 3** (Wiener–Itô chaos / Cameron–Martin theorem). *Let $\Xi = \{\xi_{mj}\}_{m,j\in\mathbb{N}}$ be as above and let $L^2(\Omega, \sigma(\Xi), \mathbb{P})$ denote the closed subspace of $L^2(\Omega)$ generated by $\Xi$. Then the linear span of $\{\xi_\alpha\}_{\alpha\in\mathcal{J}}$ is dense in $L^2(\Omega, \sigma(\Xi), \mathbb{P})$, and every $M \in L^2(\Omega, \sigma(\Xi), \mathbb{P})$ admits the Wiener–Itô chaos expansion*

$$M = \sum_{\alpha\in\mathcal{J}} \mathbb{E}[M \, \xi_\alpha] \, \xi_\alpha,$$

*with convergence in $L^2(\Omega)$.*

This is the classical Wiener–Itô/Cameron–Martin theorem; see, for example, (Lototsky et al., 2017, Theorem 1.1), (Luo, 2006, Chapter 2), or (Neufeld and Schmocker, 2024, Section 2). We do not reproduce the proof here.

The completeness of the Wick polynomial directly leads to the approximation guarantee of leveraging the Wick polynomial for approximating the solution operator of SDE and SPDE. Now we are ready to prove the main theorem of this work. We start by proving the theorem for SDE.

**Proof of the Theorem on SDE Propagator System**   The conclusion for the 1-dimensional SDE is established in (Eigel and Miranda, 2024)(Theorem 2.2), we slightly extend the Theorem to $d$-dimensional case. We first recall our theorem presented on the main page.

**Theorem 4** (Repeat of SDE Propagator, Theorem 1). *Let $X_t \in \mathbb{R}^d$ be the strong solution to the $d$-dimensional SDE* (SDE) *driven by an $m$-dimensional Brownian motion $W_t$. The solution's WCE is given by*

$$X_t = \sum_{\alpha \in \mathcal{J}} u_\alpha(t)\xi_\alpha, \tag{20}$$

*where the propagators are $u_\alpha(t) := \mathbb{E}[X_t \xi_\alpha] \in \mathbb{R}^d$. The evolution of these propagators is governed by a coupled system of ODEs. For $j \in \{1, \ldots, d\}$, the ODE for $u_\alpha^{(j)}(t)$ is:*

$$\frac{d}{dt}u_\alpha^{(j)}(t) = \mathbb{E}\left[F^{(j)}(t, X_t)\xi_\alpha\right] + \sum_{k=1}^{m}\sum_{i=1}^{\infty}\sqrt{\alpha_{ki}}\, e_i(t)\, \mathbb{E}\left[B^{(j,k)}(t, X_t)\xi_{\alpha - e_{ki}}\right], \tag{21}$$

*with $u_\alpha^{(j)}(0) = x_0^{(j)}\delta_{\alpha 0}$, and $e_{ki}$ denotes the multi-index that is 1 at position $(k, i)$ and 0 elsewhere.*

*Proof.* The proof relies on applying the expectation operator to the integral form of the SDE and leveraging the properties of the Wick-Hermite basis $\{\xi_\alpha\}$. We will derive the ODE for a single component $u_\alpha^{(j)}(t)$ of a propagator vector $u_\alpha(t)$.

The integral form of the $j$-th component of the SDE (SDE) is:

$$X_t^{(j)} = x_0^{(j)} + \int_0^t F^{(j)}(s, X_s)ds + \sum_{k=1}^{m}\int_0^t B^{(j,k)}(s, X_s)dW_s^{(k)}. \tag{22}$$

By the definition of the propagator, $u_\alpha^{(j)}(t) = \mathbb{E}[X_t^{(j)}\xi_\alpha]$. Applying this to the integral equation above, we get:

$$u_\alpha^{(j)}(t) = \mathbb{E}[x_0^{(j)}\xi_\alpha] + \mathbb{E}\left[\left(\int_0^t F^{(j)}(s, X_s)ds\right)\xi_\alpha\right] + \sum_{k=1}^{m}\mathbb{E}\left[\left(\int_0^t B^{(j,k)}(s, X_s)dW_s^{(k)}\right)\xi_\alpha\right]. \tag{23}$$

We now analyze each term on the right-hand side.

*1. The Initial Condition Term* Since $x_0^{(j)}$ is deterministic and $\{\xi_\alpha\}$ is an orthonormal system with $\xi_0 = 1$, we have $\mathbb{E}[\xi_\alpha] = \delta_{\alpha 0}$. Thus,

$$\mathbb{E}[x_0^{(j)}\xi_\alpha] = x_0^{(j)}\mathbb{E}[\xi_\alpha] = x_0^{(j)}\delta_{\alpha 0}. \tag{24}$$

*2. The Drift Term* Using the linearity of expectation and Fubini's theorem to interchange expectation and time integration, this term becomes:

$$\mathbb{E}\left[\left(\int_0^t F^{(j)}(s, X_s)ds\right)\xi_\alpha\right] = \int_0^t \mathbb{E}\left[F^{(j)}(s, X_s)\xi_\alpha\right]ds. \tag{25}$$

*3. The Diffusion Term* This is the most critical part, requiring the use of Malliavin calculus. For each term in the sum over $k \in \{1, \ldots, m\}$, we apply the duality relationship between the Itô integral and the Malliavin derivative (also known as the integration by parts formula for stochastic

integrals). For a sufficiently regular random variable $\Psi$ and a process $\Phi_s$, this is $\mathbb{E}[\Psi \int_0^t \Phi_s dW_s^{(k)}] = \mathbb{E}[\int_0^t D_s^{(k)} \Psi \cdot \Phi_s ds]$, where $D_s^{(k)}$ is the Malliavin derivative with respect to the $k$-th Brownian motion.

Letting $\Psi = \xi_\alpha$ and $\Phi_s = B^{(j,k)}(s, X_s)$, we have:

$$\mathbb{E}\left[\left(\int_0^t B^{(j,k)}(s, X_s) dW_s^{(k)}\right) \xi_\alpha\right] = \mathbb{E}\left[\int_0^t D_s^{(k)} \xi_\alpha \cdot B^{(j,k)}(s, X_s) ds\right]. \tag{26}$$

The Malliavin derivative of the Wick monomial $\xi_\alpha$ with respect to the $k$-th Brownian motion at time $s$ is given by:

$$D_s^{(k)} \xi_\alpha = \sum_{i=1}^\infty \sqrt{\alpha_{ki}} \, e_i(s) \, \xi_{\alpha - e_{ki}}. \tag{27}$$

Substituting this into the expectation and again applying Fubini's theorem:

$$\mathbb{E}\left[\int_0^t D_s^{(k)} \xi_\alpha \cdot B^{(j,k)}(s, X_s) ds\right] = \int_0^t \mathbb{E}\left[\left(\sum_{i=1}^\infty \sqrt{\alpha_{ki}} \, e_i(s) \, \xi_{\alpha - e_{ki}}\right) B^{(j,k)}(s, X_s)\right] ds$$

$$= \int_0^t \sum_{i=1}^\infty \sqrt{\alpha_{ki}} \, e_i(s) \, \mathbb{E}\left[B^{(j,k)}(s, X_s) \xi_{\alpha - e_{ki}}\right] ds. \tag{28}$$

Substituting all processed terms back into Equation (23), we obtain the integral equation for the propagator component:

$$u_\alpha^{(j)}(t) = x_0^{(j)} \delta_{\alpha 0} + \int_0^t \mathbb{E}\left[F^{(j)}(s, X_s) \xi_\alpha\right] ds + \sum_{k=1}^m \int_0^t \sum_{i=1}^\infty \sqrt{\alpha_{ki}} \, e_i(s) \, \mathbb{E}\left[B^{(j,k)}(s, X_s) \xi_{\alpha - e_{ki}}\right] ds. \tag{29}$$

This equation holds for all $t \in [0, T]$. Since the integrands are continuous in $t$, we can differentiate both sides with respect to $t$ using the Fundamental Theorem of Calculus to obtain the desired system of ODEs:

$$\frac{d}{dt} u_\alpha^{(j)}(t) = \mathbb{E}\left[F^{(j)}(t, X_t) \xi_\alpha\right] + \sum_{k=1}^m \sum_{i=1}^\infty \sqrt{\alpha_{ki}} \, e_i(t) \, \mathbb{E}\left[B^{(j,k)}(t, X_t) \xi_{\alpha - e_{ki}}\right],$$

with the initial condition $u_\alpha^{(j)}(0) = x_0^{(j)} \delta_{\alpha 0}$ obtained by setting $t = 0$. This completes the proof. $\square$

**Proof of the Theorem on SPDE Propagator System** After presenting the propagator system for SDEs, we now turn to SPDE propagators. Similar chaos-based formulations of SPDEs can be found in the Wiener–chaos literature; see, for example, (Lototsky and Rozovskii, 2006a;b; Kalpinelli et al., 2011; Neufeld and Schmocker, 2024). Our SPDE propagator system (Theorem 2) specialises these results to the semilinear SPDE (SPDE) and writes the associated deterministic PDEs for the Wiener–chaos coefficients in the compact form (6), which can be used directly for neural-operator parameterisation; see also the discussion in Remark 2.13 of (Neufeld and Schmocker, 2024). Similarly, we recall our theorem in SPDEs.

**Theorem 5** (Repeat of SPDE propagator system, Theorem 2). *The evolution of the propagator fields* $u_\alpha(t, x)$ *for* (SPDE) *is governed by a coupled system of deterministic PDEs*

$$\frac{\partial}{\partial t} u_\alpha(t, x) = (A u_\alpha)(t, x) + \mathscr{F}_\alpha\left(\{u_\gamma(t, x)\}_{\gamma \in \mathcal{J}}\right), \tag{30}$$

*with initial condition* $u_\alpha(0, x) = \chi_0(x) \delta_{\alpha 0}$, *where the nonlinear term* $\mathscr{F}_\alpha$ *is an operator uniquely determined by the nonlinearities $F$ and $B$.*

*Sketch.* The derivation follows the classical Wiener–chaos approach for SPDEs (see, e.g., Lototsky and Rozovskii (2006b;a); Kalpinelli et al. (2011); Neufeld and Schmocker (2024)), adapted to the semilinear SPDE (SPDE). Starting from the mild formulation

$$X_t = S_t \chi_0 + \int_0^t S_{t-s} F(s, X_s) \, ds + \int_0^t S_{t-s} B(s, X_s) \, dW_s,$$

we expand the solution in the Wick basis as

$$X_t(x) = \sum_{\alpha \in \mathcal{J}} u_\alpha(t,x)\,\xi_\alpha.$$

For a fixed multi-index $\beta \in \mathcal{J}$ and space–time point $(t,x)$, we apply the projection $u_\beta(t,x) = \mathbb{E}[X_t(x)\,\xi_\beta]$ to the mild solution and use linearity of the expectation to treat the three terms separately.

The initial term gives

$$\mathbb{E}[S_t\chi_0(x)\,\xi_\beta] = S_t\chi_0(x)\,\mathbb{E}[\xi_\beta] = S_t\chi_0(x)\,\delta_{\beta 0}$$

by orthogonality of the chaos basis. For the drift term, a stochastic Fubini argument yields

$$\mathbb{E}\Big[\Big(\int_0^t S_{t-s}F(s,X_s(x))\,ds\Big)\xi_\beta\Big] = \int_0^t S_{t-s}\,\mathbb{E}[F(s,X_s(x))\,\xi_\beta]\,ds,$$

which defines a deterministic source term depending on the collection $\{u_\gamma\}$ via the chaos expansion of $X_s$.

The diffusion term is treated via the Malliavin-duality (integration-by-parts) formula for Hilbert-space valued Itô integrals. For suitable integrands $B(s,X_s)$ and any test variable $\Psi$ one has

$$\mathbb{E}\Big[\Big(\int_0^t S_{t-s}B(s,X_s)\,dW_s\Big)\Psi\Big] = \mathbb{E}\Big[\int_0^t \langle D_s\Psi, S_{t-s}B(s,X_s)\rangle_{\widetilde{\mathcal{H}}_0}\,ds\Big],$$

where $D_s$ denotes the Malliavin derivative and $\widetilde{\mathcal{H}}_0$ is the Cameron–Martin space associated with $W$. Choosing $\Psi = \xi_\beta$ and using the explicit form of $D_s\xi_\beta$ in terms of the temporal basis $\{e_i\}$ and spatial KL modes $\{f_k\}$ of the $Q$-Brownian motion (analogous to the SDE case), one obtains

$$\mathbb{E}\Big[\Big(\int_0^t S_{t-s}B(s,X_s)\,dW_s\Big)\xi_\beta\Big] = \int_0^t S_{t-s}\,\mathbb{E}\big[\langle D_s\xi_\beta, B(s,X_s(x))\rangle_{\widetilde{\mathcal{H}}_0}\big]\,ds,$$

which defines another deterministic source term depending on $\{u_\gamma\}$.

Collecting the three contributions, we arrive at an integral equation of the form

$$u_\beta(t,x) = S_t\chi_0(x)\,\delta_{\beta 0} + \int_0^t S_{t-s}\,\mathscr{F}_\beta(\{u_\gamma(s,x)\})\,ds,$$

where $\mathscr{F}_\beta$ combines the drift and diffusion contributions and depends deterministically on the propagators $\{u_\gamma\}$. By the standard semigroup theory for abstract Cauchy problems, this Volterra integral equation is the mild form of

$$\partial_t u_\beta(t,x) = (Au_\beta)(t,x) + \mathscr{F}_\beta(\{u_\gamma(t,x)\}), \quad u_\beta(0,x) = \chi_0(x)\,\delta_{\beta 0},$$

which proves the PDE system in Theorem 2. For further details of this chaos-based derivation, we refer to the works cited above. $\qquad\square$

**More Results on the Form of $\mathscr{F}$**  In Theorem 2, we established that the propagators $\{u_\alpha\}$ of the SPDE solution satisfy a deterministic PDE system, where the coupling and forcing are encapsulated in the abstract term $\mathscr{F}_\alpha$, defined via expectation. For a broad and important class of problems, particularly those with polynomial nonlinearities, it is possible to derive the explicit algebraic form of $\mathscr{F}_\alpha$. This procedure replaces the computation of a high-dimensional expectation with an algebraic manipulation of the propagator coefficients. The key mathematical tool for this is the **Wick product**.

**Definition 4** (Wick Product). *Let $Z, W \in L^2(\Omega; \mathcal{H})$ be two stochastic processes that admit the Wiener Chaos Expansions:*

$$Z_t(x) = \sum_{\alpha \in \mathcal{J}} z_\alpha(t,x)\xi_\alpha, \quad \text{and} \quad W_t(x) = \sum_{\beta \in \mathcal{J}} w_\beta(t,x)\xi_\beta,$$

*where $z_\alpha, w_\beta \in C([0,T]; \mathcal{H})$ are the respective propagators. Their* Wick product, *denoted by $Z_t \diamond W_t$, is another process in $L^2(\Omega; \mathcal{H})$ whose chaos expansion is given by:*

$$(Z_t \diamond W_t)(x) := \sum_{\gamma \in \mathcal{J}} \left( \sum_{\alpha+\beta=\gamma} z_\alpha(t,x)w_\beta(t,x) \right) \xi_\gamma. \tag{31}$$

*The inner sum is over all pairs of multi-indices $(\alpha, \beta)$ that sum component-wise to $\gamma$. The $p$-fold Wick power is defined recursively as $Z_t^{\diamond p} := Z_t \diamond \cdots \diamond Z_t$ ($p$ times).*

The Wick product provides a systematic way to handle polynomial nonlinearities. A standard result in chaos expansion theory is that for any polynomial function $P$, the chaos expansion of $P(X_t)$ can be expressed as a linear combination of the Wick powers of $X_t$. This leads to the following proposition, which gives the explicit form for the source term $\mathscr{F}_\alpha$.

**Proposition 4** (Explicit Form of $\mathscr{F}_\alpha$ for Polynomial Nonlinearities). *Consider the SPDE (SPDE). Assume the drift term $F(t, X_t)$ is a polynomial in $X_t$, expressible as a finite sum of Wick powers:*

$$F(t, X_t) = \sum_{p=0}^{P} a_p(t) X_t^{\diamond p}, \tag{32}$$

*where $a_p(t)$ are deterministic operators. The corresponding component of the propagator source term, $\mathscr{F}_\alpha^{from\ drift} = \mathbb{E}[F(t, X_t)\xi_\alpha]$, is then given by the following explicit algebraic expression:*

$$\mathbb{E}[F(t, X_t)\xi_\alpha] = \sum_{p=0}^{P} a_p(t) \left( \sum_{\beta_1 + \cdots + \beta_p = \alpha} u_{\beta_1}(t) \dots u_{\beta_p}(t) \right). \tag{33}$$

*A similar, though more complex, algebraic expression exists for the contribution from the diffusion term $B(t, X_t)$ if it is also a polynomial in $X_t$.*

*Proof of Proposition 4.* Our goal is to find an explicit expression for the term $\mathbb{E}[F(t, X_t)\xi_\alpha]$. We start by substituting the assumed Wick power expansion of the nonlinearity $F(t, X_t)$:

$$\mathbb{E}[F(t, X_t)\xi_\alpha] = \mathbb{E}\left[ \left( \sum_{p=0}^{P} a_p(t) X_t^{\diamond p} \right) \xi_\alpha \right]. \tag{34}$$

By the linearity of expectation, we move the summation and the deterministic operator $a_p(t)$ outside:

$$\mathbb{E}[F(t, X_t)\xi_\alpha] = \sum_{p=0}^{P} a_p(t) \mathbb{E}\left[ X_t^{\diamond p} \xi_\alpha \right]. \tag{35}$$

The problem is now reduced to computing the expectation $\mathbb{E}[X_t^{\diamond p} \xi_\alpha]$ for each power $p$.

We first write the chaos expansion for the $p$-fold Wick power $X_t^{\diamond p}$. According to the Definition of the Wick product 4, and by recursively applying it, the propagator for the Wick monomial $\xi_\gamma$ in the expansion of $X_t^{\diamond p}$ is the convolution-like sum of the propagators of $X_t$. That is:

$$X_t^{\diamond p} = \sum_{\gamma \in \mathcal{J}} \left( \sum_{\beta_1 + \cdots + \beta_p = \gamma} u_{\beta_1}(t) \dots u_{\beta_p}(t) \right) \xi_\gamma. \tag{36}$$

Now, we substitute this expansion back into the expectation term we need to compute:

$$\mathbb{E}\left[ X_t^{\diamond p} \xi_\alpha \right] = \mathbb{E}\left[ \left( \sum_{\gamma \in \mathcal{J}} \left( \sum_{\beta_1 + \cdots + \beta_p = \gamma} u_{\beta_1}(t) \dots u_{\beta_p}(t) \right) \xi_\gamma \right) \xi_\alpha \right]. \tag{37}$$

Again, by linearity of expectation, we move the summations and propagators outside:

$$\mathbb{E}\left[ X_t^{\diamond p} \xi_\alpha \right] = \sum_{\gamma \in \mathcal{J}} \left( \sum_{\beta_1 + \cdots + \beta_p = \gamma} u_{\beta_1}(t) \dots u_{\beta_p}(t) \right) \mathbb{E}[\xi_\gamma \xi_\alpha]. \tag{38}$$

From Lemma 3, we know that the Wick monomials are orthonormal, i.e., $\mathbb{E}[\xi_\gamma \xi_\alpha] = \delta_{\gamma\alpha}$. The sum over $\gamma$ therefore collapses to a single term where $\gamma = \alpha$:

$$\mathbb{E}\left[ X_t^{\diamond p} \xi_\alpha \right] = \sum_{\beta_1 + \cdots + \beta_p = \alpha} u_{\beta_1}(t) \dots u_{\beta_p}(t). \tag{39}$$

Finally, we substitute this result back into the expression for $\mathbb{E}[F(t, X_t)\xi_\alpha]$:

$$\mathbb{E}[F(t, X_t)\xi_\alpha] = \sum_{p=0}^{P} a_p(t) \left( \sum_{\beta_1 + \cdots + \beta_p = \alpha} u_{\beta_1}(t) \dots u_{\beta_p}(t) \right). \tag{40}$$

This is the desired explicit form for the contribution of the drift term, thus completing the proof. □

**Remark 1** (Choice of chaos basis). *In this work we use the Wick–Hermite chaos basis, which is standard in the Wiener–chaos/SPDE literature and aligns naturally with the algebraic structure of Wick products and with existing propagator and error analyses (Lototsky et al., 2017; Luo, 2006; Neufeld and Schmocker, 2024). We note that there are alternative polynomial bases for approximating Brownian motion and Gaussian processes, such as the optimal bases studied by Foster et al. (Foster et al., 2020). These bases can be advantageous from a pure approximation point of view, but it is not yet clear whether they preserve the same simple chaos decomposition and Wick–product algebra needed to derive explicit propagator systems and source terms $\mathscr{F}_\alpha$. Exploring such alternative bases within the SDENO/F-SPDENO framework, and understanding the trade-offs between structural convenience and approximation optimality, is an interesting direction for future work.*

## C.3 SOME EXAMPLES

We provide some explicit examples for the conclusions provided in Theorem 1 and 2. We start with the SDEs.

**OU Process**

**Example 1** (Propagator System for the Ornstein-Uhlenbeck Process). *Consider the one-dimensional Ornstein-Uhlenbeck (OU) process, driven by a one-dimensional standard Brownian motion (i.e., $d = 1, m = 1$):*

$$dX_t = -\theta X_t dt + \sigma dW_t, \quad X_0 = x_0.$$

*Here, the drift is $F(t, X_t) = -\theta X_t$ and the diffusion is $B(t, X_t) = \sigma$ (a constant). We apply Theorem 1 to find the system for the propagators $u_\alpha(t) = \mathbb{E}[X_t \xi_\alpha]$. Since $m = 1$, the multi-index simplifies to $\alpha = (\alpha_i)_{i \in \mathbb{N}}$.*

*The ODE for $u_\alpha(t)$ is then given by:*

$$\frac{d}{dt} u_\alpha(t) = \mathbb{E}\left[-\theta X_t \xi_\alpha\right] + \sum_{i=1}^{\infty} \sqrt{\alpha_i}\, e_i(t)\, \mathbb{E}\left[\sigma \xi_{\alpha-e_i}\right]. \tag{41}$$

*We analyze the two terms on the right-hand side:*

- *Drift Term: Due to the linearity of expectation and the definition of $u_\alpha(t)$, we have:*
$$\mathbb{E}[-\theta X_t \xi_\alpha] = -\theta \mathbb{E}[X_t \xi_\alpha] = -\theta u_\alpha(t).$$

- *Diffusion Term: Since $\sigma$ is a constant and the Wick monomials are orthonormal with $\mathbb{E}[\xi_\beta] = \delta_{\beta 0}$:*
$$\mathbb{E}[\sigma \xi_{\alpha-e_i}] = \sigma \mathbb{E}[\xi_{\alpha-e_i}] = \sigma \delta_{\alpha-e_i, 0}.$$
*This term is non-zero only if the multi-index $\alpha - e_i$ is the zero index. This occurs only when $\alpha = e_i$ (i.e., $|\alpha| = 1$ and the only non-zero component is $\alpha_i = 1$).*

*Combining these results, the infinite system of ODEs for the propagators decouples into a clear hierarchy:*

- *For the mean ($|\alpha| = 0$, i.e., $\alpha = 0$):*
$$\frac{d}{dt} u_0(t) = -\theta u_0(t), \quad u_0(0) = x_0.$$

- *For the first-order chaos modes ($|\alpha| = 1$, i.e., $\alpha = e_i$ for some $i \in \mathbb{N}$):*
$$\frac{d}{dt} u_{e_i}(t) = -\theta u_{e_i}(t) + \sigma e_i(t), \quad u_{e_i}(0) = 0.$$

- *For all higher-order modes ($|\alpha| \geq 2$):*
$$\frac{d}{dt} u_\alpha(t) = -\theta u_\alpha(t), \quad u_\alpha(0) = 0.$$

*This example explicitly shows how a single SDE is transformed into an infinite, yet structured and sequentially solvable, system of deterministic ODEs. The solution for the higher-order modes is trivially $u_\alpha(t) \equiv 0$ for $|\alpha| \geq 2$.*

**Time-Reversal d-dimensional OU Process** In the previous example, we showed the application of Theorem 1 to the 1D OU process. It is well-known that the time-reversal OU process forms the denoising diffusion process for many generative tasks.

**Example 2** (Propagator System for the d-Dimensional Reverse-Time OU Process)**.** *In score-based generative models (Song et al., 2020), the reverse-time dynamics of the Ornstein-Uhlenbeck process are of central importance. Let $X_t \in \mathbb{R}^d$ be the state, driven by a $d$-dimensional standard Brownian motion $\overline{W}_t$ (hence, $m = d$). The reverse-time SDE is given by:*

$$dX_t = \left[ -\frac{1}{2}\beta(t)X_t - \beta(t)s_\theta(t, X_t) \right] dt + \sqrt{\beta(t)}d\overline{W}_t,$$

*where $\beta(t) > 0$ is a deterministic variance schedule, and $s_\theta(t, X_t) \approx \nabla_x \log p_t(X_t)$ is the score function, typically approximated by a neural network with parameters $\theta$. For our framework, we identify the drift and diffusion terms for the $j$-th component, $X_t^{(j)}$:*

- *Drift: $F^{(j)}(t, X_t) = -\frac{1}{2}\beta(t)X_t^{(j)} - \beta(t)s_\theta^{(j)}(t, X_t)$. This term is highly non-linear due to the neural network $s_\theta$.*

- *Diffusion: $B(t, X_t) = \sqrt{\beta(t)}I_d$, where $I_d$ is the $d \times d$ identity matrix. Thus, its components are $B^{(j,k)}(t, X_t) = \sqrt{\beta(t)}\delta_{jk}$.*

*We now apply Theorem 1 to derive the ODE system for the propagators $u_\alpha(t) \in \mathbb{R}^d$. The equation for the $j$-th component $u_\alpha^{(j)}(t)$ is:*

$$\frac{d}{dt}u_\alpha^{(j)}(t) = \mathbb{E}\left[ F^{(j)}(t, X_t)\xi_\alpha \right] + \sum_{k=1}^{d}\sum_{i=1}^{\infty} \sqrt{\alpha_{ki}}\, e_i(t)\, \mathbb{E}\left[ B^{(j,k)}(t, X_t)\xi_{\alpha - e_{ki}} \right].$$

*Let's analyze the two terms on the right-hand side.*

*1. Diffusion Term Since $B^{(j,k)}$ is deterministic and proportional to $\delta_{jk}$, the sum over $k$ collapses to a single term where $k = j$.*

$$\sum_{k=1}^{d}\sum_{i=1}^{\infty} \sqrt{\alpha_{ki}}e_i(t)\mathbb{E}[\sqrt{\beta(t)}\delta_{jk}\xi_{\alpha - e_{ki}}] = \sum_{i=1}^{\infty} \sqrt{\alpha_{ji}}e_i(t)\mathbb{E}[\sqrt{\beta(t)}\xi_{\alpha - e_{ji}}]$$

$$= \sqrt{\beta(t)}\sum_{i=1}^{\infty} \sqrt{\alpha_{ji}}e_i(t)\delta_{\alpha - e_{ji}, 0}. \tag{42}$$

*This term is non-zero only if $\alpha = e_{ji}$ for some $i \in \mathbb{N}$. In this case ($|\alpha| = 1$), the term simplifies to $\sqrt{\beta(t)}e_i(t)$.*

*2. Drift Term We can split the drift term using the linearity of expectation:*

$$\mathbb{E}\left[ F^{(j)}(t, X_t)\xi_\alpha \right] = \mathbb{E}\left[ \left( -\frac{1}{2}\beta(t)X_t^{(j)} - \beta(t)s_\theta^{(j)}(t, X_t) \right)\xi_\alpha \right]$$

$$= -\frac{1}{2}\beta(t)\mathbb{E}[X_t^{(j)}\xi_\alpha] - \beta(t)\mathbb{E}[s_\theta^{(j)}(t, X_t)\xi_\alpha]$$

$$= -\frac{1}{2}\beta(t)u_\alpha^{(j)}(t) - \beta(t)\mathbb{E}[s_\theta^{(j)}(t, X_t)\xi_\alpha]. \tag{43}$$

*The second term, $\mathbb{E}[s_\theta^{(j)}(t, X_t)\xi_\alpha]$, represents the projection of the non-linear score function onto the chaos basis. This term couples all propagator modes $\{u_\gamma\}$ in a highly complex way and generally does not have a simple closed form.*

*Combining the results, the ODE for the propagator component $u_\alpha^{(j)}(t)$ is:*

$$\boxed{\frac{d}{dt}u_\alpha^{(j)}(t) = -\frac{1}{2}\beta(t)u_\alpha^{(j)}(t) - \beta(t)\mathbb{E}[s_\theta^{(j)}(t, X_t)\xi_\alpha] + \sqrt{\beta(t)}\sum_{i=1}^{\infty} e_i(t)\mathbf{1}_{\{\alpha = e_{ji}\}}.} \tag{44}$$

*This example clearly illustrates how the Wiener Chaos Expansion transforms the problem. The simple, state-independent diffusion term of the original SDE becomes a deterministic forcing term that only acts on the first-order chaos modes. In contrast, the complex, non-linear part of the drift (the score function) is isolated into an abstract expectation term which is precisely what a neural operator is trained to approximate. This equation guides us to design $\mathcal{U}$-SDENO for the image generative model one-step sampler, see more details in Appendix F.3.*

**Stochastic Heat Equation**    We now show an example of Theorem 2 for SPDE.

**Example 3** (Propagator System for the Stochastic Heat Equation). *Consider the stochastic heat equation on a spatial domain $\mathcal{D} \subset \mathbb{R}^n$ with appropriate boundary conditions (e.g., Dirichlet), driven by additive Q-Brownian noise:*

$$dX_t = \Delta X_t dt + dW_t, \quad X_0(x) = \chi_0(x). \tag{45}$$

*In the context of our general SPDE form, the linear operator is the Laplacian, $A = \Delta$. The drift term is zero, $F = 0$, and the diffusion term $B$ is the identity operator. The Q-Brownian motion is expanded as $W_t(x) = \sum_{k=1}^{\infty} \sqrt{\lambda_k} \beta_t^k f_k(x)$. We apply Theorem 2 to find the system for the propagators $u_\alpha(t, x) = \mathbb{E}[X_t(x)\xi_\alpha]$. The general form of the PDE for $u_\alpha$ is:*

$$\frac{\partial}{\partial t} u_\alpha(t, x) = (\Delta u_\alpha)(t, x) + \mathscr{F}_\alpha(\{u_\gamma\}).$$

*The comprehensive deterministic source term $\mathscr{F}_\alpha$ is composed of contributions from the original drift $F$ and diffusion $B$, but since $F = 0$, then the contribution only from $B$, and it can be derived via Malliavin calculus. For this linear additive noise case, a detailed calculation shows that this contribution is non-zero only for first-order chaos modes. Specifically, for $\alpha = e_{ki}$ (the multi-index which is 1 at position $(k, i)$ and 0 elsewhere), the source term becomes:*

$$\mathscr{F}_{e_{ki}}(t, x) = \sqrt{\lambda_k} e_i(t) f_k(x). \tag{46}$$

*For all other multi-indices $\alpha$ (where $|\alpha| \neq 1$), the contribution from the diffusion term is zero. This analysis leads to a decoupled system of deterministic PDEs for the propagators:*

- *For the mean field ($|\alpha| = 0$, i.e., $\alpha = 0$):*

$$\frac{\partial}{\partial t} u_0(t, x) = \Delta u_0(t, x), \quad u_0(0, x) = \chi_0(x).$$

- *For the first-order chaos modes ($|\alpha| = 1$, i.e., $\alpha = e_{ki}$):*

$$\frac{\partial}{\partial t} u_{e_{ki}}(t, x) = \Delta u_{e_{ki}}(t, x) + \sqrt{\lambda_k} e_i(t) f_k(x), \quad u_{e_{ki}}(0, x) = 0.$$

- *For all higher-order modes ($|\alpha| \geq 2$):*

$$\frac{\partial}{\partial t} u_\alpha(t, x) = \Delta u_\alpha(t, x), \quad u_\alpha(0, x) = 0,$$

*which implies their solution is $u_\alpha(t, x) \equiv 0$.*

*This example demonstrates powerfully how a complex SPDE can be decomposed into a system of (often much simpler) deterministic PDEs.*

### C.4    A WCE OF PATH SIGNATURES AND A POSSIBLE ROUGH-SDENO

Differential equations driven by signals rougher than Brownian motion with Hurst parameter $\frac{1}{4} < H < \frac{1}{2}$ appear as variants of common SDEs (Cheridito et al., 2003), as well as in finance and parameter estimation. Such equations are defined by first taking the depth-$N$ (Stratonovich) signature of their drivers

$$S(X)_{s,t}^N = \sum_{n=0}^{N} \int_{s < u_1 < \cdots < u_n < t} dX_{u_1} \otimes \cdots \otimes dX_{u_n} \in T^{\leq N}(\mathbb{R}^d), \tag{47}$$

and solving the RDE $dY_t = f(Y_t)\,d\mathbf{X}_t$ where $\mathbf{X}$ is lift to a geometric $p$-rough path obtained via the signature. This signature can be viewed as a Taylor expansion capturing higher-order oscillations and cross-dimensional relations, and it uniquely characterises the path up to retracings (Lyons, 1998). The truncation level $N$ is set by the driver's roughness and the vector-field algebra: for a Gaussian driver with Hurst $H$, choose any $p > 1/H$ and take $N = \lfloor p \rfloor$. If the vector fields commute or are $k$-step nilpotent, the solution depends only on levels up to $k$, so one may take $N = \min\{k, \lfloor p \rfloor\}$ (e.g., $N = 2$ for $H > \frac{1}{3}$ and $N = 3$ for $\frac{1}{4} < H \le \frac{1}{3}$).

For centered Gaussian paths, each grade of $T^{\le N}(\mathbb{R}^d)$ admits a finite Wiener–Itô decomposition: the grade-$n$ signature component splits into chaos orders $m \in \{n, n-2, \ldots\}$ (Ferrucci and Cass, 2023). Let $w_m$ be the orthogonal projection onto the $m$-th Wiener chaos. For any multi-index of tensor words $(\gamma_1, \ldots, \gamma_n)$ with $n \le N$,

$$w_m\, S(X)_{s,t}^{\gamma_1 \cdots \gamma_n} = \sum_{\Pi \in \mathcal{P} \dashv\rangle \nabla_{n,m}} \delta^{(m)}\Big(P_{\Pi;s,t}^{\gamma_1, \ldots, \gamma_n}\Big), \qquad m \equiv n \pmod 2,\ m \le n,$$

where $\mathcal{P} \dashv\rangle \nabla_{n,m}$ is the set of partitions of $\{1, \ldots, n\}$ into $(n-m)/2$ unordered pairs and $m$ singletons. Each kernel $P_{\Pi;s,t}^{\gamma_1, \ldots, \gamma_n}$ is obtained by contracting, for every pair $\{i,j\} \in \Pi$, the elementary time-ordered kernels with the covariance $R^{\gamma_i, \gamma_j}$ (and its derivatives), while singleton slots remain as symmetrised time kernels; thus $P_{\Pi;s,t}^{\gamma_1, \ldots, \gamma_n} \in L^2([s,t]^m)$ is symmetric in its $m$ variables and $\delta^{(m)}$ denotes the $m$-fold Wiener integral. Ferrucci and Cass's key contribution is an "integrate–out" step for singletons, via Gaussian integration by parts, that replaces raw time variables with combinations of covariance derivatives (e.g., $\frac{1}{2} \partial_v R(v,v) - \partial_2 R(s,v)$), yielding $L^2$-admissible kernels and convergence for $H > \frac{1}{4}$ (Ferrucci and Cass, 2023).

> **Remark 2** (Minimal orders for chaos expansions of RDEs). *As each signature level requires corresponding chaos orders, and signature levels are governed by the driver's roughness and the vector-field algebra, the following guidelines select a chaos truncation for RDEs:*
>
> 1. *Additive or affine RDEs are captured exactly by truncation at first chaos.*
> 2. *For a Gaussian driver with Hurst parameter $H$, choose any $p > 1/H$ and form a geometric $p$-rough path (levels $1, \ldots, \lfloor p \rfloor$); to represent these levels, a chaos truncation of order $\lfloor p \rfloor$ is the minimal choice.*
> 3. *If the vector fields are Lie–nilpotent of step $k$ (all commutators of length $\ge k+1$ vanish), the solution depends only on levels up to $k$, so one may take the truncation order $L = \min\{k, \lfloor p \rfloor\}$.*

# D   DETAILED FORMULATION AND ANALYSIS OF $\mathcal{F}$-SPDENO

This section provides a more detailed formulation of the $\mathcal{F}$-SPDENO framework and a simple error-analysis viewpoint based on existing approximation results. Our goal is not to derive a new sharp theorem, but to decompose the approximation error into interpretable components that highlight the role of the main hyperparameters: the number of temporal basis functions $K$, the maximum Hermite order $M$, and the capacity of the FNO backbone.

## D.1   WIENER–CHAOS AND TEMPORAL-BASIS EXPANSIONS

Let $X_t(x)$ be the stochastic solution process of the SPDE (SPDE). As established, it admits a Wiener–chaos expansion:

$$X_t(x) = \sum_{\alpha \in \mathcal{J}} u_\alpha(t,x)\,\xi_\alpha,$$

where $\{\xi_\alpha\}$ are the Wick–Hermite chaos basis polynomials and $\{u_\alpha(t,x)\}$ are the deterministic propagator fields.

The core idea of our framework is to further represent each deterministic propagator field $u_\alpha(t, x)$ in an orthonormal temporal basis $\{\phi_k(t)\}_{k=1}^K \subset L^2([0, T])$ (e.g., a Haar basis), i.e.,

$$u_\alpha(t, x) \approx \sum_{k=1}^K u_{\alpha,k}(x)\, \phi_k(t),$$

where the coefficients $u_{\alpha,k}(x) = \int_0^T u_\alpha(t, x)\, \phi_k(t)\, dt$ are time-independent deterministic spatial fields.

Substituting this back into the chaos expansion gives a double-expansion representation for a single realization $u(t, x; \omega) \equiv X_t(x)(\omega)$:

$$u(t, x; \omega) \approx \sum_{\alpha \in \mathcal{J}} \Big( \sum_{k=1}^K u_{\alpha,k}(x)\, \phi_k(t) \Big) \xi_\alpha(\omega) = \sum_{k=1}^K \Big( \sum_{\alpha \in \mathcal{J}} u_{\alpha,k}(x)\, \xi_\alpha(\omega) \Big) \phi_k(t). \tag{48}$$

### D.2 NEURAL-OPERATOR FORMULATION AND RECONSTRUCTION

The previous expansion reveals a natural learning target for our operator. For a fixed noise realization $\omega$, the solution can be seen as an expansion in the temporal basis $\{\phi_k(t)\}$ with stochastic spatial coefficients

$$C_k(x; \omega) := \sum_{\alpha \in \mathcal{J}} u_{\alpha,k}(x)\, \xi_\alpha(\omega). \tag{49}$$

The underlying $u_{\alpha,k}(x)$ are deterministic, but $C_k(x; \omega)$ are random spatial fields through the chaos variables $\{\xi_\alpha(\omega)\}$.

$\mathcal{F}$-SPDENO is designed to learn the operator map from the initial condition $\chi_0(x)$ and truncated Wick features of the noise realization to the set of spatial coefficients $\{C_k(x; \omega)\}_{k=1}^K$. The Wick features for a realization $\omega$, truncated at a maximum chaos order $M$, are the scalar values

$$\mathcal{W}(\omega, M) = \{\xi_\alpha(\omega) : |\alpha| \le M\}.$$

The ideal operator can be written as

$$\{\widehat{C}_k(x; \omega)\}_{k=1}^K = \mathcal{G}_{M,K}\big(\chi_0(x), \mathcal{W}(\omega, M)\big), \tag{50}$$

and our $\mathcal{F}$-SPDENO model parameterises this map with an FNO:

$$\{\widehat{C}_k(x; \omega)\}_{k=1}^K = \mathcal{G}_\theta\big(\chi_0(x), \mathcal{W}(\omega, M)\big) = \mathrm{FNO}_\theta\big(\chi_0(x), \mathcal{W}(\omega, M)\big). \tag{51}$$

Here the scalar Wick features are treated as additional channels concatenated to the spatial field $\chi_0(x)$.

The final approximate solution $\hat{u}(t, x; \omega)$ is reconstructed from the FNO outputs via the temporal basis:

$$\hat{u}(t, x; \omega) = \sum_{k=1}^K \widehat{C}_k(x; \omega)\, \phi_k(t). \tag{52}$$

The parameters $\theta$ are trained by minimizing an empirical $L^2$ loss between reconstructed and ground-truth solutions over a dataset of $N$ trajectories:

$$\mathcal{L}(\theta) = \frac{1}{N} \sum_{i=1}^N \int_0^T \int_{\mathcal{D}} \big| u^{(i)}(t, x) - \hat{u}^{(i)}(t, x; \theta) \big|^2 \, dx\, dt. \tag{53}$$

### D.3 ERROR DECOMPOSITION, SENSITIVITY, AND UNIVERSAL APPROXIMATION

The above formulation suggests a natural way to think about the approximation error of $\mathcal{F}$-SPDENO. Rather than aiming for a new sharp bound, we use existing results on temporal approximations, Wiener–chaos truncations, and neural-operator universality to form a simple error decomposition that highlights the effect of $(K, M)$ and the FNO capacity.

Let $u$ be the true solution and $\hat{u}$ the approximation produced by $\mathcal{F}$-SPDENO with hyperparameters $(K, M)$ and parameters $\theta$. Using the triangle inequality and viewing each approximation step separately, it is natural to write

$$\mathbb{E}\big[\|u - \hat{u}\|_{L^2([0,T];\mathcal{H})}^2\big]^{1/2} \lesssim E_K + E_M + E_\theta, \tag{54}$$

where

- $E_K$ is the temporal projection error from using only $K$ basis functions,
- $E_M$ is the error from truncating the Wiener–chaos/Wick features to order $M$, and
- $E_\theta$ is the intrinsic approximation error of the FNO in learning the truncated operator $\mathcal{G}_{M,K}$.

Below we summarise how each term can be controlled under standard assumptions.

**Temporal projection error $E_K$.**

**Proposition 5** (Bound on temporal projection error $E_K$). *Assume that, for almost every $(x, \omega)$, the map $t \mapsto u(t, x; \omega)$ is Hölder continuous with exponent $s \in (0, 1]$. Then for an orthonormal basis such as the Haar basis,*

$$E_K^2 = \mathbb{E}\bigg[\bigg\|u - \sum_{k=1}^{K} C_k(x; \omega)\, \phi_k(t)\bigg\|_{L^2([0,T];\mathcal{H})}^2\bigg] \leq C_K\, K^{-2s}, \tag{55}$$

*where the constant $C_K$ depends on the Hölder norms of the solution paths but is independent of $K$.*

*Sketch.* By definition, $E_K^2$ is the expected $L^2([0,T];\mathcal{H})$-norm of the projection error onto $\mathrm{span}\{\phi_k\}_{k=1}^{K}$. Stochastic Fubini reduces this to bounding the $L^2([0,T])$-projection error of the scalar functions $t \mapsto u(t, x; \omega)$. For Hölder continuous functions, classical results on Haar (or wavelet) approximations yield

$$\|f - P_K f\|_{L^2([0,T])}^2 \leq C_{\mathrm{Haar}}\, K^{-2s}\, \|f\|_{C^s([0,T])}^2$$

for all $f \in C^s([0,T])$ (see, e.g., Bolin et al. (2020, Section 3.3) and references therein). Applying this pointwise in $(x, \omega)$ and integrating gives (55), with $C_K$ absorbing the expected Hölder norms. $\quad\square$

> **Remark 3** (Interpretation of temporal error). *Proposition 5 provides a guideline for choosing the number of temporal basis functions $K$: as $K$ increases, the temporal projection error decreases at a rate controlled by the Hölder exponent $s$. For smoother SPDE solutions (larger $s$), relatively small $K$ may already suffice, whereas highly irregular temporal dynamics may require larger $K$ to control this source of error.*

**Wick feature truncation error $E_M$.** The truncation error $E_M$ arises from providing the ideal operator $\mathcal{G}_{M,K}$ with a finite set of Wick features $\mathcal{W}(\omega, M)$ instead of the full chaos expansion. This error is governed by the decay of the Wiener–chaos tail and is closely related to the Malliavin smoothness of the SPDE solution with respect to the noise.

For a broad class of SPDEs (in particular, equations with affine coefficients as in Neufeld and Schmocker (2024)), it is known that the chaos tail decays factorially in the chaos order. Under the assumptions in Neufeld and Schmocker (2024, Sections 4 and 6.6), one obtains bounds of the form

$$E_M^2 \lesssim \sum_{|\alpha|>M} \|u_\alpha\|_{L^2([0,T];\mathcal{H})}^2 \leq C_M\, \frac{(\Gamma_T)^{M+1}}{(M+1)!}, \tag{56}$$

for suitable constants $C_M, \Gamma_T$ depending on the SPDE coefficients and time horizon but independent of $M$. The proof uses the Stroock–Taylor representation of chaos coefficients in terms of Malliavin derivatives and inductive bounds on these derivatives (see Neufeld and Schmocker (2024, Propositions 6.3 and 6.5)); we refer to that work for full details.

**Remark 4** (Interpretation of Wick feature error). *The factorial decay in* (56) *indicates that for sufficiently smooth SPDEs most of the stochastic energy is carried by low-order chaos modes, and the contribution of high-order modes becomes negligible very rapidly. This provides a theoretical justification for using relatively small truncation orders $M$ (e.g., $M \in \{2, \ldots, 5\}$) in practice.*

**Neural-operator approximation error $E_\theta$.**    The universality of FNO as a neural operator has been established in Kovachki et al. (2023, Section 8.3, Theorem 13). Let $\mathcal{G}_{M,K} : \mathcal{H} \times \mathbb{R}^{N_M} \to \mathcal{H}^K$ denote the ideal operator mapping an initial condition $\chi_0$ and a finite vector of Wick features $\mathcal{W}(\omega, M)$ to the true temporal coefficients $\{C_k(x; \omega)\}_{k=1}^K$. Our $\mathcal{F}$-SPDENO model, with an FNO backbone, is a specific parametrisation of this general neural-operator framework. Under the regularity assumptions required by Kovachki et al. (2023), the universal approximation theorem guarantees that for any $\varepsilon > 0$ there exists an FNO architecture $\mathcal{G}_\theta$ with sufficient capacity such that

$$E_\theta = \mathbb{E}\left[ \left\| \mathcal{G}_{M,K}(\chi_0, \mathcal{W}) - \mathcal{G}_\theta(\chi_0, \mathcal{W}) \right\|_{\mathcal{H}^K}^2 \right]^{1/2} < \varepsilon. \tag{57}$$

**Remark 5** (Interpretation of operator error). *The universality of FNO means that, under the stated assumptions, the intrinsic approximation error $E_\theta$ can be made arbitrarily small by increasing the capacity of the FNO (e.g., number of layers, hidden dimension, or Fourier modes).*

**Putting the pieces together.**    Combining the temporal projection bound (55), the chaos-truncation bound (56), and the neural-operator approximation guarantee (57) yields the heuristic decomposition (54). We emphasise that this decomposition is not intended as a new sharp error theorem, but rather as a simple framework for understanding how the architectural choices $(K, M)$ and the FNO capacity affect the overall approximation error. In Appendix F, we provide empirical evidence that the behaviour of $\mathcal{F}$-SPDENO under variations of $K$, $M$, and model capacity is consistent with these qualitative predictions.

## E    DETAILED FORMULATION AND ANALYSIS OF SDENOS

Following the analysis of $\mathcal{F}$-SPDENO, we briefly summarise the formulation of SDENOs for SDEs. The construction again has three steps: (i) Wiener–chaos representation of the solution, (ii) parametrisation of the propagators with a time-dependent neural network, and (iii) reconstruction and training.

**WCE of the SDE solution.** By Theorem 1, the strong solution $X_t \in \mathbb{R}^d$ to the SDE (SDE) admits a Wiener–chaos expansion

$$X_t = \sum_{\alpha \in \mathcal{J}} u_\alpha(t)\, \xi_\alpha, \tag{58}$$

where $\{\xi_\alpha\}_{\alpha \in \mathcal{J}}$ are scalar Wick–Hermite chaos basis elements and $\{u_\alpha(t)\}_{\alpha \in \mathcal{J}}$ are deterministic propagators $u_\alpha : [0, T] \to \mathbb{R}^d$. SDENO aims to approximate these propagator functions up to a finite chaos order $M$.

**Propagator network.** We parametrise the truncated set of propagators with a single time-dependent neural network, the *propagator network* $\mathcal{P}_\theta$. Let $\mathcal{J}_M = \{\alpha \in \mathcal{J} : |\alpha| \leq M\}$ denote the truncated index set and $N_M = |\mathcal{J}_M|$ its cardinality. The network realises a map

$$\mathcal{P}_\theta : [0, T] \to \mathbb{R}^{d \times N_M}, \tag{59}$$

so that, for each $t \in [0, T]$, the columns of $\mathcal{P}_\theta(t)$ are the approximated propagator vectors $\{\hat{u}_\alpha(t)\}_{\alpha \in \mathcal{J}_M}$. In practice we use a simple MLP (possibly with positional encodings of $t$) as the backbone for $\mathcal{P}_\theta$, although other architectures are possible.

**Reconstruction and training.** For a given noise realisation $\omega$, we compute the truncated vector of Wick features

$$\mathcal{W}(\omega, M) = (\xi_\alpha(\omega))_{\alpha \in \mathcal{J}_M} \in \mathbb{R}^{N_M}.$$

The SDENO approximation of the SDE solution at time $t$ is then reconstructed as

$$\hat{X}_t(\omega; \theta) = \sum_{\alpha \in \mathcal{J}_M} \hat{u}_\alpha(t) \, \xi_\alpha(\omega), \tag{60}$$

i.e., as a linear combination of the learned propagator values and the Wick features. The parameters $\theta$ are trained by minimising the mean-squared error between the true and reconstructed solutions over time and over the noise law:

$$\mathcal{L}(\theta) = \mathbb{E}_{\omega \sim \mathbb{P}} \Big[ \int_0^T \| X_t(\omega) - \hat{X}_t(\omega; \theta) \|^2 \, dt \Big]. \tag{61}$$

**Remark 6.** *For SDENOs, in the SDE setting there is no temporal basis truncation, so the total error can be naturally viewed as the sum of two contributions: (i) the Hermite/WCE truncation error at order M, and (ii) the approximation error of the propagator network. Given the analysis is very similar to the case in $\mathcal{F}$-SPDENO, we omit the analysis here.*

# F   DETAILED EXPERIMENTAL SETTINGS AND ADDITIONAL RESULTS

In this section, we provide details about the experiments conducted in the main body of the paper. For each experiment, we will include model data generation, ground truth solution generation, train-test split, model architecture, baseline information, and additional ablation studies, as well as hyperparameter sensitivity analyses, parameter comparisons, and execution times. Furthermore, we also present additional exploration results for our model in a broader range of SPDE/SDEs.

## F.1   DETAILS IN DYNAMIC $\Phi_1^4$

We recall that the dynamic $\mathbf{\Phi_1^4}$ model, which takes the form

$$dX_t = (\Delta X_t + 3X_t - X_t^3)dt + dW_t, \quad X_0 = \chi_0, \tag{62}$$

**Data Generation**   The dataset for the one-dimensional dynamic $\Phi_1^4$ model (also known as the Allen-Cahn or Ginzburg-Landau equation) was generated using a custom numerical solver mirroring the implementation in the `torchspde` (Salvi et al., 2022) library to ensure consistency with established benchmarks. The process involves two main components: generating the driving Q-Brownian motion and numerically solving the SPDE. The spatial domain was set to the one-dimensional torus $\mathbb{T}^1 = [0, 1]$, discretized with a spatial step of $\Delta x = 1/128$. The dynamics were simulated over the time interval $[0, T]$ with a final time of $T = 0.05$ and a time step of $\Delta t = 10^{-3}$. The driving Q-Brownian motion, $W_t$, was synthesized using a Karhunen-Loève-like expansion. Specifically, a standard one-dimensional Brownian motion was generated for each of the $N_x + 1$ spatial grid points. These independent Brownian paths were then projected onto a sinusoidal basis $\{\sin(j\pi x/L)\}_{j=1}^{N_x+1}$ to create a spatially correlated noise field. This construction yields a cumulative space-time noise field $W \in \mathbb{R}^{N \times (N_t+1) \times (N_x+1)}$, where $N$ is the number of trajectories. The global intensity of the noise was controlled by a parameter $\sigma$, which was set to $\sigma = 0.1$. The initial conditions, $\chi_0$, were generated to be either deterministic (zero field) or stochastic. In the stochastic case, they were sampled from a random field constructed from a sum of sinusoidal functions with decaying amplitudes, controlled by a parameter $\kappa$. This allows for the study of the solution operator's dependence on both the noise path $W$ and the initial state $\chi_0$. It should be noted that $N = 1000$ represents the number of samples used for training; in total, we sampled 1200 samples, resulting in 200 samples for testing.

**Stochastic simulation of Wiener processes**   The space-time noise $W(t, x)$ was generated to be "white" in time and spatially correlated. This construction follows the spectral approximation of a Q-Brownian motion, where the spatial correlation structure is determined by the eigenfunctions of the covariance operator $Q$. The process begins by generating $N_x + 1$ independent one-dimensional standard Brownian motions, $(\beta_t^j)_{j=0}^{N_x}$, one for each discrete spatial point. The spatially correlated noise field $W(t, x)$ is then constructed by projecting these independent Brownian motions onto a deterministic orthonormal basis, which, in this case, consists of sinusoidal functions $W(t, x) = \sum_{j=0}^{N_x} \beta_t^j \phi_j(x)$. Here, the basis functions are given by $\phi_j(x) = \sqrt{2/L} \sin(j\pi x/L)$, where $L$ is the length of the spatial domain. This corresponds to the Karhunen-Loève expansion of a Q-Brownian motion whose covariance operator $Q$ is approximated by this sinusoidal basis, a method detailed in (Lord et al., 2014). See (Salvi et al., 2022) for more detailed descriptions.

**Baselines** We exactly follow the baselines as well as their architectures included in (Salvi et al., 2022; Gong et al., 2023), including NCDE, NCDE-FNO, NRDE (Lu et al., 2022), FNO (Li et al., 2020), and NSPDE (Salvi et al., 2022). We note that we did not include the models presented in (Gong et al., 2023; Hu et al., 2022) in our baselines as they are designed specifically to conquer SPDEs with regularity structures. Furthermore, it may be possible to directly insert the noise trajectory into FNO; however, given that such a setting is not very meaningful, and FNO is commonly used to learn the deterministic coefficients in the Fourier domain, we follow Salvi et al. (2022) and claim that $(X_0, W) \mapsto X$ is not applicable to FNO. It is worth noting that, even though such a setting does not apply to FNO, $\mathcal{F}$-SDENO overcomes this limitation by reordering the chaos and feeding FNO with Q-Brownian trajectory's Wick features, i.e., $\mathcal{W}(x)$, so that the static, time-independent coefficients can be learned inside FNO.

**Model Architecture and Training** The $\mathcal{F}$-SPDENO in this experiment is conducted with two major modules, (i) the initial computation of the Wick features, i.e., $\mathcal{W}(x)$ and the selected Haar basis, and (ii) a backbone FNO model (Li et al., 2020) to learn the time-independent coefficients. The $\mathcal{F}$-SPDENO is trained in a supervised learning framework. The models were trained by minimizing the Mean Squared Error (MSE) loss, which corresponds to the empirical $L^2$ error over the training samples. We employed the Adam optimizer for its efficiency and stability in training deep neural networks. The initial learning rate was set to (1e-3), and a batch size of (64) was used. The models were trained for a total of (200) epochs on NVIDIA H200 GPUs. To ensure robust convergence and prevent overfitting, a learning rate scheduler was implemented to reduce the learning rate by a factor of (10) whenever the validation loss plateaued for (15) consecutive epochs. It is worth noting that, in our SPDE settings, we did not include the task of $X_0 \mapsto X$ as this setting is not applicable to our model since our model computes Wick features based on the chaos trajectories.

**Hyperparameter Sensitivity** Based on our sensitivity analysis in D.3, we conduct empirical verification for the conclusion which in. For SPDE experiments, we consider the model's sensitivity on the following hyperparameters. (i) number of Haar basis; (ii) Hermite polynomial orders $M$; (iii) Number of layers of FNO. For each test of the sensitivity, we change the value of on one hyperparameter while fixing the remaining two. Model's learn accuracy ($L^2$ error) is presented in the Figure 9. From the initial observation, one can find that with the increase in the Haar basis, the error drops, with the lowest value obtained at 64, and the error re-increases at the value of 96. This suggests a trade-off between representation capacity and generalization. While a larger number of Haar basis functions provides a finer resolution to represent the temporal dynamics of the noise, an excessively large basis for a short time horizon (50 time steps in the $\Phi_1^4$ setting) may cause the model to overfit to spurious patterns in the training set's specific noise paths, thereby degrading performance on the unseen test set.

A similar trend of diminishing returns is observed for the **Hermite polynomial order** ($M$). The error decreases significantly as $M$ increases from 1 to 4, indicating the importance of capturing higher-order stochastic moments of the solution. However, increasing $M$ beyond 4 yields only a marginal improvement, suggesting that lower-order chaos modes contain most of the solution's variance.

Finally, increasing the **number of FNO layers** consistently improves performance up to 6 layers, reflecting the benefits of a deeper model with greater expressive power for learning the complex propagator operator. A further increase to 8 layers leads to a slight degradation in performance, likely due to optimization challenges or overfitting. Based on this analysis, an optimal and computationally efficient configuration was selected for our main experiments.

## F.2 EXPERIMENTAL DETAILS FOR 2D NAVIER STOKES EQUATION

We further show the details in our experiment on 2D NS equation, which takes the form as

$$dX_t = \left( \nu\,\Delta X_t \; - \; U[X_t]\cdot\nabla X_t \; + \; f \right) dt \; + \; \sigma\, dW_t, \;\; (t,x) \in [0,T] \times \mathbb{T}^2, \;\; X_0 = \chi_0 \in L_0^2(\mathbb{T}^2).$$

Here we recall $U[X_t] \coloneqq \nabla^\perp (-\Delta)^{-1} X_t$ with $\nabla^\perp \coloneqq (-\partial_{x_2}, \partial_{x_1})$ where $\partial$ denotes the partial derivative of the spatial variable (i.e., $x = (x_1, x_2)$). The parameter $\nu > 0$ is the viscosity; $f$ is a deterministic vorticity forcing.

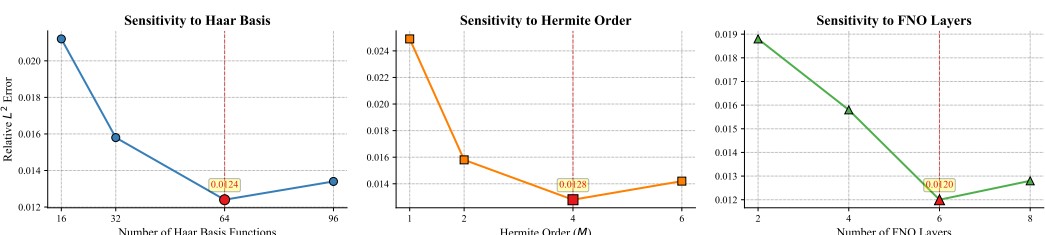

Figure 9: Accuracy sensitivity on hyperparameters of number of Haar basis, Hermite polynomial, and number of Fourier layers.

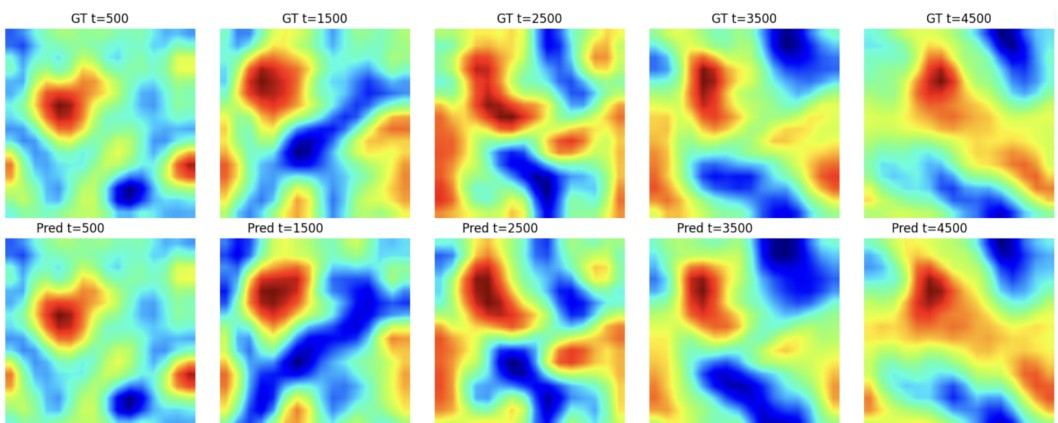

Figure 10: Performance on 2D NS equation in $16 \times 16$ resolution.

**Data Generation** We generate the dataset for the 2D Navier-Stokes equation on a two-dimensional torus $\mathbb{T}^2 = [0,1]^2$, discretized with a spatial resolution of $16 \times 16$ and $64 \times 64$. The procedure is the same as illustrated in (Salvi et al., 2022; Gong et al., 2023; Hu et al., 2022). The ground truth solution for the vorticity field $X(t,x)$ is obtained by numerically solving the equation using a pseudo-spectral method. The simulation is run for a total duration of $T = 15$ seconds with a time step of $\Delta t = 10^{-3}$, resulting in 15,000 time steps per trajectory.

The numerical solver operates in the Fourier domain, employing the Crank-Nicolson scheme for the viscous term and an explicit scheme for the non-linear advection term. A 2/3 dealiasing rule is applied to ensure numerical stability. The viscosity is set to $\nu = 10^{-4}$. The initial condition for the vorticity, $X_0$, is sampled from a Gaussian Random Field (GRF). This is achieved by defining a power spectrum in the Fourier domain with parameters $\alpha = 3$ and $\tau = 3$, and then transforming it back to the physical space, generating spatially correlated random fields.

The forcing consists of both deterministic and stochastic components. The deterministic forcing $f$ is a stationary and defined as $f(x_1, x_2) = 0.1(\sin(2\pi(x_1 + x_2)) + \cos(2\pi(x_1 + x_2)))$. The stochastic forcing term $\sigma dW_t$ is modeled as a Wiener process that is band-limited in the Fourier domain. Specifically, its power is concentrated between wavenumbers $k_{\min} = 4$ and $k_{\max} = 8$, with a noise amplitude of $\sigma = 0.01$. This setup ensures that the stochastic energy is injected at specific spatial scales. In total, we generated 10 distinct trajectories, each serving as a sample for our training and evaluation processes.

**Model Architecture and Training** The backbone FNO architecture consists of 6 Fourier layers with a latent dimension (width) of 64. For the spatial dimensions, we truncate the Fourier series at 8 modes. The model takes two inputs: the initial vorticity field $X_0(x)$ and the pre-computed Wick-Hermite features of the stochastic forcing, calculated up to order $M = 2$. Its objective is to directly predict the coefficients of the solution trajectory projected onto a Haar temporal basis of size 256. We note that even in theory, the approximation power of our model increases with the temporal

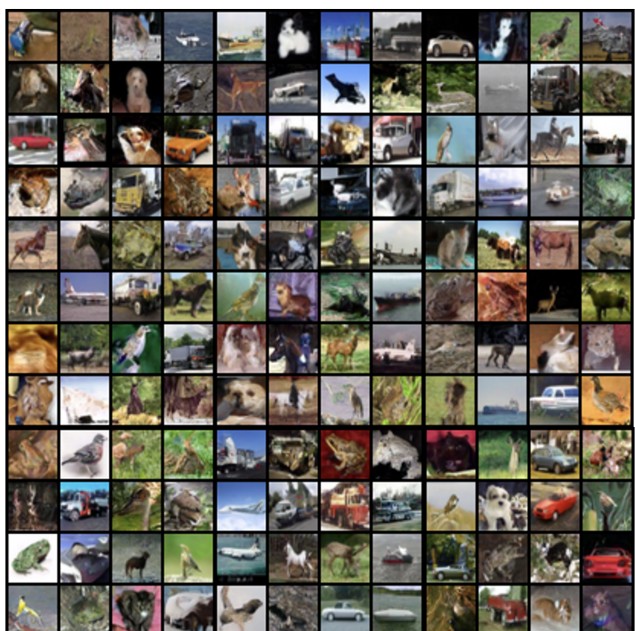

Figure 11: Additional CIFAR10 results.

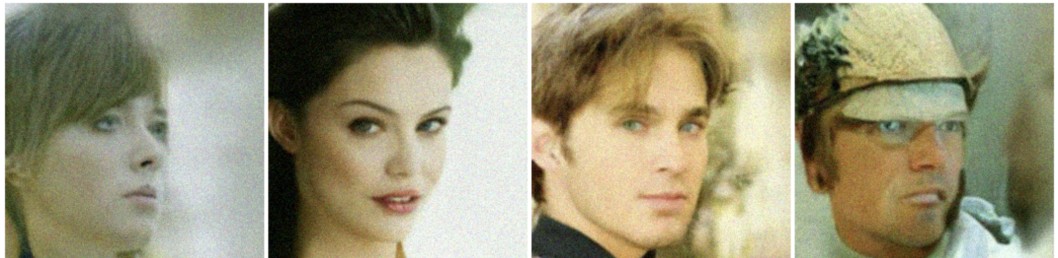

Figure 12: Additional unconditional generation on Celeb-HQ dataset.

basis dimension, in practice, we found that even in this long-range trajectory, time basis up to 256 is enough. The model is trained end-to-end for 30 epochs using the Adam optimizer with a learning rate of $10^{-3}$ and a weight decay of $3 \times 10^{-4}$. We employ a ReduceLROnPlateau learning rate scheduler to stabilize training. The loss function is the relative $L^2$ error, computed between the full solution trajectories reconstructed from the model's output and the ground truth solutions. Finally, we present our model's performance on 16×16 resolution in Figure 10.

### F.3 EXPERIMENTAL DETAILS FOR DIFFUSION ONE-STEP SAMPLER

$\mathcal{U}$-**SDENO Model Architecture Motivation** The architecture of our $\mathcal{U}$-SDENO sampler is directly motivated by the structure of the propagator system's governing ODEs, as derived in Equation (44). We recall it takes the form

$$\frac{d}{dt}u_\alpha^{(j)}(t) = -\frac{1}{2}\beta(t)u_\alpha^{(j)}(t) - \beta(t)\mathbb{E}[s_\theta^{(j)}(t, X_t)\xi_\alpha] + \sqrt{\beta(t)}\sum_{i=1}^{\infty}e_i(t)\mathbf{1}_{\{\alpha=e_{ji}\}}.$$

This equation reveals that the dynamics of each propagator $u_\alpha(t)$ are composed of two parts: (i) a simple, linear term and a deterministic forcing term (acting only on first-order chaos), both of which are analytically straightforward; and (ii) a complex, non-linear coupling term, $-\beta(t)\mathbb{E}[s_\theta(t, X_t)\xi_\alpha]$, which encapsulates the entire difficulty of the problem. Crucially, this challenging term is determined by the score function $s_\theta(t, X_t)$, which is approximated by the **U-Net** in the pre-trained diffusion model. Standard U-Net architectures for diffusion are explicitly conditioned on the time step $t$, typically via a learned time embedding (Huang et al., 2020). This means the U-Net can serve as a

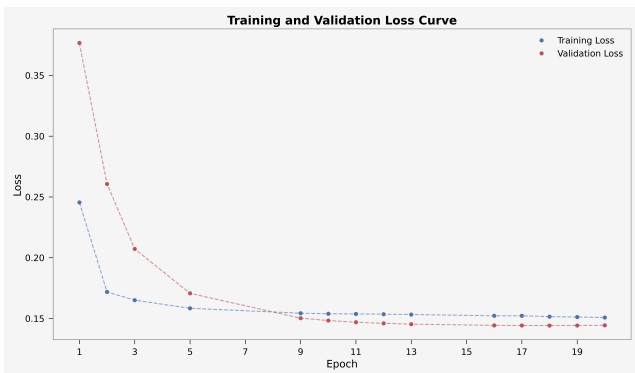

Figure 13: Training loss for CIFAR10

network that outputs the solution of time-reversal SDE implicitly, after feeding it with $t$ and Wick features, i.e., $\xi(\omega)$.

**Pretrained Models and Data Trajectory Sampling**  we utilize pretrained the Denoising Diffusion Probabilistic Models (DDPM) available from the Hugging Face model hub, including the `google/ddpm-cifar10-32` and `google/ddpm-celebahq-256` checkpoints. The core of our data generation process is to sample complete solution trajectories from these models. For each desired final image, we execute the standard DDPM reverse diffusion process, stepping backward from $t = T$ to $t = 0$ for a total of 1000 timesteps. At each step $t$, the pre-trained U-Net predicts the noise, from which the posterior mean $\mu_t$ and variance $\sigma_t^2$ are computed. The subsequent state $X_{t-1}$ is then sampled as $X_{t-1} = \mu_t + \sigma_t z_t$, where $z_t \sim \mathcal{N}(0, \mathbf{I})$. Crucially, for each full reverse trajectory, we record the entire sequence of the sampled standard Gaussian noise vectors, $\{z_t\}_{t=T-1}^0$. This sequence serves as a discrete realization of the driving reverse-time Brownian path, $d\overline{W}_t$. The final training dataset for our $\mathcal{U}$-SDENO is then constructed as a set of pairs, where each pair consists of a complete noise trajectory $\{z_t\}$ and its corresponding final generated image $X_0$.

We note that for both CIFAR10 and CelebA-HQ datasets, we sample 1 million $d\overline{W}$ trajectories for our model training, followed by the previous work in (Zheng et al., 2023). More importantly, given that each trajectory contains 1000 steps, it is not possible to train a model with all steps included. For this case, we conducted the uniform sampling to reduce the time steps to 32 (Zheng et al., 2023).

**Unconditional Sampling**  We conduct the unconditional sampling for our model for Celebhq dataset. We first synthesize a novel, random noise trajectory $\{\hat{z}_t\}_{t=T-1}^0$. Each noise vector $\hat{z}_t$ in this sequence is independently drawn from the standard multivariate Gaussian distribution, $\mathcal{N}(0, \mathbf{I})$, mimicking the statistics of the noise paths used during training. This complete noise path is then pre-processed to compute its corresponding Wick-Hermite features. These features are subsequently fed into the trained $\mathcal{U}$-SDENO model in a single forward pass.

**Additional Results**  In Figure 11 we show some additional CIFAR10 learning outcomes (without ground truth from the pretrained model), and in Figure 12 we present unconditional sampling results for Celebhq. In addition, we also show the training loss curve for CIFAR10 in Figure 13.

F.4    EXPERIMENTAL DETAILS FOR TSBM

$\mathcal{G}$-**SDENO Model Architecture Motivation**  Similar to the motivation of $\mathcal{U}$-SDENO, we design the $\mathcal{G}$-SDENO followed by the fact that graph neural networks (Kipf and Welling, 2017) can be served as the feature propagator of the brain regions. Different from the $\mathcal{U}$-SDENO, in GNNs, there is no time encoding inside; accordingly, time embedding is conducted outside the GNN process. Two outpus, i.e., GNN feature and time embedding, will be merged by inner product.

**Data Acquisition and Preprocessing**  The brain graph data used in this study is constructed sourced from the Human Connectome Project (HCP) Young Adult dataset (Van Essen et al., 2013). The graph

topology is defined by the Glasser multi-modal parcellation (MMP1.0), which delineates 360 distinct cortical areas that serve as the nodes of our graph (Glasser et al., 2016).

To determine the spatial embedding for each of the 360 nodes, we processed the HCP's standard 32k fs_LR cortical surface meshes. First, for each cerebral hemisphere, we identified the complete set of vertices belonging to each of the 360 parcels using the provided `.dlabel` file. The definitive 3D coordinate for each node (parcel) was then computed as the **median** of the coordinates of all its constituent vertices. This robust aggregation provides a geometrically central and stable position for each brain region. The edges of the graph, representing the underlying functional connectivity, are determined from a sparse inverse covariance matrix estimated from the fMRI signals, following the methodology in Yang (2025). The features associated with each node are derived from task-based functional MRI (tfMRI) signals corresponding to a working memory task. Specifically, our experiment aims to model the transition of brain states from a high-energy "2-back" working memory task (the initial distribution) to a lower-energy "0-back" task (the target distribution), providing a meaningful context for evaluating the topological interpolation capabilities of the model (Yang, 2025).

In terms of the node feature variation metric. We use the well-known Dirichlet energy. Given the node feature vector (fMRI signal) $x \in \mathbb{R}^N$ and the graph Laplacian matrix $L \in \mathbb{R}^{N \times N}$, the total Dirichlet energy of the graph is defined as $x^T L x$. Following the implementation in our analysis code, we define the energy $e_i$ for an individual node $i$ as the $i$-th component of the element-wise product between the signal and its Laplacian-transformed counterpart.

**TSBM Pretraining** The pretraining of the Topological Schrödinger Bridge Matching (TSBM) model follows the **alternating training scheme**. The process involves iteratively training a forward SDE and a backward SDE to progressively match the initial and target distributions. In our brain signal experiment, this corresponds to mapping the "2-back" task brain states to the "0-back" task states. Each SDE is driven by a GNN with 4 residual blocks, tasked with learning the drift term of the process. We employ a variance-exploding (VE) SDE formulation with a noise parameter of $\sigma = 0.5$. The model is trained for a total of 2000 iterations. For each iteration in the alternating loop, we first train the forward SDE for one epoch and then train the backward SDE for one epoch. The training is performed using the Adam optimizer with a learning rate of $10^{-3}$ for both networks and a batch size of 64. A step-wise learning rate scheduler is used, reducing the learning rate by a factor of 0.5 every 500 iterations to ensure stable convergence. Finally, the model is evaluated via the 1-Wasserstein distance, resulting in 9.51.

$\mathcal{G}$-**SDENO Training and An Interesting Observation** The training of $\mathcal{G}$-SDENO is similar to $\mathcal{U}$-SDENO, except the time embedding is conducted outside the GNN propagation; therefore, we omit it here. One interesting observation we found in our experiment is that our results are generally invariant across the GNNs, such as GCN (Kipf and Welling, 2017), and ChebyNet (Defferrard et al., 2016). We initially believe that this is due to the fact that all these GNNs will tend to smooth the node features so that the feature difference will asymptotically become identical, resulted in the over-smoothing problem (OSM). To this end, we test those GNNs that can provably avoid the OSM problem, such as generalised framelet model (Han et al., 2022). To our surprise, even when the GCN model is high-frequency dominant, meaning that node features tend to distinguish from each other through the propagation, we will find that both TSBM and our model show similar results compared to those GCN and ChebyNet. That may lead to a deeper exploration on the relationship between the diffusion task (i.e., interpolation) and the denoising models (propagator networks in our case), we leave this task to future works.

F.5    EXPERIMENTAL DETAILS FOR EXTRAPOLATION

**Data Generation** For both OU and Heston cases, we simulate a total of 500 sample paths ($P = 500$), each spanning a time horizon of $T = 1.0$ second with 100 discrete time steps ($N = 100$). The ground truth trajectories are generated using the Euler-Maruyama numerical integration scheme.

**Model Architecture and Training** For simplicity reasons, our design of SDENO for OU and Heston models comes with a relatively simple architecture, in which the propagator ODE is approximated by a MLP. The difference is that for the OU process, which only contains one Brownian motion, we only need one MLP for the propagator, whereas for Heston, we deploy two MLPs. The model is

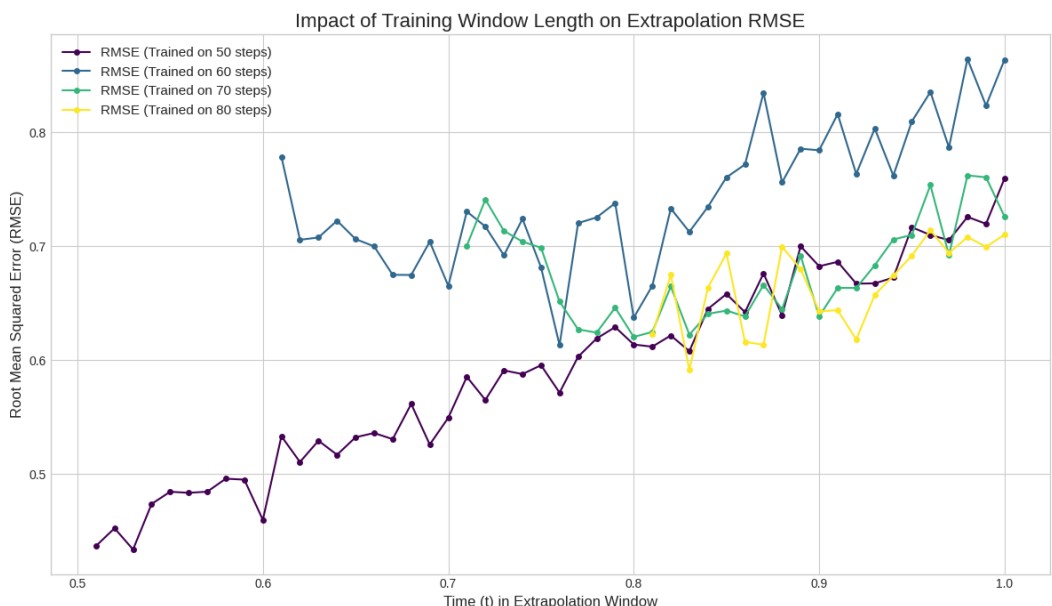

Figure 14: RMSE of SDENO extrapolation with varying training time lengths from 50 to 80.

trained in a specific extrapolation setting. For each sample path in the training set, which spans a total of $N = 100$ time steps, the model is only provided with the initial segment of the noise trajectory, corresponding to the first 75 time steps ($T_{\text{train}} = 75$). The loss, calculated as the Mean Squared Error (MSE), is also only computed over this initial $[0, 75]$ time interval. However, during evaluation, the model is fed the complete noise trajectory and tasked with predicting the solution over the entire $[0, 100]$ interval, thereby testing its ability to generalize to the unseen future segment $[76, 100]$. The training for all models is performed for 2000 epochs using the AdamW optimiser with a learning rate of $10^{-3}$.

**Trade off Between Training and Extrapolation Quality**  It is assumed that if the model is fed with more training data, then is extrapolation power would go up. To verify this assumption, we retain our model with the training window ranging from 50 to 80, i.e., extrapolation set from 50 to 20. The result is contained in Figure 14. One can see that (i) all error goes up with the increase of time in the unseen future; (ii) the model with the highest training samples (i.e., 80) shows the average lowest extrapolation error, whereas in general such error increases with the decrease of the training set. This empirical result is well-supported by foundational principles in statistical learning theory. Specifically, a longer observation window provides the model with more information to better learn the true underlying dynamics of the stochastic process, resulting in a learned operator with lower generalization error.

### F.6 Experimental Details for Parameter Estimation

**Generating Meta-Path and Meta-SDENO Training**  The foundation of our parameter estimation experiment is the creation of a diverse "Meta-path" dataset and the training of a generalized "Meta-SDENO" model on it. This pre-trained model serves as the high-fidelity forward solver within the Ensemble Kalman Filter framework.

**Meta-Path Generation**: The dataset is generated using the Ornstein-Uhlenbeck (OU) process, defined as $dX_t = \theta(\mu - X_t)dt + \sigma dW_t$. Instead of using a fixed set of parameters, we create a rich ensemble of trajectories by sampling the SDE parameters $\boldsymbol{\gamma} = [\theta, \mu, \sigma]$ from wide prior distributions. Specifically, for each training epoch, we sample $\ell = 250$ unique parameter combinations from uniform distributions ($\theta \sim U(0, 5)$, $\mu \sim U(-3, 3)$, $\sigma \sim U(0, 0.5)$). For each sampled parameter triplet, we generate a batch of $P = 300$ trajectories using an Euler-Maruyama solver. An important detail is that the same set of underlying Wiener increments $dW_t$ is reused across all parameter sets, which allows the model to isolate the influence of the parameters on the solution path.

**Meta-SDENO Architecture and Stepwise Training**: The *Meta*-SDENO model shares the same architecture as the one used in the extrapolation experiments. However, its training objective is fundamentally different, designed specifically to prepare it for the parameter estimation task. The training proceeds as follows: Within each epoch, we first sample a diverse batch of $\ell = 250$ parameter triplets $\boldsymbol{\gamma} = [\theta, \mu, \sigma]$. For each triplet, we generate $P = 300$ full-length OU process trajectories. The core of the supervision scheme occurs at each time step $t$ along these trajectories. The *Meta*-SDENO is provided with the history of the Wiener increments $dW$ up to time $t$ and is tasked with predicting only the next state, $\widehat{X}_{t+1}$. The Mean Squared Error (MSE) is then computed between this prediction and the ground truth state $X_{t+1}$. The total loss is the average of this single-step prediction error across all time steps, all paths, and all sampled parameter sets for that epoch:

$$\mathcal{L}_{\text{MSE}} = \frac{1}{\ell \times P \times T} \sum_{l=1}^{\ell} \sum_{p=1}^{P} \sum_{t=0}^{T-1} \|\widehat{X}_{t+1}(p,l) - X_{t+1}(p,l)\|_2^2 \tag{63}$$

This stepwise training forces the *Meta*-SDENO to learn the intricate relationship between the SDE parameters and the instantaneous evolution of the system, making it an effective and accurate forward model for the iterative update steps of the Ensemble Kalman Filter.

**Ensemble Kalman Filter and Parameter Estimation Process**   After pretraining, we freeze the *Meta*-SDENO and treat it as a forward operator for the parameter estimation, which is performed using an Ensemble Kalman Filter (EnKF). For this process, we first generate a single ground-truth trajectory, $X_t$, by simulating the OU process with its true underlying parameters $(\theta^\star = 4, \mu^\star = 1, \sigma^\star = 0.05)$. This trajectory serves as the sequence of observations for the filter.

The EnKF begins with an initial ensemble of $\ell = 250$ parameter candidates, $\boldsymbol{\Gamma}_0 \in \mathbb{R}^{3 \times \ell}$, where each column is a random sample drawn from the wide prior distributions ($U(0,5)$ for $\theta$, etc.). At each time step $t$, the EnKF performs an update cycle:

1. **Prediction Step**: Each parameter candidate $\boldsymbol{\gamma}_i \in \boldsymbol{\Gamma}_t$ is passed through the frozen *Meta*-SDENO to produce a one-step-ahead prediction of the state, forming the prediction ensemble $\widehat{\mathbf{X}}_t \in \mathbb{R}^{1 \times \ell}$.

2. **Update Step**: The filter then updates the parameter ensemble by optimally blending the predictions with the ground-truth observation $X_t$.

The update is governed by the following equations. First, the ensemble means ($\bar{\boldsymbol{\Gamma}}_t, \bar{x}_t$) and anomaly matrices ($\mathbf{A}_z, \mathbf{A}_x$) are computed:

$$\bar{\boldsymbol{\Gamma}}_t := \tfrac{1}{\ell} \boldsymbol{\Gamma}_t \mathbf{1}, \quad \bar{x}_t := \tfrac{1}{\ell} \widehat{\mathbf{X}}_t \mathbf{1} \tag{64}$$

$$\mathbf{A}_z := \frac{\boldsymbol{\Gamma}_t - \bar{\boldsymbol{\Gamma}}_t \mathbf{1}^\top}{\sqrt{\ell - 1}}, \quad \mathbf{A}_x := \frac{\widehat{\mathbf{X}}_t - \bar{x}_t \mathbf{1}^\top}{\sqrt{\ell - 1}} \tag{65}$$

The Kalman gain $\mathbf{K}$ is calculated using an observation noise variance $r_t$ (a small, constant hyperparameter as a diagonal matrix):

$$\mathbf{K} = \left(\mathbf{A}_z \mathbf{A}_x^\top\right) \left(\mathbf{A}_x \mathbf{A}_x^\top + r_t\right)^{-1}$$

Finally, letting $Y_t := X_t \mathbf{1}^\top$ be the replicated observation, the parameter ensemble is updated to $\boldsymbol{\Gamma}_{t+1}$:

$$\boldsymbol{\Gamma}_{t+1} = \boldsymbol{\Gamma}_t + \mathbf{K}\left(Y_t - \widehat{\mathbf{X}}_t\right)$$

This process is repeated for $T = 300$ steps. The final parameter estimate is reported as the mean of the ensemble $\boldsymbol{\Gamma}_{300}$ over the last five time steps. As shown in the main page Fig. 7, this coupling of our *Meta*-SDENO with the EnKF successfully recovers the ground-truth parameters.

**Theoretical Discussion: Mean-Field View and Ensemble Collapse**   In this section, we are more interested in linking our results to the mean-field theory and how the sampling space in the Kalman filter collapses during the parameter update.

From a theoretical perspective, our EnKF-based parameter estimation process can be interpreted as a system of $\ell$ interacting particles, $\{\boldsymbol{\gamma}_i(t)\}_{i=1}^{\ell}$, evolving in the parameter space. The interaction is

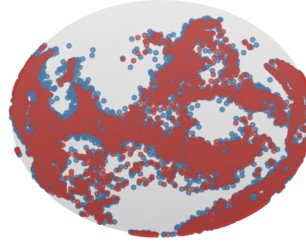

Figure 15: Comparison between ground truth (red) and predicted (blue) flood event.

of a **mean-field** type, as each particle's update depends not on individual other particles, but on the collective statistics of the entire ensemble. The central object of study is the empirical probability measure of the ensemble:

$$\mu_t^\ell = \frac{1}{\ell} \sum_{i=1}^{\ell} \delta_{\boldsymbol{\gamma}_i(t)} \tag{66}$$

where $\delta_{\boldsymbol{\gamma}}$ is the Dirac measure at position $\boldsymbol{\gamma}$. As established in the foundational work on this topic, in the mean-field limit where the number of ensemble members $\ell \to \infty$, this empirical measure converges weakly to a deterministic probability measure $\mu_t$ (Law et al., 2016; Ertel, 2025). The evolution of this limiting measure is described by a non-linear Fokker-Planck type partial differential equation, often referred to as a McKean-Vlasov equation (Chan, 1994).

A key characteristic of this dynamic, which is observable in practice, is the systematic reduction of the ensemble variance, a phenomenon often referred to as **ensemble collapse** or "sample space shrinkage." This is a direct consequence of the Kalman update step. Let $P_t^\ell$ be the sample covariance matrix of the parameter ensemble. The update from the prior at time $t$ to the posterior at time $t+1$ (before the next prediction step) follows the exact formula:

$$P_{zz}^{(t+1)} = P_{zz}^{(t)} - P_{zx}^{(t)}(P_{xx}^{(t)} + r_t I)^{-1}(P_{zx}^{(t)})^T \tag{67}$$

where $P_{zx}^{(t)} = \mathbf{A}_z \mathbf{A}_x^\top$ is the sample cross-covariance between parameters and predictions, and $P_{xx}^{(t)} = \mathbf{A}_x \mathbf{A}_x^\top$ is the sample covariance of the predictions. Since the matrix $(P_{xx}^{(t)} + r_t I)$ is positive definite, the term subtracted is positive semi-definite, which mathematically guarantees that the posterior variance is reduced ($P_{zz}^{(t+1)} \preceq P_{zz}^{(t)}$). This variance reduction reflects the filter gaining information and increasing its confidence about the true parameter values. However, it also highlights a well-known trade-off: with a finite ensemble size, excessively fast variance shrinkage can lead to filter degeneracy. This underscores the importance of choosing a sufficiently large ensemble size, such as $\ell = 250$ in our study, to ensure a robust estimation process.

### F.7 WCE ON MANIFOLD

In this section, we show experimental details on our flood prediction, which employs the SDE on the sphere. Similarly, we pretrain the Riemannian score based generative models (De Bortoli et al., 2022) on the Flood dataset, and sample 200 Riemannian Brownian increments on the manifold tangent bundle.

**Model Training** Our approach to solving the SDE on the sphere $S^2$ involves transforming the problem from the manifold to a 2D Euclidean tangent space, where our established SDENO architecture can be applied. This process consists of data preparation, model training in the tangent space, and evaluation on the manifold.

*Geometric Data Preparation*: The core of our method is to "flatten" the problem. For each sample trajectory, we first establish a reference point $\mathbf{x}_{\text{ref}} \in \mathbb{S}^2$.

1. *Input Processing*: The driving Brownian path on the sphere, which consists of tangent vectors at different points along the trajectory, is mapped to a single, fixed 2D tangent plane at $\mathbf{x}_{\text{ref}}$ using **parallel transport**. This transforms the complex, path-dependent noise into a standard 2D Wiener process increment path $\Delta W \in \mathbb{R}^{N \times 2}$, which serves as the input to our model.

2. *Target Processing*: The ground truth solution path on the sphere is projected onto the same tangent plane using the **logarithmic map**. The resulting 2D vector becomes the prediction target for our model.

We show the comparison between ground truth (RSGM) and our SDENO prediction in Figure 15.

**Limitations and Future Work.**   While SDENO provides a unified way to encode stochasticity for a broad class of SDEs and SPDEs, our current study has several limitations.

**Observing Brownian Increments.** First, our formulation assumes access to the driving noise trajectories (or their increments) so that Wiener–chaos features can be constructed explicitly. This is natural in synthetic benchmarks and controlled simulation settings, but in many real-world applications only noisy observations of the solution $X_t$ are available and the underlying noise path $W_t$ must be inferred. Extending SDENO to settings with latent or partially observed noise—for example by combining our chaos-based representation with likelihood-based SDE inference, score-based models, or stochastic filtering techniques—is an important direction for future work.

**Truncation and Complexity.** Second, the temporal and chaos expansions used in SDENO and $\mathcal{F}$-SPDENO are truncated to a finite number of modes. Although this is analogous to classical spectral methods and we find that modest truncation levels are sufficient for the temporal resolutions and regular SPDEs considered in our experiments, highly irregular dynamics or very long time horizons may require richer bases and higher truncation orders, which would increase memory and computation. Developing adaptive or data-driven strategies for selecting chaos and temporal modes, or multi-resolution variants of our framework, is a promising avenue to improve scalability.

**Manifold Constraints.** Third, our manifold SDE experiment on the sphere $\mathbb{S}^2$ relies on a practical simplification: we linearize the problem by parallel-transporting the entire stochastic process to a single fixed tangent plane and constructing Wick features there. This "global flattening" works well on the sphere, a manifold with constant positive curvature, but its generalization to more complex Riemannian manifolds is non-trivial. On manifolds with varying curvature or complicated topology, a single global reference frame may not exist, and parallel transport can be path-dependent and introduce distortion, potentially degrading the statistical properties of the transported Brownian motion. Intrinsic manifold-native constructions, such as Wiener–chaos expansions built directly from the isometry group on specific spaces (Kalpinelli et al., 2013), avoid these issues but require sophisticated, manifold-dependent machinery. Extending SDENO to such intrinsic constructions, or designing robust local/patch-wise linearization schemes on general manifolds, is left for future work.

**Singular SPDEs.** Finally, our empirical evaluation focuses on non-singular SPDEs (e.g., dynamic $\Phi_1^4$ in 1D and stochastic Navier–Stokes at moderate resolutions) and several downstream SDE-based tasks (image diffusion, brain graphs, and manifold flows). We do not yet evaluate SDENO on truly singular SPDEs that require renormalisation, where regularity-structure-based neural operators (Hu et al., 2022; Gong et al., 2023) are particularly well suited. A natural next step is to combine our chaos-based noise representation with such advanced architectures and to benchmark SDENO on more challenging singular SPDE datasets, for example those in recent SPDE benchmarks.

