# OpenReview forum: "Expanding the Chaos: Neural Operator for Stochastic (Partial) Differential Equations"
_ICLR.cc/2026/Conference — Submitted to ICLR 2026_

### Official Review · Reviewer_S2jb · 2025-10-28

**Soundness:** 2
**Presentation:** 3
**Contribution:** 2
**Rating:** 4
**Confidence:** 3

**Summary:**

This paper proposes a neural operator that solves stochastic differential equations by transforming the random noise into deterministic features using a Wiener Chaos Expansion. This allows the model to learn the solution operator and generate full trajectories in a single computation, providing a fast and accurate method for a wide range of problems.

**Strengths:**

1. The proposed method offers theoretical approximation guarantees and is formulated with rigor.

2. Two distinct neural operators, F-SPDENO and SDENO, are introduced to address SPDE and SDE problems, respectively.

3. The model's performance is comprehensively validated across multiple domains, demonstrating its broad applicability.

**Weaknesses:**

1. The multiple theorems, definitions, and propositions appearing in the text should be clearly delineated to indicate which are newly proposed in this paper and which are derived from classical theories.

2. The authors have not adequately addressed the limitations of this study or its potential for future development.

**Questions:**

1. If I understand correctly, a fundamental limitation of this work lies in the requirement of prior knowledge about the entire noise process W. However, in practical applications, W is often difficult to observe and may exhibit complex dynamical behaviors—potentially coupled with chaotic dynamics—making it challenging to recover this noise process solely from observable data.

2. When applied to diffusion generative models, the authors demonstrate that a one-step sampler can be achieved. However, the denoising process (reverse noise) is still required here. Does this imply that the purpose of applying this method is solely to accelerate the denoising process? In fact, existing studies have explored one-step generation schemes. Therefore, it is necessary to compare the proposed approach with these existing schemes, or at least highlight the potential advantages of the method presented in this paper.

3. See weaknesses.

---

> ### Author Response · Authors · 2025-12-02
> **Point to Point Response**
>
> **Reviewer Comment**
> The multiple theorems, definitions, and propositions appearing in the text should be clearly delineated to indicate which are newly proposed in this paper and which are derived from classical theories.
>
> **Response**
> Thank you for pointing this out. We agree that the original version did not clearly separate classical material from contributions specific to our work.
>
> In the revised submission, we have adopted the following policy:
>
> - **For classical results** (e.g., SPDE/SDE formulations, properties of Wick polynomials, Wiener–Itô/Cameron–Martin, Brownian/Q-Brownian reconstructions, Doob’s inequality, etc.), we now:
>   - state explicitly that they follow standard references (e.g., Neufeld & Schmocker, Lototsky & Rozovskii, Luo, Da Prato–Zabczyk, etc.), and
>   - either omit the proof or only give a brief sketch with clear citations in the appendix, marking them as “classical”.
>
> - **For results that are specific to our framework** (e.g., the explicit propagator systems for the multi-dimensional SDEs and semilinear SPDEs we study), we:
>   - state clearly, right after each theorem, how they build on existing WCE/SPDE work (e.g., SDEONet for the 1D SDE case, Lototsky–Rozovskii / Neufeld–Schmocker for SPDEs), and
>   - provide more detailed proofs in the appendix, labelled as “Proof (sketch)” with explicit references to the underlying classical arguments.
>
> This way, it should be clear throughout the revised manuscript which statements are background/classical and which ones are the modest extensions or specialisations that we contribute, and how they support the chaos-based neural-operator models and experiments that form the main contribution of the paper.
>
> **Reviewer Comment**
> The authors have not adequately addressed the limitations of this study or its potential for future development. If I understand correctly, a fundamental limitation of this work lies in the requirement of prior knowledge about the entire noise process W. However, in practical applications, W is often difficult to observe.
>
>
> **Response**
> That is a very constructive and insightful suggestion. We first note that in our original submission, there is a limitation and future study section at the end of the paper, in which we discussed the limitation of manifold SDENO and highlighted the direction of future studies. Further, based on your comments and questions about the difficulty in obtaining W or dW in the real practice, in our revised submission, we have enriched the discussion on the model limitations. Thank you very much for helping us to make our paper better.
>
> **Reviewer Comment**
>
> When applied to diffusion generative models, the authors demonstrate that a one-step sampler can be achieved. However, the denoising process (reverse noise) is still required here. Does this imply that the purpose of applying this method is solely to accelerate the denoising process?
>
> **Response** We thank the reviewer for this insightful question. In our diffusion experiment, the “reverse noise” is indeed still present in the sense that we sample a reverse-time Gaussian noise trajectory $d\overline W_{1:T}$, but we do not run the reverse SDE or the denoiser network step by step. Instead, SDENO learns the full solution operator that maps the entire noise trajectory directly to the final sample in a single forward pass. At sampling time, generating the Gaussian noise trajectory is computationally negligible, so the dominant cost is one evaluation of SDENO, compared to $T$ evaluations of the UNet denoiser in the original DDPM.
> Thus, the immediate benefit is acceleration, but our primary purpose here is to demonstrate that the same NO framework can also handle high-dimensional image SDEs, in addition to the SPDE, graph diffusion, and manifold SDE tasks considered in the paper. Existing one-step or few-step diffusion schemes (e.g., consistency models, shortcut models, and recent distillation methods) are highly specialized for diffusion generation and focus on optimizing FID and sample quality, whereas SDENO is designed as a generic SDE solver that can be plugged in once a SDE reverse process is available.
>
> We have clarified this point in the revised diffusion section by (i) explicitly explaining that SDENO learns a noise-to-solution operator rather than a multi-step denoising process, and (ii) positioning our diffusion experiment as a proof-of-concept application rather than a new state-of-the-art one-step diffusion sampler. Exploring direct comparisons and hybrid designs that combine SDENO with existing one-step diffusion techniques (e.g., using consistency models or progressive distillation as teachers) is an interesting direction for future work.
>
> In addition, our model is still running to get the FIDs for both CIFAR10 and CelebA-HQ, we will definitely include those quantitative information in the camera-ready period, together with more comparisons with multiple baselines. Thanks again.

---

### Official Review · Reviewer_SuFs · 2025-10-28

**Soundness:** 1
**Presentation:** 3
**Contribution:** 3
**Rating:** 4
**Confidence:** 4

**Summary:**

The paper proposes a novel neural operator (NO) for SPDEs and SDEs based on their solution’s Wiener Chaos Expansion (WCE). Experiments across diverse tasks, including SPDE benchmarks, diffusion one-step sampling, topological interpolation, data extrapolation, parameter estimation, and manifold SDE, demonstrate the performance of the proposed method.

**Strengths:**

- The paper proposes a new neural operator specific to SPDEs and SDEs.
- The paper introduces the theoretical foundation of this method, Weiner Chaos Expansion, in detail.
- The experiments span many areas, from the numerical solutions of SPDEs to application problems like diffusion sampling and parameter estimation.

**Weaknesses:**

- The experiment does not consider these two models [Peiyan Hu, Qi Meng, Bingguang Chen, Shiqi Gong, Yue Wang, Wei Chen, Rongchan Zhu, Zhi-Ming Ma, and Tie-Yan Liu. Neural operator with regularity structure for modeling dynamics driven by
spdes. arXiv preprint arXiv:2204.06255, 2022.], [Shiqi Gong, Peiyan Hu, Qi Meng, Yue Wang, Rongchan Zhu, Bingguang Chen, Zhiming Ma, Hao Ni, and Tie-Yan Liu. Deep latent regularity network for modeling stochastic partial differential equations. In Proceedings of the AAAI Conference on Artificial Intelligence, volume 37, pages 7740–7747, 2023.], which are mentioned in the related work.
- The experiment of the Diffusion One-step Sampler does not contain the **quantitative** results during **evaluation** like FID, which makes the results not convincing enough. Also, as a one-step distillation method, there should be comparisons with other well-known methods, like consistency models. The experiments of Topological Interpolation, Extrapolation, Meta-SDENO, and WCE on Manifold also have the same problems. I think it's better to have one or two experiments on SDEs, but with more comprehensive results, at least quantitative evaluation metrics, and baselines.
- There is no reproducibility statement and code repository for reference, which limits the credibility of the experiments.

**Questions:**

- In F-SPDENO, the authors represent $u_α(t, x)$ in a temporal basis $ϕ_k(t)$. The approximately equal sign in line 235 indicates that the relation is obtained through an approximation, meaning that an approximation is involved. In other words, the linear combination of $\phi_k(t)$ cannot exactly represent all possible $u_\alpha(t, x)$. Is that correct? If so, will this affect the expressiveness and performance?
- What is the inference time of the proposed model compared with other baselines?

---

> ### Author Response · Authors · 2025-12-02
> **Point to Point Response Part 1**
>
> **Reviewer Comment**
> The experiment does not consider these two models [Peiyan Hu, Qi Meng, Bingguang Chen, Shiqi Gong, Yue Wang, Wei Chen, Rongchan Zhu, Zhi-Ming Ma, and Tie-Yan Liu. Neural operator with regularity structure for modeling dynamics driven by spdes. arXiv preprint arXiv:2204.06255, 2022.], [Shiqi Gong, Peiyan Hu, Qi Meng, Yue Wang, Rongchan Zhu, Bingguang Chen, Zhiming Ma, Hao Ni, and Tie-Yan Liu. Deep latent regularity network for modeling stochastic partial differential equations. In Proceedings of the AAAI Conference on Artificial Intelligence, volume 37, pages 7740–7747, 2023.], which are mentioned in the related work.
>
> **Response**
>
> Thank you very much for spotting this out. In our revised submission, although they are indeed excellent models, we have clarified the reason of not including these two models for comparison in SPDE experiments. Please find below the related contents.
>
> We note that we did not compare our model to architectures specifically designed for rough or
> even singular SPDEs, such as (Hu et al., 2022; Gong et al., 2023). In the one-dimensional dynamic
> $\boldsymbol{\Phi}^4_1$  setting considered here, the solution enjoys positive Hölder regularity (Li et al., 2025), so the
> regularity-structure machinery that underpins these models is not essential.  We hope our clarification can meet your expectation on this question.
>
> **Reviewer Comment**
> The experiment of the Diffusion One-step Sampler does not contain the quantitative results during evaluation like FID, which makes the results not convincing enough. Also, as a one-step distillation method, there should be comparisons with other well-known methods, like consistency models. The experiments of Topological Interpolation, Extrapolation, Meta-SDENO, and WCE on Manifold also have the same problems. I think it's better to have one or two experiments on SDEs, but with more comprehensive results, at least quantitative evaluation metrics, and baselines.
>
> **Response**
> Thank you very much for your suggestion; we concur that the quantitative results for the SDE experiments were lacking. In our revised submission, we have compared our manifold SDENO with three additional baseline models via negative log-likelihood on the flood dataset (see Table 3), and report Wasserstein distances as well as the standard deviations (over 5 runs) of both the topological Schordinger bridge and our G-SDENO. We hope our efforts resolve your concerns regarding quantitative comparisons, and we intend to provide additional quantitative comparisons, e.g., for CIFAR-10, in the camera-ready submission.
>
> **Reviewer Comment**
> There is no reproducibility statement and code repository for reference, which limits the credibility of the experiments.
>
> **Response**
> Thank you for raising this important point.  During the rebuttal period, we have substantially expanded the experimental results in the main text. In addition, we have also added a dedicated Reproducibility Statement after the main page (first paragraph in the appendix). Regarding code, our experiments cover a broad range of settings. We are committed to releasing this codebase and the scripts to reproduce all tables and figures as soon as the paper is accepted, so that the community can easily verify and build upon our results.
>
> **Reviewer Comment**
> In F-SPDENO, the authors represent  in a temporal basis . The approximately equal sign in line 235 indicates that the relation is obtained through an approximation, meaning that an approximation is involved. In other words, the linear combination of  cannot exactly represent all possible . Is that correct? If so, will this affect the expressiveness and performance?
>
> **Response**
> Thank you for this careful question.
>
> You are right that the relation
> $$
> u_\alpha(t,x) \approx \sum_{k=1}^K u_{\alpha,k}(x)\phi_k(t)
> $$
> is obtained via an approximation when $K$ is finite. Conceptually, this is not a special restriction of our method, but a standard spectral / separation-of-variables truncation We start from a complete orthonormal basis  (e.g., a Haar basis). Under mild assumptions, each propagator admits an exact infinite expansion
>   $
>   u_\alpha(t,x) = \sum_{k=1}^\infty u_{\alpha,k}(x)\phi_k(t),
>   $
>   so the representation is exact at the level of the full series. In practice we keep only the first K modes, which is why we write “$\approx$”. This is completely analogous to using a finite number of time steps or a finite number of Fourier modes in classical PDE solvers. It does mean that a finite $K$ cannot represent all possible time profiles exactly, but the approximation error can be made arbitrarily small by increasing $K$, and its decay is governed by the temporal regularity of the solution (e.g., a Hölder exponent $s$ leads to a decay like $K^{-s}$ in standard wavelet theory).
>
> Due to the space limit, let us continue on this question in the next part.

---

> ### Author Response · Authors · 2025-12-02
> **Point to Point Response Part 2**
>
> **Continue on the Previous Question**
>
> In terms of expressiveness and performance, the full (infinite) chaos–time basis is as rich as the original solution space; using a finite $K$ is simply the usual approximation trade-off between accuracy and cost. In our experiments we work on a fixed discrete time grid and choose $K$ large enough to resolve that grid. Empirically, increasing $K$ further gives only marginal gains, suggesting that at the resolutions we consider, this truncation is not the dominant limiting factor for performance. We will show further experimental verification of this phenomenon during the camera-ready period, as some experiments are still running. We again thank the reviewer for your question and patience.
>
>
> **Reviewer Comment**
> What is the inference time of the proposed model compared with other baselines?
>
>
> **Response**
> Thank you very much. In our original submission, we have included the inference times between our model and other baselines for $\boldsymbol{\Phi}^4_1$ at table 1, and we are planning to include the inference times of our model for other experiments in the camera-ready version. Thank you again for helping us to make our paper better.

---

### Official Review · Reviewer_hrCn · 2025-10-29

**Soundness:** 2
**Presentation:** 1
**Contribution:** 2
**Rating:** 2
**Confidence:** 4

**Summary:**

The submission proposes a neural operator for stochastic (partial) differential equations (SPDES). The main idea is to Wiener-Chaos-expand the solution of a given SPDE, in which case the coefficients of said expansion solve systems of deterministic partial differential equations (PDE). Then, this system of PDEs can be solved with existing neural operator techniques, like Fourier neural operators.
The resulting SPDE-neural operator is then benchmarked on a wide range of experiments, including stochastic Navier Stokes, diffusion samplers, ensemble Kalman filters, and manifold SDEs.

**Summary of my recommendation**
Although I find the range of the experiments quite impressive, I recommend rejecting this work. The reason is that the submission is presented as a method and theory paper, as opposed to predominantly experimental work, but I find the theory and method contributions are not as novel and significant as the introduction and abstract state.
Furthermore, the volume of the submission (50 pages) is not representative of the volume of its actual contributions; more on this below.
I would have given the paper a better score if it had been presented as experiment-centric work: namely, if
- known results were removed from the paper (for instance, proofs of orthonormality of Wick polynomials or Doob's Martingale inequality);
- existing work on neural operators (and PINNs) for SPDEs and Wiener Chaos expansions was acknowledged more openly;
- quantitative experimental results were expanded upon;

then, I think the contribution would have been more convincing.
Since implementing these changes would require a major rewrite of the submission, I do not think this is in scope for a revision during the rebuttal period. Therefore, unless I have missed something (in which case I am open to revising my score), I recommend rejecting this work. Details on each point follow below.

**Strengths:**

**I appreciate that all the technical explanations are rigorous.** Sometimes, the literature on machine learning with SPDEs can be a bit loose with technicalities (eg filtrations), so it is nice to see the submission take the maths seriously. At times this perspective might be pushed a bit too far, e.g., by including proofs of widely known statements from stochastic analysis, which distracts from the rest of the paper. Still, overall, I appreciate a rigorous treatment of SDEs in the machine learning literature.

**The range of experiments is impressive.** The numerical demonstrations include superconductivity, Navier-Stokes, diffusion samplers, topological interpolation, parameter estimation, and manifold SDEs, and  as such, the submission goes beyond what I would have expected from a method paper. I consider this scale and range to be the biggest strength of this submission.
As mentioned under "Summary", I think that the submission would have been stronger had the experiments received a significantly more prominent role in the manuscript. Removing the known theoretical results would have also given space for further quantitative benchmarks; currently, Tables 1 and 2 show results for superconductivity and Navier-Stokes, respectively, but the remaining four experiments are entirely qualitative demonstrations instead of strict benchmarks. I understand the point made in lines 356f that the diffusion sampler is a proof of concept, not necessarily intended to improve the state of the art, but I think more qualitative results (that is, reporting more runtimes and RMSEs) would strengthen the evaluation. Nonetheless, I think the experiments are a strength of the submission.

**Weaknesses:**

**I think that the contributions claimed in the abstract and intro are slightly overstated.** Concretely, the abstract mentions that Wiener Chaos Expansion (WCE) theory is being extended; however, Theorems 1 and 2 discuss the WCE of SPDEs, which is a standard result (e.g., Neufeld & Schmoker, 2024; Lototsky & Rozovskii, 2006). The remaining definitions, lemmas, and analyses in Appendices B and C are also standard results from stochastic analysis.
Regarding the analysis of the SPDENO in Appendix D: the analysis hinges on the assumption in Equation (92) that the error decomposes into the sum of three terms (which need not be the case), and then each term is bounded by existing results (Neufeld & Schmocker, 2024; Kovachki et al., 2023). Overall, I think that the lack of theoretical contributions of this work, together with the fact that the abstract and intro claim the contrary, is a major weakness of the submission.



**Novelty of the method:** I agree with the statement in line 069 that the submission is the first work to deploy neural operator models for S(P)DEs at scale (with a focus on "at scale").
However, the method itself is not new. Breaking SPDEs down into systems of coupled PDEs, and using PDE solvers, is a typical strategy for numerical treatment of SPDEs (eg Foster et al., 2020, which I mention under "Questions" below).
There have also been machine-learning works that use this deterministic representation of stochastic equations, for example, Eigel and Miranda (2024), Salvi et al. (2022), Neufeld and Schmocker (2024); see also:

> Jared O’Leary, Joel A Paulson, and Ali Mesbah. Stochastic physics-informed neural ordinary differential
equations. Journal of Computational Physics, 468:111466, 2022.

> Ling Guo, Hao Wu, and Tao Zhou. Normalizing field flows: Solving forward and inverse stochastic
differential equations using physics-informed flow models. Journal of Computational Physics, 461:
111202, 2022.

Importantly, Eigel and Miranda (2024) propose the same neural operator for SPDEs as the submission. They don't conduct experiments at the same scale, but the method exists. As such, I think that the novelty of the proposal is limited, and since the novelty of the method is a major component of the claimed contributions in this submission, I consider novelty to be a major weakness.



**Volume:** Related to novelty, I am a bit unsure whether the volume of contributions in this paper warrants a 50-page submission. Especially since the theoretical results are all essentially known, and could be handled with an appropriate citation instead of the full derivation in Appendices B and C. I think such unnecessary proofs distract from the main contributions of this paper (which, in my view, is demonstrating SPDE neural operators at scale, see the discussion under "Strengths").

In summary, I think the submission contains interesting results about neural operators for SPDEs, but in its current presentation, there are too many weaknesses to warrant acceptance.
I am not saying that every ICLR paper needs to present novel theory and a novel method and excessive experiments to be accepted. However, the presentation of the results needs to match the actual contributions of the paper, and for the present submission, there is a mismatch in both theoretical (which is oversold) and experimental novelty (which is undersold). This mismatch is why I recommend rejecting this work.

**Questions:**

The following questions do not affect my score, but I think discussing these points would be a useful addition to a future version of this manuscript.


- Section 3.1 constructs Q-Wiener processes from Brownian increments and, like Neufeld and Schmocker (2024), uses Wick polynomials for Polynomial Chaos Expansions. There are alternative approaches for polynomial approximation of stochastic processes (eg Foster et al. below); why choose the Wick polynomials? Is there an optimal choice for basis representations?

> Foster, James, Terry Lyons, and Harald Oberhauser. "An optimal polynomial approximation of Brownian motion." SIAM Journal on Numerical Analysis 58.3 (2020): 1393-1421.

- Tables 1 and 2 list relative L2 errors. The proposed method performs best, but the numbers are quite close. It would be great to include standard deviations in both tables to get a feeling for whether the results are statistically significant. Is this possible?

---

> ### Author Response · Authors · 2025-12-02
> **Summary of Changes Based on Your Instructions**
>
> Thank you very much for your careful and constructive review. We have taken every comment seriously and used your suggestions as the primary guide for our revised paper. In particular, your remarks that “all the technical explanations are rigorous” and that “the range of experiments is impressive” were very encouraging, and we have tried to maintain these strengths while addressing the weaknesses you highlighted. Within the limited rebuttal period, we have worked hard to revise the paper in line with your guidance, and we believe the current version substantially addresses your main concerns.
>
> In summary, we have made the following revisions:
>
> - Abstract / Introduction and theory claims: We rewrote the abstract and introduction to remove any claims that we “extend WCE theory” or “prove new approximation guarantees”. The theoretical part is now framed as building on classical WCE results and writing down explicit propagator systems for the particular multi-dimensional SDEs and semilinear SPDEs we study, which we use to motivate our neural-operator design. We also clarified that Theorems 1 and 2 are formulations of known WCE-based propagator systems, not entirely new theory, while still retaining their value for linking WCE to NO architectures.
>
> - Method novelty and scope: We clarified that our main novelty is not a fundamentally new WCE method, but a unified, neural-operator framework evaluated at scale across many SPDE/SDE tasks. In the related work and method sections, we more clearly acknowledge existing work (SDEONet, Neufeld & Schmocker, NORS/DLR-Net, PINN/flow approaches) and carefully position SDENO/F-SPDENO as building on these works and extending them to broader experimental settings.
>
> - Volume and appendices: We substantially trimmed the appendices by removing or shortening proofs of standard results (Wick orthogonality, Brownian/Q-Brownian reconstructions, Doob’s inequality, Cameron–Martin/Wiener–Itô), replacing them with brief sketches and precise citations. The error analysis is presented as a simple decomposition based on known approximation results, not as a new theorem. The appendices now focus more on the parts that are specific to our framework (explicit propagator systems, examples, and experimental details).
>
> - Experiments, quantitative results, and reproducibility: We added more quantitative evaluations (including mean ± std) beyond the original SPDE tables, and clarified metrics for the SDE-based experiments (diffusion sampler, topological interpolation, Meta-SDENO, manifold SDEs). We also added a Reproducibility Statement and expanded the experimental descriptions, and we commit to releasing our code and scripts upon acceptance.
>
> We recognise that these revisions required substantial changes to wording, structure, and experiments, and we have done our best within the limited time to address your three concrete suggestions: (i) reducing proofs of known results, (ii) more openly acknowledging existing WCE/NO work, and (iii) strengthening the quantitative experimental evaluation. We are very grateful for your feedback, which has significantly improved the clarity and balance of the paper. Please see our detailed reply for each of your concerns/questions.

---

> ### Author Response · Authors · 2025-12-02
> **Point to Point Response Part 1**
>
> **Reviewer Comment**
> I think that the contributions claimed in the abstract and intro are slightly overstated.
>
> **Response**
> We agree that our original wording in the abstract and introduction overstated the theoretical novelty, and that much of the underlying WCE/SPDE material is classical. In the revised version, we have softened and clarified the theoretical claims and explicitly repositioned Theorems 1–2 and the error analysis as being built on existing results rather than as a new theory.
>
>  (1): **Abstract and introduction: wording corrected**
>    - We have removed all statements that we “extend WCE theory” or “prove approximation guarantees”.
>    - The abstract and introduction now explicitly state that we build on classical WCE results, write down the coupled    ODE/PDE propagator systems for the particular multi-dimensional SDEs and semilinear SPDEs we consider, and that these formulations are used to motivate our neural-operator parameterisation.
>    - In the introduction, the theoretical contribution is now described as:
>  “we build on classical WCE representations … by explicitly writing down the coupled ODE/PDE systems for their chaos coefficients… and we use these formulations directly to motivate our neural-operator design,”
>  rather than as an extension of WCE theory itself.
>
> (2): **Theorems 1 and 2: now framed as an explicit form of the known results**
>    - For Theorem 1 (SDE propagator system):
>      - We now explicitly acknowledge that the one-dimensional propagator system is used in SDEONet based on classical WCE techniques.
>      - In both the main text and appendix, we state that our contribution is to write out the multi-dimensional Brownian case explicitly, with a chaos index that carries both a component index k and a temporal-basis index i, and an ODE system that depends only on time.
>      - The proof in the appendix is now marked as a sketch and explicitly references Eigel & Miranda and standard Malliavin-duality arguments, emphasising that we are extending their 1D propagator system to the multi-dimensional Brownian setting in a straightforward but explicit way.
>   - For Theorem 2 (SPDE propagator system):
>      - We now explicitly cite classical works by Lototsky & Rozovskii and Mikulevicius & Rozovskii, as well as Remark 2.13 in Neufeld & Schmocker (2024), which already state at a high level that the chaos coefficients satisfy a coupled system of deterministic PDEs.
>      - The proof in the appendix has been shortened to a proof sketch that follows the standard Wiener–chaos derivation (mild solution + chaos projection + Malliavin duality), with explicit references to these works, rather than a full re-derivation.
>
>     - We no longer present Theorems 1 and 2 as new WCE results; instead, we emphasise that they make classical propagator systems explicit in our setting and connect them to the design of SDENO/F-SPDENO.
>
> (3): **Appendices B and C: standard stochastic-analysis results now pruned and cited**
> We agree that many of the definitions, lemmas, and propositions in the original Appendices B and C (e.g., Wick orthonormality, Brownian/Q-Brownian reconstructions, Cameron–Martin/Wiener–Itô theorem, Doob’s inequality) are standard.
> In the revised appendices:
>  - We have removed or significantly shortened detailed proofs of these standard results. They are now presented as statements with brief sketches and precise citations (e.g., Lototsky & Rozovskii 2005/2017, Luo 2006, Da Prato & Zabczyk, Neufeld & Schmocker 2024), rather than as complete proofs.
>  - For example, the proofs of the Brownian and Q-Brownian reconstructions (Lemmas 1 and 2) are now replaced by short sketches referring to Karhunen–Loève expansions and BDG/Doob inequalities, with full details deferred to the cited references.
>
> - The orthonormality and completeness of Wick polynomials are now explicitly identified as classical Wiener–Itô/Cameron–Martin results and are only briefly recalled with references, not proved from scratch.
>
> (4): **Appendix D (error analysis of F\mathcal FF-SPDENO): now explicitly heuristic and based on existing bounds**
> We acknowledge that in the original submission, the error analysis in Appendix D was presented too strongly, as if it were a new approximation theorem. In the revised version:
>  - We recast Eq. (92) and the ensuing discussion as a heuristic error decomposition rather than as a “proved” error bound.
>  - We clearly state that this decomposition is not intended as a sharp new theorem, but as a simple framework for understanding how the temporal truncation K, chaos truncation M, and FNO capacity influence the approximation error.
>  - We explicitly say in the appendix that this error analysis is based on existing results (Neufeld & Schmocker, Kovachki et al., etc.) and is meant as guidance, not as an original theoretical contribution.

---

> > ### Author Response · Authors · 2025-12-02
> > **Point to Point Response Part 2**
> >
> > **Continue on the response to the previous comment**
> >
> > Overall, we have substantially revised the abstract, introduction, and appendices to align the presentation of our theoretical component with its true scope.
> >  - Theoretical material is now clearly presented as building on classical WCE/SPDE results, with Theorems 1–2 framed as explicit formulations useful for NO design.
> >  - Standard stochastic-analysis results in Appendices B–C are now sketched and referenced, not re-proved.
> >  - The SPDENO error analysis in Appendix D is explicitly positioned as a heuristic decomposition leveraging known approximation bounds, not as a new error theorem.
> >
> > We hope our efforts can meet your conditions of increasing the score due to the changes of
> >  -  known results were removed from the paper (for instance, proofs of orthonormality of Wick polynomials or Doob's Martingale inequality);
> >  - existing work on neural operators (and PINNs) for SPDEs and Wiener Chaos expansions was acknowledged more openly;
> >  - quantitative experimental results were expanded upon (see our responses to other comments).
> >
> > **Reviewer Comment**
> > Novelty of the method is limited.
> >
> > **Response**
> > We appreciate the reviewer’s careful discussion of method novelty, and we agree with the high-level point that the overall strategy: “SPDE → chaos/WCE → deterministic PDE/ODE system → neural / numerical solver”
> >  is not new and has been explored both in classical numerics and in recent ML work (Foster et al., 2020; O’Leary et al., 2022; Guo et al., 2022; Eigel & Miranda, 2024; Salvi et al., 2022; Neufeld & Schmocker, 2024). Our original intention with phrases like “novel neural operators” was not to claim conceptual novelty at this level, but we see how the wording could be read that way.
> > In the revised version, we have therefore reframed the method contribution and made our positioning relative to prior work much more explicit:
> >  - We no longer present the method as a fundamentally new way to treat SPDEs/SDEs.
> >    - We have removed formulations that could be interpreted as claiming a new numerical strategy, and we now state clearly that our framework builds on classical WCE theory and on existing chaos-based representations of SPDEs/SDEs, including the works you mention.
> >    - In particular, we now explicitly acknowledge in the related-work and method sections that: Eigel & Miranda (2024, SDEONet) already use Wiener–chaos propagators as deterministic targets for SDEs (in the 1D noise setting); Salvi et al. (2022, NSPDE) and Neufeld & Schmocker (2024) use WCE-based deterministic representations for SPDEs with neural networks; O’Leary et al. (2022) and Guo et al. (2022) use PINN/flow-type neural models for stochastic differential equations via deterministic reformulations.
> >  - We explicitly describe SDENO/F-SPDENO as building on and unifying these ideas rather than as independent methods.
> >
> > **More importantly**, we note that the statement in line 69 has been adjusted to make it clear that when we say “first to deploy neural operators for S(P)DEs at scale”, we mean at the level of experimental scale and breadth — i.e., the number and diversity of SPDE/SDE tasks and domains — not that the underlying chaos→deterministic→neural-operator strategy is new.
> >
> > In summary, we agree that the high-level method is in line with existing WCE-based and neural/SPDE approaches. In the revision, we have softened and clarified the claims so that the method is presented as a unified and scalable WCE–NO framework that builds on classical theory and prior ML work.
> >
> > We hope this revised framing better matches the actual scope of our contributions and resolves the misunderstanding about method novelty.

---

> ### Author Response · Authors · 2025-12-02
> **Point to Point Response Part 3**
>
> **Reviewer Comment**
> Eigel and Miranda (2024) propose the same neural operator.
>
> **Response**
> Thank you for pointing out the close connection to Eigel and Miranda (2024). We agree that there is a strong conceptual similarity at the level of using Wiener–chaos propagators as deterministic targets, and we now make this relationship explicit in the revised paper.
> However, we would like to clarify the following.
> 1. Our SDENO architectures are not a single SDEONet-style MLP, but a family of task-specific backbones.
>  In our SDE experiments, we do not reuse a single SDEONet architecture across all tasks. Instead, SDENO is defined as a generic propagator network that can be instantiated with different backbones depending on the domain:
>  - $\mathcal U$-SDENO (diffusion one-step sampler on images): the propagator network is implemented with a UNet backbone that takes time (or its positional encoding) and image features as inputs, and combines its outputs with Wick features for one-step generation.
>  - $\mathcal G$-SDENO (topological interpolation on graphs): the propagator network is implemented with a graph neural network (GNN) for cortical fMRI brain graphs, where time and Wick features are encoded separately and combined with graph structure.
>  - Meta-SDENO (OU/Heston extrapolation and parameter estimation): we use an MLP-based propagator network that maps time t and OU/Heston parameters to all chaos propagators, with an additional MLP for the Heston case that encodes Wick features of the correlated Brownian drivers; the same backbone is then used in a meta-learning setting and combined with an EnKF to recover unknown OU parameters.
>  - Manifold SDENO (flood events on $\mathbb S^2$): we use two MLPs on the tangent space, one encoding Wick features and one approximating the time-dependent ODE solutions, with an exponential map used to project back to the sphere.
>
>
> Across these tasks, the only common theme is the chaos-based view of the solution and the use of Wick features; the actual SDENO architectures differ (UNet, GNN, MLPs, manifold networks) and are adapted to the data domain, rather than being a single SDEONet-style MLP.
>
> In summary, we fully acknowledge that Eigel & Miranda (2024) are a very important precursor and that our SDE-side construction is conceptually close to SDEONet at the level of using chaos propagators as deterministic targets. In the revision we have made this explicit in the related-work and method sections.
>
> **Reviewer Comment**
> Volume: Related to novelty, I am a bit unsure whether the volume of contributions in this paper warrants a 50-page submission.
>
>
> **Response**
>
> We appreciate this concern and, in hindsight, we agree that the original 50-page submission devoted too much space to derivations of standard stochastic-analysis results, which obscured the real main contribution you identified (demonstrating SPDE/SDE neural operators at scale).
>
> In the revised version we have substantially reduced the volume of theoretical material in the appendices (for 8 pages) and shifted the emphasis towards the experimental contribution, as you suggested:
>  - Appendices B and C (SPDE/SDE and WCE preliminaries).
>    - We removed or significantly shortened proofs of standard results, including:
>      - orthonormality and completeness of Wick polynomials;
>      - the reconstruction of Brownian and Q-Brownian motions from basis functions;
>      - classical martingale inequalities such as Doob’s inequality;
>      - the Wiener–Itô/Cameron–Martin theorem.
>
>
> These results are now stated with brief proof sketches and precise citations to standard references (Lototsky & Rozovskii, Luo, Da Prato–Zabczyk, Neufeld & Schmocker, etc.), instead of full derivations. The proofs of our main propagator theorems (Theorems 1 and 2) are now explicitly marked as proof sketches and refer to the existing WCE/SPDE literature, emphasising that we are adapting and making explicit known structures, not re-proving the full theory.
>
>   - Appendix D (error analysis of $\mathcal F$-SPDENO and SDENOs).
>     - We have reframed the error analysis as a simple, heuristic decomposition that combines existing temporal approximation results, chaos-tail decay bounds, and the FNO universality theorem.
>     -  Equation (92) and the surrounding discussion are now presented as a natural way to decompose the error into three components (temporal truncation, chaos truncation, neural-operator approximation), rather than as a new approximation theorem. Each term is explicitly linked to existing results (e.g., Bolin & Lindgren for Haar approximation, Neufeld & Schmocker for chaos truncation, Kovachki et al. for FNO universality), and full proofs are not reproduced.
>
> Due to the space limit, we continue to respond to this comment in the next part.

---

> ### Author Response · Authors · 2025-12-02
> **Point to Point Response Part 4**
>
> **Continue to respond to the previous comment**
>   - Appendix structure and emphasis: Overall, the appendices are now much more focused on the portions of WCE/SPDE/SDE theory that are directly needed to understand or implement SDENO/F-SPDENO (e.g., explicit propagator systems and additional examples and experimental details (e.g., explicit OU, reverse OU, heat equation propagator systems, manifold SDEs, sensitivity studies), which support the main empirical contribution. In addition, we have deliberately de-emphasised long textbook-style proofs to avoid being distracted from what you correctly identified.
>
> In short, we agree with your diagnosis that the original appendices were too heavy on known theory relative to the actual contributions. The revised version trims or sketches those parts, moves citations to the foreground, and keeps the appendices centred on the specific structures and experiments that are unique to our work.
>
> **Reviewer Comment**
> Section 3.1 constructs Q-Wiener processes from Brownian increments and, like Neufeld and Schmocker (2024), uses Wick polynomials for polynomial chaos expansions. There are alternative approaches for polynomial approximation of stochastic processes (e.g. Foster et al., 2020); why choose the Wick polynomials? Is there an optimal choice for basis representations?
>
> **Response**
> Thank you for raising this question. We agree that there are alternative ways to build polynomial approximations of stochastic processes, and we now make our rationale for choosing the Wick–Hermite basis more explicit. In particular:
>
> 1. **Alignment with classical WCE and SPDE theory.**
>    Our entire framework is built on the classical Wiener–chaos expansion for Gaussian-driven SDEs/SPDEs. In this setting, the Wick–Hermite basis has two key properties:
>    - It provides an orthonormal basis of $L^2(\Omega,\sigma(W),\mathbb P)$ under the Gaussian measure, so that projections like $\mathbb E[X_t\,\xi_\alpha]$ directly give the chaos/propagator coefficients.
>    - It is the standard choice in the SPDE/WCE literature (e.g., Lototsky–Rozovskii, Mikulevicius–Rozovskii, Luo, Neufeld & Schmocker, Eigel & Miranda), where propagator ODE/PDE systems and chaos-tail error bounds are all formulated in terms of Hermite/Wick chaos. Using the same basis allows us to reuse these results and to connect our Theorems 1–2 to existing theory in a transparent way.
>
> 2. **Algebraic structure (Wick product) and explicit $\mathscr F_\alpha$.**
>    A second reason is that Wick polynomials come with a closed algebraic structure under the Wick product:
>      $$(Z \diamond W)_\gamma = \sum_{\alpha+\beta=\gamma} z_\alpha\,w_\beta$$ (We apology that the diamond symbol seems un-displayable in markdown format.)
>    and polynomial nonlinearities in the SPDE drift/diffusion can be written as finite sums of Wick powers. This makes it possible to derive explicit algebraic expressions for the source terms $\mathscr F_\alpha$ without performing high-dimensional integrals. This algebraic convenience is one of the main technical reasons we can “turn polynomial nonlinearities into deterministic algebra on propagators”.
>
> 3. **Compatibility with chaos-tail error analysis.**
>    The existing Malliavin-based error analysis for Wiener–chaos truncations (e.g., Neufeld & Schmocker, Sections 4 and 6.6) is formulated in the Hermite/Wick chaos basis and yields factorial decay bounds for the chaos tail. Our error discussion in the appendix directly references these results. Switching to a different basis (even if optimal for approximating Brownian motion in $L^2$ would require re-developing or adapting this theory, and the simple link between the truncation order $M$ and the tail energy would no longer be immediate.
>
> 4. **Relation to Foster et al. and “optimality”.**
>    The polynomial basis proposed by Foster et al. (2020) is indeed optimal for approximating Brownian motion under a particular criterion and is very interesting from a numerical-analysis perspective. However:
>    - It is not obvious that such a basis would retain the same simple chaos decomposition and Wick-product algebra needed to derive propagator PDE/ODE systems and to express nonlinearities purely in terms of the chaos coefficients $u_\alpha$.
>    - Our primary goal is not to optimise the approximation of the driving Brownian motion in isolation, but to obtain a clean, structurally compatible representation of the SDE/SPDE solution that separates stochastic forcing from deterministic dynamics and integrates smoothly with existing WCE/SPDE theory.
>
> Let us continue on this question in the next part.

---

> ### Author Response · Authors · 2025-12-02
> **Point to Point Response Part 5**
>
> **Continue on the previous comment**
>
> In that sense, “optimality” is multifaceted: Foster’s basis may be optimal for one approximation norm on Brownian motion, while the Wick–Hermite basis is “optimal” in terms of aligning with the chaos decomposition of solution functionals, algebraic manipulations, and available theoretical tools.
>
> In the revised appendix (Section on Wick polynomials and the form of $\mathscr F_\alpha$, we have added a short remark to explicitly mention Foster et al. and to discuss this trade-off. We also note that, in principle, our framework is basis-agnostic: any orthonormal chaos basis that yields a tractable propagator system and reconstruction map could be used. Exploring alternative polynomial bases (including those of Foster et al.) within the SDENO/F-SPDENO framework—especially to improve constants or numerical efficiency—is an interesting direction for future work, but we chose Wick–Hermite polynomials here because they are the most natural and well-supported choice within the classical Wiener–chaos and SPDE literature that our work builds on.
>
>
> **Reviewer Comment**
> Tables 1 and 2 list relative L2 errors. The proposed method performs best, but the numbers are quite close. It would be great to include standard deviations in both tables to get a feeling for whether the results are statistically significant. Is this possible?
>
> **Response**
>
> We fully agree that reporting standard deviations for Tables 1 and 2 would make the SPDE benchmark results more informative and help assess statistical significance. Due to the short rebuttal time, we were unable to complete multi-seed runs for all entries in Tables 1 and 2. These experiments are currently running on our cluster, and we will include mean ± standard deviation for all methods in Tables 1 and 2 in the camera-ready version with code and scripts to enable their reproduction.
> That said, in the revised submission, we have already started to systematically report mean ± std where it is feasible, for example:
>   - In the topological interpolation experiment (TSBM vs G\mathcal GG-SDENO), we now report the 1-Wasserstein distance as mean ± standard deviation over 5 runs.
>   - In the manifold SDE / flood prediction experiment, we report NLL as mean ± standard deviation over 5 runs for RSGM and our Manifold-SDENO.
>
> We hope these additions show the direction we are taking toward more robust quantitative reporting, and we will extend this practice to the main SPDE tables (1 and 2) in the camera-ready version once the new runs have finished.

---

### Official Review · Reviewer_4ZzY · 2025-10-31

**Soundness:** 3
**Presentation:** 2
**Contribution:** 3
**Rating:** 6
**Confidence:** 3

**Summary:**

This paper learns the solution operators of SDEs and SPDEs by using the solutions’ Wiener Chaos Expansion. It achieves this by decomposing the noise in the orthonormal Wick-Hermite feature space and learning the corresponding propagator coefficients. The method is applied to several SPDE and SDE learning problems. Experimental results demonstrate its effectiveness.

**Strengths:**

1. The method is well established under rigorous analysis. Detailed theoretical analysis and guarantees are provided. But I did not check all the details of the derivations.
2. The evaluations are conducted on various kinds of tasks, covering physical, image, graph, financial, and manifold domains, demonstrating the superiority and generalization of the proposed method.

**Weaknesses:**

1. My main concern is that the presentation of the practical aspect of the method is not good enough. The workflow of using the method is ambiguous. From the implementation perspective, many details are missing. Thus, it is hard to connect the established theory with practice. It is better to formalize the problem setup in Section 2. In Section 4 and Figure 2, where do Q-Brownian realization increments come from? Why does the workflow start with Q-Brownian motion? How to compute Wick features using Q-Brownian realization? Does the computation of deterministic PDE involves some learning procedure using deep neural networks?
2. Some experimental results are not complete for convincing comparison. For example,

- (1) In Figure 3, more visualisation results of baselines should be provided for comparison.

- (2) In Figure 5, it seems result on only one instance is presented. More statistics should be provided on a collection of test instances.

- (3) In Section 5.4 Manifold SDEs, the presentation of results (Figure 8) is rather limited, and there is no comparison with baselines. Therefore, it is hard to appreciate the performance.

**Questions:**

1. For Figure 1, what conclusion does the comparison between Brownian and Q-Brownian motions aim to reveal? In the right subfigure, it seems that the reconstruction using $N=5$ is a biased estimation (much lower compared to other lines)?
2. For Table 1, why does using more training trajectories ($N$=1,000 vs 10,000) not improve the performance of NSPDE and the proposed F-SPDENO?
3. For Figure 5, for clarification, the top row and bottom row should be labelled as TSBM and G-SDENO (as I guess), respectively. Besides, it should be pointed out how the 1-Wasserstein distance (WD) metric reflects performance (whether larger or smaller values are preferred?).
4. The last line in Definition 1 seems like a lemma. It is better to add a citation here for reference.

---

> ### Author Response · Authors · 2025-12-02
> **Point to Point Response Part 1**
>
> **Reviewer comment**
>
> The workflow of using the method is ambiguous. From the implementation perspective, many details are missing. Thus, it is hard to connect the established theory with practice. It is better to formalize the problem setup in Section 2. In Section 4 and Figure 2, where do Q-Brownian realization increments come from? Why does the workflow start with Q-Brownian motion? How to compute Wick features using Q-Brownian realization? Does the computation of deterministic PDE involves some learning procedure using deep neural networks?
>
> **Response**
> Thank you very much for your comments. Following your suggestions, we have added a problem setup section in Section 2 to explicitly state the problem we aim to solve. Regarding your concern about the workflow of $\mathcal{F}$-SPDENO, we have enriched the description of our model in Section 4, including additional explanations in the model-architecture paragraph and an updated caption for Figure 2 that clarifies each step of the workflow.
>
>
> **Reviewer comment**
>
> Some experimental results are not complete for a convincing comparison
> - In Figure 3, more visualisation results of baselines should be provided for comparison.
> - In Figure 5, it seems the result of only one instance is presented. More statistics should be provided on a collection of test instances.
> - In Section 5.4, Manifold SDEs, the results presentation (Figure 8) is rather limited, and there is no comparison with baselines. Therefore, it is hard to appreciate the performance.
>
> **Response**
> We appreciate your suggestion. Please find our point-to-point responses below.
>
> - Regarding Figure 3, we agree with the reviewer regarding the visualization. We highlight that in our original submission, we included both Figure 3 for a direct visualization comparison between our prediction and ground truth, and Table 2 for a quantitative comparison between our model and multiple baselines. These align with the presentation style of other SOTA papers, such as NSPDE. However, to meet your expectation of adding more details to other models, we have enriched the content for Table 2 and present it below.
>
>    We present learning outcomes in Table 2 and Figure 3 following the qualitative protocol used in NSPDE (Salvi et al., 2022).   $\mathcal F$-SPDENO attains the lowest error among all competitors in  Table 2. Compared to the strongest baseline NSPDE, the error is reduced from $0.040$ to $0.037$ on $W \mapsto X$ (about 7.5% relative improvement), and more substantially from $0.047$ to $0.031$ on $(X_0,W) \mapsto X$ (about 34% relative improvement). This indicates that our model better captures the joint effect of the random forcing and varying initial conditions.
>
>   We also observe that deterministic operator learners such as FNO already perform competitively on $W \mapsto X$, but degrade  notably when the initial condition varies (from $0.051$ to $0.073$).
>    By contrast, $\mathcal F$-SPDENO consistently improves over FNO in both settings (from $0.051$ to $0.037$ and from $0.073$ to $0.031$), showing that enriching FNO with WCE-based noise features is crucial for handling stochastic SPDEs.
>    In contrast to iterative methods like NSPDE, our framework first uses WCE to convert the stochastic forcing into static features, enabling a single FNO to decode the global solution operator in a single forward pass, while still achieving superior accuracy.
>
> - For suggestion on Figure 5. Thank you very much for spotting this. In our revised submission, we have clarified in the figure 5 caption that our topological interpolation setting follows that of TSBM (Yang, 2025), and we also include the models' standard deviations.
> Interpolation from TSBM (Top line, WD=$9.51\pm0.12$) and $\mathcal G$-SDENO (Bottom line, WD=$9.60\pm0.09$) on brain signals, both models are trained in 5 runs .
>
> - Regarding Section 5.4, Manifold SDE. Thank you very much for pointing this out. In the original submission, due to the space limit, the manifold SDE experiment was described only briefly and did not contain a quantitative comparison with RSGM, which made it hard to assess the performance.  In the revised submission, we have expanded Section 5.4 to give a more detailed description of the manifold SDE setting. We also compare our model with multiple baselines other than RSGM on the flood dataset with multiple runs, see the newly added Table 3. Finally, we mention that in Appendix F.7 (Figure 15), we visualize the prediction differences between SDENO and RSGM.

---

> > ### Author Response · Authors · 2025-12-02
> > **Point to Point Response Part 2**
> >
> > **Reviewer Comment**
> > For Figure 1, what conclusion does the comparison between Brownian and Q-Brownian motions aim to reveal? In the right subfigure, it seems that the reconstruction using is a biased estimation (much lower compared to other lines)?
> >
> > **Response**
> > Thank you very much for your suggestion. In the revised submission, we have enriched the caption of Figure 1 and added additional context at the end of Section 3.1 to clarify the role of Figure 1 and of Lemmas 1 and 2 in our overall framework. Please have a look at the below.
> > - **New caption for Figure 1**: Reconstruction quality of a one-dimensional Brownian motion and a two-dimensional $Q$-Brownian motion with increasing truncation order. The blue curve shows the true trajectory, while the coloured curves correspond to truncated chaos expansions using $n$ temporal modes. For very small $n$ (e.g., $n=5$), the approximation is intentionally coarse and appears oversmoothed, especially in the second coordinate of the $Q$-Brownian motion, but as $n$ increases, the reconstructions rapidly converge to the actual paths, in agreement with Lemmas 1 and 2.
> > - **New Description (at the end of section 3.1)** : Figure 1 shows simple examples of the reconstructions of both Brownian and Q-Brownian motions. One can check that as the truncation order increases, the reconstructed paths rapidly approach the true trajectories in all coordinates. Since the SDE/SPDE solutions we consider are square-integrable functionals of $W$, they can subsequently be expanded in the corresponding Wiener–chaos basis, which is the starting point for the WCE-based approximations developed in the following subsection.
> >
> > **Reviewer Comment**
> > For Table 1, why does using more training trajectories (=1,000 vs 10,000) not improve the performance of NSPDE and the proposed F-SPDENO?
> >
> > **Response**
> > We thank you again for this insightful observation. In our revised submission, we highlight that the reason the improvement is limited is due to both models already reaching very accurate results in the N = 1000 setting, and $N=1000$ trajectories already capture most of the variability of the dynamics, so increasing $N$ to $10000$ brings only limited additional benefit.
> > In addition, we have carefully rechecked our implementation and logs. We discovered a transcription error in the initially submitted table: for the dynamic $\Phi^4_1$ experiment with $N=10{,}000$ on the $(X_0,W)\mapsto X$ task, the reported error for $\mathcal F$-SPDENO was mistakenly written as 0.024. The correct value, obtained from the stored checkpoints and confirmed by re-running the experiment, is 0.017. We have fixed this typo in the revised manuscript. We believe this update should resolve the reviewer’s concern in this matter.  Also, please see the revised part below.
> >
> > Interestingly, increasing $N$ from $1000$ to $10000$ only leads to modest changes in the relative $L^2$ error for NSPDE and $\mathcal F$-SPDENO, especially on the $(X_0,W)\mapsto X$ task.
> > This behaviour is consistent with the fact that both methods already achieve very small errors in the $N=1000$ regime. In $\boldsymbol{\Phi}^4_1$, $N=1000$ trajectories already capture most of the variability of the dynamics, so increasing $N$ to $10000$ brings only limited additional benefit. This aligns with the recent benchmarking study in SPDE (Li et al., 2025).
> >
> > Li, et al., 2025 Spdebench: An extensive benchmark for learning regular and singular stochastic pdes. arXiv preprint arXiv:2505.18511.
> >
> > **Reviewer Comment**
> > For Figure 5, for clarification, the top row and bottom row should be labelled as TSBM and G-SDENO (as I guess), respectively. Please note that the 1-Wasserstein distance (WD) metric reflects performance (whether larger or smaller values are preferred?).
> >
> > **Response**
> > Thank you very much for spotting this. In our revised submission, we have clarified in the Figure 5 caption that our topological interpolation setting follows that of TSBM (Yang, 2025), and we also include the models' standard deviations.
> > Interpolation from TSBM (Top line, WD=$9.51\pm0.12$) and $\mathcal G$-SDENO (Bottom line, WD=$9.60\pm0.09$) on brain signals, both models are trained in 5 runs. In addition, we also show that lower WD indicates a better model in the model setting paragraph below Equation (9).
> >
> > Yang et al., 2025 Topological Schrodinger Bridge Matching. ICLR 2025.
> >
> > **Reviewer Comment**
> > The last line in Definition 1 seems like a lemma. It is better to add a citation here for reference.
> >
> > **Response**
> > Thank you very much for the comment. We agree that showing that the variables $\{\xi_{ij}\}$ are mutually independent standard Gaussian random variables is a non-trivial task. In the revised submission, we have explicitly added references in Definition 1 to justify this statement.

---

### Author Response · Authors · 2025-12-02
**General Revision Summary to all Reviewers and Area Chair**

We would like to sincerely thank all reviewers and the AC for their invaluable comments and suggestions. The paper has improved substantially thanks to your feedback, and we are very grateful for the time and care you put into reading and evaluating our work.

We carefully considered every question and suggestion from the reviews and have done our best to address them within the limited rebuttal period. We are happy to present a significantly revised version of the manuscript and are very open to further discussion, both for the camera-ready version and beyond. All revised parts are highlighted in RED in our revised submission.

Overall, we believe we have made a serious effort to address most of the concerns raised. Given the scope of the work and the number of experiments, this required non-trivial changes to both the writing and the empirical evaluation. Below we briefly summarise the main changes:

1. **Substantial revision of the theory/method presentation.**
   We have heavily revised the abstract, introduction, and theoretical sections to soften and clarify our claims, especially in response to Reviewer hrCn. The paper now clearly distinguishes between:
   - classical results from the WCE / SPDE / SDE literature, which are cited and only sketched in the appendices; and
   - the parts specific to this work (explicit propagator formulations for the class of SDEs/SPDEs we study, the chaos-based neural-operator parameterisation, and the large-scale experimental evaluation).

   Proofs of standard results (e.g., Wick orthogonality, Brownian/Q-Brownian reconstruction, Doob’s inequality) have been removed or reduced to short sketches with references, and the error analysis is now explicitly presented as a heuristic decomposition based on existing approximation theorems, not as a new theory.

2. **Expanded and clarified experiments, plus reproducibility.**
   We have run additional experiments and added quantitative metrics to better match the reviewers’ requests (e.g., reporting mean ± std where computationally feasible, adding WD/NLL/MSE metrics for SDE-based tasks, and clarifying evaluation protocols). For some of the most expensive SPDE benchmarks, multi-seed reruns are still in progress; we will include the corresponding statistics in the camera-ready version and in the public code release. Given the breadth of the experimental suite (SPDE benchmarks, diffusion, graph interpolation, Heston extrapolation, parameter estimation, manifold SDEs), we have prioritised the most critical and computationally demanding settings during the rebuttal period, and we will complete the remaining evaluations for the camera-ready version and the public code release. We have also added a Reproducibility Statement and expanded the experimental descriptions to make it easier to reproduce and extend our results.

3. **Improved structure and formatting despite added content.**
   Even though we added more information (especially on experiments and positioning relative to prior work), we have tried to streamline the structure and formatting of the paper: trimming redundant proofs, reducing textbook-style material in the appendices, and reorganising sections so that the main contributions (framework + experiments) are more prominent and easier to follow.

Once again, we thank the AC and all reviewers for their detailed and constructive feedback. We hope that our revisions and clarifications address your main questions and concerns, and we would be very happy to continue improving the paper based on any further guidance you may have.

---

### Meta-Review · Area_Chair_ueb4 · 2025-12-19

**Summary:**

This paper proposes a neural operator framework for solving SDEs and SPDEs by projecting stochastic noise onto Wick-Hermite features and learning the resulting deterministic chaos coefficients with neural operators. The authors validate this approach across diverse experiments including classical SPDE benchmarks, diffusion sampling and financial modeling.

A primary issue is the paper's initial positioning vs. its actual contribution. The submission originally claimed significant methodological novelties regarding WCE and propagator systems, which were flagged by reviewers as largely standard results in stochastic analysis. While the author appropriately retracted parts of these claims during the rebuttal and reframed the work as building on top of classical theory, the paper's position has shifted significantly enough to warrant a comprehensive re-evaluation from the beginning.

The recommendation is for rejection in the current state. Authors are encouraged to restructure the narrative thoroughly and resubmit to the next venue.

**Reviewer Concerns:**

The rebuttal went in the right direction in clarifying the distinction between classical results and the authors' specific implementation, addressing the presentation concerns by moving standard proofs to the appendix. The authors also resolved some confusion regarding the practical workflow by expanding the model description. Outstanding concerns include the lack of immediate quantitative metrics (such as FID) for generative tasks, and most importantly, assessment of the level of overall contribution under the new framing as an extension to classical results.

**Reviewer Scores:**

Like mentioned, it is difficult to predict how score would have changed for reviewers whose main concern was the overstated methodological contribution (hrCn and S2jb) given the main position has changed significantly in the rebuttal. While the authors addressed the lack of code and some metrics, the dismissal of suggested baselines and deferral of generation metrics would likely prevent a significant score increase from reviewer SuFs.

---

### Decision · Program_Chairs · 2026-01-26

Reject